



# Carbon-concentration and carbon-climate feedbacks in CMIP6 models, and their comparison to CMIP5 models

Vivek. K. Arora[1],  Anna Katavouta[2], Richard G. Williams[2], Chris D. Jones[3], Victor Brovkin[4], Pierre Friedlingstein[5], Jörg Schwinger[6], Laurent Bopp[7], Olivier Boucher[7], Patricia Cadule[7], Matthew A. Chamberlain[8], James R. Christian[1], Christine Delire[9], Rosie A. Fisher[10], Tomohiro Hajima[11], Tatiana Ilyina[4], Emilie Joetzjer[9], Michio Kawamiya[11], Charles Koven[12], John Krasting[13], Rachel M. Law[14], David M. Lawrence[15], Andrew Lenton[8], Keith Lindsay[15], Julia Pongratz[4,16], Thomas Raddatz[4], Roland Séférian[9], Kaoru Tachiiri[11], Jerry F. Tjiputra[6], Andy Wiltshire[3], Tongwen Wu[17], Tilo Ziehn[14]

[1]Canadian Centre for Climate Modelling and Analysis, Environment Canada, University of Victoria, Victoria, B.C., V8W 2Y2, Canada
[2]School of Environmental Sciences, Liverpool University, Liverpool, United Kingdom
[3]Met Office Hadley Centre, Exeter, United Kingdom
[4]Max Planck Institute for Meteorology, Bundesstraße 53, 20146 Hamburg, Germany
[5]College of Engineering, Mathematics and Physical Sciences, University of Exeter, Exeter, EX4 4QF, UK
[6]NORCE Norwegian Research Centre, Bjerknes Centre for Climate Research, Bergen, Norway
[7]IPSL, CNRS, Sorbonne Université, Paris, France
[8]CSIRO Oceans and Atmosphere, Hobart, Tasmania, Australia
[9]CNRM, Université de Toulouse, Météo-France, CNRS, Toulouse, France
[10]National Center for Atmospheric Research, Boulder, CO, USA and Centre Européen de Recherche et de Formation Avancée en Calcul Scientifique, (CERFACS). Toulouse, France.
[11]Research Institute for Global Change, Japan Agency for Marine-Earth Science and Technology, Yokohama 236-0001, Japan
[12]Climate and Ecosystem Sciences Division, Lawrence Berkeley National Lab, Berkeley California, USA
[13]NOAA/Geophysical Fluid Dynamics Laboratory, Princeton, New Jersey, United States of America
[14]CSIRO Oceans and Atmosphere, Aspendale, Victoria, Australia
[15]Climate and Global Dynamics Laboratory, National Center for Atmospheric Research, Boulder, CO, USA
[16]Ludwig-Maximilians University, Munich
[17]Beijing Climate Center, China Meteorological Administration, 46 Zongguancun Nandajie, Haidian District, Beijing, China
–





**Abstract**
Results from the fully-, biogeochemically-, and radiatively-coupled simulations in which $CO_2$
increases at a rate of 1% per year (1pctCO2) from its pre-industrial value are analyzed to quantify
the magnitude of two feedback parameters which characterize the coupled carbon-climate
system. These feedback parameters quantify the response of ocean and terrestrial carbon pools
to changes in atmospheric $CO_2$ concentration and the resulting change in global climate. The
results are based on eight comprehensive Earth system models from the fifth Coupled Model
Intercomparison Project (CMIP5) and eleven models from the sixth CMIP (CMIP6). The
comparison of model results from two CMIP phases shows that, for both land and ocean, the
model mean values of the feedback parameters and their multi-model spread has not changed
significantly across the two CMIP phases. The absolute values of feedback parameters are lower
for land with models that include a representation of nitrogen cycle. The sensitivity of feedback
parameters to the three different ways in which they may be calculated is shown and, consistent
with existing studies, the most relevant definition is that calculated using results from the fully-
and biogeochemically-coupled configurations. Based on these two simulations simplified
expressions for the feedback parameters are obtained when the small temperature change in
the biogeochemically-coupled simulation is ignored. Decomposition of the terms of these
simplified expressions for the feedback parameters allows identification of the reasons for
differing responses among ocean and land carbon cycle models.





## 1. Introduction

The Earth system responds to the perturbation of its atmospheric $CO_2$ concentration ($[CO_2]$),
caused by anthropogenic fossil fuel and land use change emissions of $CO_2$ or any other forcing,
via both changes in its physical climate and the biogeochemical carbon cycle. Changes in both
the physical climate and the biogeochemical carbon cycle affect each other through multiple
feedbacks. The surface-atmosphere exchange of $CO_2$ over both land and ocean is modulated by
the changes in physical climate and $[CO_2]$, and the resulting changes in $[CO_2]$ modulates the
physical climate, among other climate forcings.
The response of the Earth's carbon cycle for both land and ocean components has been
characterized in terms of carbon-concentration and carbon-climate feedback parameters which
quantify their response to changes in $[CO_2]$ and the physical climate, respectively (Friedlingstein
et al., 2006; Arora et al., 2013a). The carbon-concentration feedback ($\beta$) quantifies the response
of the carbon cycle to changes in $[CO_2]$ and is expressed in units of carbon uptake or release per
unit change in $[CO_2]$ (PgC ppm$^{-1}$). The carbon-climate feedback ($\gamma$) quantifies the response of the
carbon cycle to changes in physical climate and is expressed in units of carbon uptake or release
per unit change in global mean temperature (PgC °C$^{-1}$). The changes in physical climate, in this
framework, are expressed simply in terms of changes in global mean near surface air
temperature although, of course, the carbon cycle also responds to other aspects of changes in
climate (in particular precipitation over land and circulation changes in the ocean). The





assumption is that the effect of other aspects of changes in climate on the carbon cycle can be
broadly expressed in terms of changes in near surface air temperature. These feedback
parameters can be calculated from Earth system model (ESM) simulations globally, separately
over land and ocean, regionally, or over individual grid cells (which makes somewhat more sense
over land than over ocean) to investigate their geographical distribution (Friedlingstein et al.,
2006; Yoshikawa et al., 2008; Boer and Arora, 2010; Tjiputra et al., 2010; Roy et al., 2011; Arora
et al., 2013a). The feedback analysis has shown that the carbon-concentration feedback is
negative from the atmosphere's perspective. That is, an increase in [$CO_2$] leads to an increased
carbon uptake by land and ocean which leads to a decrease in [$CO_2$] thereby slowing $CO_2$
accumulation in the atmosphere. The carbon-climate feedback, in contrast, has been shown to
be positive in ESM simulations (at the global scale) from the atmosphere's perspective since an
increase in temperature decreases the capacity of land and ocean to take up carbon, thereby
contributing to a further increase in atmospheric $CO_2$.
The carbon-concentration and carbon-climate feedback parameters serve several purposes.
First, these feedback parameters allow comparison of models in a simple and straightforward
manner despite their underlying complexities and different model structures. Inter-model
comparisons, of course, offer several benefits as has been shown for multiple model
intercomparison projects (MIPs). Second, they allow the quantification of the contribution of the
two feedback processes to allowable anthropogenic emissions for a given $CO_2$ pathway. For
example, Arora et al. (2013) and Gregory et al. (2009) showed that the contribution of the carbon-
concentration feedback to allowable diagnosed emissions is about 4-4.5 times larger than the



carbon-climate feedback. Third, they allow the comparison of feedbacks between climate and
the carbon cycle to other feedbacks operating in the climate system as was done by Gregory et
al. (2009). Fourth, the feedback parameters can be considered as emergent properties of the
coupled carbon-cycle climate system which can potentially be constrained by observations as
Wenzel et al. (2014) attempted for the carbon-climate feedback parameter over land.
Here, we build on the work done in earlier studies that compared the strength of the carbon-
concentration and carbon-climate feedback in coupled general circulation models with land and
ocean carbon cycle components. Friedlingstein et al. (2006) (hereafter F06) reported the first
such results from the Coupled Climate Carbon Cycle Models Intercomparison Project ($C^4MIP$).
Arora et al. (2013) (hereafter A13) compared the strength of the carbon-concentration and
carbon-climate feedbacks from models participating in the fifth phase of the Coupled Model
Intercomparison Project (CMIP5, http://cmip-pcmdi.llnl.gov/cmip5/forcing.html, Taylor et al.
(2012)). The A13 study found that the strength of the two feedbacks was weaker and the spread
between models was smaller in their study than in F06. While this comparison is useful, the
primary caveat when comparing results between these two studies is that their results are based
on different scenarios. The results from the F06 study were based on the SRES A2 emissions
scenario, while those in the A13 study were based on the 1% per year increasing $CO_2$ experiment
in which the atmospheric $CO_2$ concentration increases from its pre-industrial value of around 285
ppm until it quadruples over a 140-year period (referred to as the 1pctCO2 experiment in the
framework of the Coupled Model Intercomparison Project, CMIP). The absolute values of the
feedback parameters are known to be dependent on the state of the system, the timescale of



forcing (i.e. underlying emissions/concentration scenario) and the approach used to calculate
them (Plattner et al., 2008; Gregory et al., 2009; Boer and Arora, 2010; Zickfeld et al., 2011;
Hajima et al., 2014). The varying approaches employed over the past decade have made the
cross-comparison of feedbacks among the studies and different generations of Earth System
Models difficult.

In order to address the diversity of approaches to diagnose climate carbon cycle feedbacks, and
to promote a robust standard moving forward, the C[4]MIP community has endorsed a framework
of tiered experiments (Jones et al., 2016) that builds upon the core preindustrial control and
1pctCO2 experiments performed as part of the CMIP DECK (Diagnostic, Evaluation and
Characterization of Klima) experiments (Eyring et al., 2016). Here, we compare carbon-
concentration and carbon-climate feedbacks from models participating in the C[4]MIP (Jones et al.,
2016) contribution to the sixth phase of CMIP (CMIP6, Eyring et al., 2016). To maintain continuity
and consistency, feedback parameters are derived from the 1pctCO2 experiments as was done
in A13.  The 1pctCO2 experiment is a DECK  experiment in the CMIP6 framework. All participating
modelling groups are expected to perform DECK experiments to help document basic
characteristics of models across different phases of CMIP (Eyring et al., 2016).

2. Feedbacks in the coupled climate-carbon system

We largely follow the climate carbon cycle feedbacks framework presented in A13 (which in turn
was built on F06) but with some additional modifications that are explained below. Only the



primary equations are presented here while the bulk of the framework is summarized in the
Appendix for completeness. We also provide some history of how the carbon feedbacks analysis
reached its current stage.

Carbon feedbacks analysis is traditionally based on simulations run with fully-, radiatively-, and
biogeochemically-coupled model configurations of an Earth system model. The objective of these
simulations is to isolate feedbacks discussed above. In a biogeochemically-coupled simulation
(referred to here as the BGC simulation), biogeochemical processes over land and ocean respond
to increasing atmospheric $CO_2$ while the radiative transfer calculations in the atmosphere use a
$CO_2$ concentration that remains at its preindustrial value. Small climatic changes occur in the BGC
simulation due to changes in evaporative (or latent heat) flux resulting from stomatal closure
over land (associated with increasing $[CO_2]$), changes in vegetation structure, and changes in
vegetation coverage and composition (in models which dynamically simulate competition
between their plant functional types) all of which affect latent and sensible heat fluxes at the
land surface. In a radiatively-coupled simulation (referred to here as the RAD simulation)
increasing atmospheric $CO_2$ affects the radiative transfer processes in the atmosphere and hence
climate but not the biogeochemical processes directly over land and ocean, for which the
preindustrial value of atmospheric $CO_2$ concentration is prescribed. In a fully-coupled simulation
(referred to here as the COU simulation) both the biogeochemical and the radiative processes
respond to increasing $CO_2$.





Following the F06 methodology which uses time-integrated fluxes (which are the same as the
changes in carbon pool sizes), the changes in land ($L$) or ocean ($O$) carbon pools ($\Delta C_X, X = L, O$)
can be expressed using three equations corresponding to the BGC, RAD, and COU experiments,
as shown in equation (1) (see also the Appendix).

Radiatively coupled simulation $\qquad\qquad \Delta C_X^+ = \int F_X^+ \, dt = \gamma_X T^+$ $\qquad\qquad$ (1a)
Biogeochemically coupled simulation $\qquad \Delta C_X^* = \int F_X^* \, dt = \beta_X c' + \gamma_X T^*$ $\qquad$ (1b)
Fully coupled simulation $\qquad\qquad\qquad \Delta C_X' = \int F_X' \, dt = \beta_X c' + \gamma_X T'$ $\qquad\qquad$ (1c)

where $F^+$, $F^*$, and $F'$ are the $CO_2$ flux changes (PgC year$^{-1}$), $\Delta C_X^+$, $\Delta C_X^*$, and $\Delta C_X'$ the changes in
global carbon pools (PgC), and $T^+$, $T^*$, and $T'$ the temperature changes (°C) in the RAD, BGC, and
COU simulations, respectively, and the subscript $X = L, O$ refers to either the land or ocean
model components. $c'$ is the change in [$CO_2$]. . Here and elsewhere uppercase $C$ is used to denote
pools and lowercase $c$ is used to denote atmospheric $CO_2$ concentration, [$CO_2$]. All changes are
defined relative to a pre-industrial equilibrium state represented by the pre-industrial control
simulation. In the context of a specified-concentration simulation (the 1pctCO2 experiment in
our case), $c'$ is the same in BGC and COU simulations. There is no $\beta_X c'$ term in the RAD simulation
since the biogeochemistry sees pre-industrial value of [$CO_2$] and therefore $c' = 0$ although $T^+$ is
a function of increasing $c'$ that is seen only by the radiative transfer calculations.

These equations assume linearization of the globally integrated surface-atmosphere $CO_2$ flux (for
land and ocean components) in terms of global mean temperature and [$CO_2$] change (compared



to a pre-industrial control run) and serve to define the carbon-concentration ($\beta_X$) and carbon-
climate ($\gamma_X$) feedback parameters. A similar set of equations can be written that define the
instantaneous values of the feedback parameters and is based on fluxes rather than their time-
integrated values (see equations A4 and A5 in the appendix). Both the time-integrated flux and
instantaneous flux based versions of the feedback parameters evolve over time as shown in A13.

There are several different ways in which the feedbacks ($\beta_X$ and $\gamma_X$) in a coupled climate and
carbon cycle system may be evaluated: 1) the experiments may use specified (concentration-
driven) or freely evolving (emissions-driven) [$CO_2$], 2) any two of the three configurations of an
experiment (COU, RAD, and BGC) may be used to calculate the two feedback parameters, and 3)
the experiment may be based on an idealized scenario (like the 1pctCO2 experiment) or a more
realistic emissions scenario. In addition, the small temperature change in the BGC simulation, T*,
may be ignored, and other external forcings such as nitrogen (N) deposition, or land use change,
which directly affect carbon fluxes may or may not be taken into account. The original framework
proposed by F06 used COU and BGC versions (referred to as coupled and uncoupled in the F06
study) of an emissions driven simulation for the SRES A2 scenario. The F06 framework assumed
that the small temperature change in the BGC simulation can be ignored. A13 used BGC and RAD
versions of the 1pctCO2 experiment in which the evolution of [$CO_2$] is specified and took into
account the small global mean temperature change in the BGC simulation.

With regard to the use of concentration-driven versus emissions-driven simulations, Gregory et
al. (2009) recommended the use of specified concentration simulations, which ensures



consistency of [CO₂] across models, and this recommendation has now been adopted since
CMIP5. C⁴MIP has also adopted the use of the 1pctCO2 simulation, i.e., an idealized scenario is
preferred over a more realistic scenario. This recommendation was also made by Gregory et al.
(2009). The 1pctCO2 experiment provides an ideal experiment to compare carbon-climate
interactions across models as the experiment does not include the confounding effects of other
climate forcings (including land use change, non-CO₂ greenhouse gases, and aerosols) and is a
CMIP DECK experiment, as mentioned earlier.

Using equation (1) as an example, Table 1 shows how any two combinations of the three
configurations of an experiment can be used to calculate the values of the two feedback
parameters. The A13 study showed that under the assumption of a linear system and if the
conditions $F' = F^+ + F^*$ and $T' = T^+ + T^*$ are met, i.e. if the sum of flux and temperature changes
in the RAD and BGC simulations is the same as that in the COU simulation, then all approaches
yield exactly the same solution. However, this is not the case because of the non-linearities
involved (see also Schwinger et al., 2014).

The use of BGC and RAD simulations that have only biogeochemistry or radiative forcing
responding to increases in [CO₂] to find the feedback parameters is attractive since these
simulations were designed to isolate the feedbacks. In the RAD simulation (whose purpose is to
quantify the carbon-climate feedback, $\gamma_X$) the pre-industrial global carbon pools for both land
and ocean typically decrease in response to an increase in global temperature (hence the positive
carbon-climate feedback and the negative value of $\gamma_X$). Consequently, negative values of $\gamma_X$



(positive carbon-climate feedback) are obtained when using the RAD-BGC and RAD-COU
approaches (see Table 1). If, however, $\gamma_X$ is determined using the BGC-COU approach, then $\gamma_X$ is
calculated using BGC and COU simulations in both of which the globally-summed carbon pools
for land and ocean are increasing in response to increasing [$CO_2$]. As a result, the calculated value
of $\gamma_X$ is different than that obtained using the RAD-BGC and RAD-COU approaches. In the ocean,
the RAD simulation mainly measures the loss of near-surface carbon owing to warming of the
surface ocean layer (Schwinger et al., 2014).  The RAD simulation misses the suppression of
carbon drawdown to the deep ocean due to weakening ocean circulation, because there is no
buildup of a strong carbon gradient from the surface to the deep ocean in contrast to the BGC
and COU simulations.  Therefore, the absolute value of $\gamma_X$ is smaller (less negative) when
calculated using the RAD simulation (Schwinger et al., 2014). Over land, in the RAD simulation
carbon is lost in response to increasing temperatures primarily due to an increase in
heterotrophic respiration. However, an increase in temperature also potentially increases
photosynthesis at high latitudes, and this increase compensates for carbon lost due to increased
heterotrophic respiratory losses, especially in the presence of continuously increasing [$CO_2$] seen
in the COU configuration. These are some mechanisms that lead to non-linearities. Since the
ongoing climate change (predominantly caused by increasing [$CO_2$]) is best characterized by the
COU simulation, it can be argued that feedback parameters are more representative when
calculated using the BGC-COU approach. Here, we propose to use the COU and BGC
configurations of an experiment as the standard set from which to calculate the feedback
parameters as recommended in the C[4]MIP protocol (Jones et al., 2016). However, we also
quantify the values of feedback parameters when using the RAD simulation for comparison. The



calculated values of the carbon-concentration feedback parameter ($\beta_X$) in contrast, are less
sensitive to the approach used as shown in A13.

There is no broad consensus on whether temperature change in the BGC simulation should be
assumed to be zero ($T^* = 0$) as standard practice when calculating the strengths of the
feedbacks, as done in F06. While the globally-averaged value of $T^*$ is an order of magnitude
smaller than $T'$, the spatial pattern of $T^*$ is quite different from that of $T'$. The spatial pattern of
temperature change in the COU simulation ($T'$) is dominated by radiative forcing of increased
[$CO_2$] with greater warming at high latitudes  and over land than over ocean. In contrast, the
spatial pattern of temperature change in the BGC simulation ($T^*$) is determined primarily by
reduction in latent heat flux associated with stomatal closure as [$CO_2$] increases which reduces
transpiration from vegetation (Ainsworth and Long, 2005; Bounoua et al., 1999). This process
leads to a much more spatially variable pattern of temperature change (than $T'$) and the
associated changes in precipitation patterns due to soil moisture-atmosphere feedbacks
(Chadwick et al., 2017; Skinner et al., 2017). The difference in spatial patterns of temperature
and precipitation change in the RAD versus the COU simulation is another reason that the values
of the carbon-climate feedback ($\gamma_X$) depend on the simulation used, and this is another pathway
for non-linearities to occur. A complete analysis of the effect of differences in spatial patterns of
climate change and the carbon state on the calculated value of $\gamma_X$ when using the RAD versus
the COU simulation, and if or not the assumption of $T^* = 0$ should be a standard practice, is
beyond the scope of this study but remains a topic for additional scientific investigation. In the



interim, we report here values of $\beta_X$ and $\gamma_X$ by explicitly considering $T^*$ but also assuming $T^* =$

277    0.


Following Table 1, when using results from the BGC and the COU versions of a specified-
concentration experiment the values of the feedback parameters are written as

$$\beta_X = \frac{1}{c'}\left(\frac{\Delta C_X^* T' - \Delta C_X' T^*}{T' - T^*}\right)$$    (2)
$$\gamma_X = \frac{\Delta C_X' - \Delta C_X^*}{T' - T^*}$$    (3)

Equations (2) and (3) may be rearranged to explicitly calculate the effect of the $T^* = 0$
assumption on calculated values of feedback parameters, as shown in equations (4) and (5). Here,
the $T^*$ term is retained only in the second part of the equations whose contribution becomes
zero when $T^*$ is ignored.

$$\beta_X = \frac{\Delta C_X^*}{c'} + \frac{1}{c'}\left[\frac{(\Delta C_X' - \Delta C_X^*)T^*}{(T' - T^*)}\right]$$    (4)
$$\gamma_X = \frac{\Delta C_X' - \Delta C_X^*}{T'} + \frac{(\Delta C_X' - \Delta C_X^*)T^*}{T'(T' - T^*)}$$    (5)

Finally, in regards to other external forcings such as nitrogen (N) deposition that directly affect
carbon fluxes, the C[4]MIP protocol for CMIP6 (Jones et al., 2016) recommended performing
additional simulations for BGC and COU versions of the 1pctCO2 experiment with time varying N
deposition in addition to their standard versions which keep N deposition rates at their pre-
industrial level. Simulations with N deposition can only be performed for models that explicitly





model the N cycle and its interactions with the carbon (C) cycle. The rationale for recommending
increasing N deposition, in conjunction with temperature and $CO_2$ increase, is to be able to
quantify the response of feedback parameters to this third forcing. However, here we restrict
ourselves to the traditional analysis that considers the climate and $CO_2$ forcings only. We do
highlight, however, which models include coupled C-N cycle interactions over land. Analysis of
runs with N deposition forcing is left for future studies.

2.1. Reasons for differences in feedback parameters among models

As shown later in this paper, the contribution of the second term involving $T^*$ in expressions for
the carbon-concentration ($\beta_X$) and carbon-climate ($\gamma_X$) feedback parameters (in equations 4 and
5, when using the BGC-COU approach) is around 1% to 5%.  This allows to investigate reasons for
differences in the feedback parameters across models as the expressions for the feedback
parameters can be simplified in terms of the changes in the sizes of carbon pools ($\Delta C_X'$ and $\Delta C_X^*$),
the temperature change in the COU simulation ($T'$) and the specified change in $[CO_2]$ ($c'$) as
follows.

$$\beta_X \approx \frac{\Delta C_X^*}{c'} \tag{6}$$
$$\gamma_X \approx \frac{\Delta C_X' - \Delta C_X^*}{T'} \tag{7}$$


2.1.1 Land




Over land, equations (6) and (7) can be expanded to investigate, firstly, the contributions from
changes in live vegetation pool ($\Delta C_V$) and dead litter plus soil carbon pools ($\Delta C_S$), to the strength
of the feedback parameters, since $\Delta C_L = \Delta C_V + \Delta C_S$. Secondly, equation (6) can be further
decomposed to gain insight into the reasons for differences across models, in a manner similar
to Hajima et al. (2014).
$$\beta_L \approx \frac{\Delta C_L^*}{c'} = \frac{\Delta C_V^* + \Delta C_S^*}{c'} = \left( \frac{\Delta C_V^*}{\Delta NPP^*} \frac{\Delta NPP^*}{\Delta GPP^*} \frac{\Delta GPP^*}{c'} \right) + \left( \frac{\Delta C_S^*}{\Delta R_h^*} \frac{\Delta R_h^*}{\Delta LF^*} \frac{\Delta LF^*}{c'} \right)$$

$$= \tau_{veg\Delta} . CUE_\Delta . \frac{\Delta GPP^*}{c'} + \tau_{soil\Delta} \frac{\Delta R_h^*}{\Delta LF^*} \frac{\Delta LF^*}{c'} \qquad (8)$$

$$\gamma_L \approx \frac{\Delta C_L' - \Delta C_L^*}{T'} = \frac{\Delta C_V' - \Delta C_V^*}{T'} + \frac{\Delta C_S' - \Delta C_S^*}{T'} \qquad (9)$$

The superscript * in equation (8) implies that the terms are calculated here using the BGC version
of the 1pctCO2 experiment. In equation (8), $\Delta NPP$ and $\Delta GPP$ represent the change in net and
gross primary productivity, $\Delta LF$ the change in litterfall flux, and $\Delta R_h$ the change in heterotrophic
respiration, compared to the preindustrial control experiment. The multiplicative terms in
equation (8) do indeed have some physical meaning although they are based on change in the
magnitude of quantities as opposed to their absolute magnitudes. We note here explicitly that
as such, these terms cannot be compared directly to the terms which are based on absolute
magnitudes.
The term $\frac{\Delta NPP}{\Delta GPP}$ (fraction) is the fraction of GPP (above its pre-industrial value) that is turned into
NPP after autotrophic respiratory losses are taken into account. We use the term carbon use



efficiency but subscripted by $\Delta$ ($CUE_\Delta$) to represent $\frac{\Delta NPP}{\Delta GPP}$. The subscripted $\Delta$ allows $CUE_\Delta$ to be
differentiated from $CUE$ as used in the existing literature (Choudhury, 2000) which represents
the fraction of absolute GPP that is converted to NPP rather than its change over some time
period, as well as the point that we consider globally-integrated rather than locally-derived
quantities. Similarly, the term $\frac{\Delta C_V}{\Delta NPP}$ represents a measure of turnover or residence timescale of
carbon in the vegetation pool ($\tau_{veg\Delta}$, years). The term $\frac{\Delta GPP}{c'}$ (PgC yr$^{-1}$ ppm$^{-1}$) is a measure of the
strength of the globally-integrated $CO_2$ fertilization effect. However, in the models that
dynamically simulate changes in vegetation cover, the effect of changes in vegetation coverage is
implicitly included in this term. The term $\frac{\Delta C_S}{\Delta R_h}$ is a measure of the average residence time of carbon
in the dead litter and soil carbon pools ($\tau_{soil\Delta}$, years). However, as with CUE, this quantity cannot
be compared directly to the residence time of carbon in the litter plus soil carbon pool calculated
using the absolute values of $C_S$ and $R_h$. Nor can it be compared to the changes in carbon residence
time due to the "false priming effect" associated with the increase in NPP inputs, as [$CO_2$]
increases, into the dead carbon pools (Koven et al., 2015). $\frac{\Delta R_h}{\Delta LF}$ (fraction) is a measure of the
increase in heterotrophic respiration per unit increase in litterfall rate, and $\frac{\Delta LF}{c'}$ (PgC yr$^{-1}$ ppm$^{-1}$)
indicates global increase in litterfall rate per unit increase in $CO_2$, which in principle, should be
close to the change in net primary productivity per unit increase in $CO_2$, $\left( CUE_\Delta \frac{\Delta GPP}{c'} \right)$.
Comparison of these terms across models can potentially yield insight into the reasons for large
differences in land carbon uptake across models.



### 2.1.2 Ocean

The change in the ocean carbon inventory, $\Delta C_O$, is defined by an integral of the change in the dissolved inorganic carbon, $\Delta DIC$, and density over the ocean volume,

$$\Delta C_O = 12 \ gC \ mol^{-1} \int_V \ \Delta DIC \ dV \ \times 10^{-15} \tag{10}$$

where $\Delta C_O$ is in PgC, the ocean dissolved inorganic carbon, $DIC$ in mol m$^{-3}$ and the ocean volume V in m3, and the multiplier $10^{-15}$ converts g to Pg of carbon.

To gain insight into how the ocean carbon distribution is controlled, the ocean dissolved inorganic carbon, $DIC$, may be defined in terms of separate carbon pools (Ito and Follows, 2005; Williams and Follows, 2011; Lauderdale et al., 2013; Schwinger and Tjiputra, 2018):

$$
\begin{aligned}
DIC = \ & DIC_{preformed} \quad\quad\quad + DIC_{regenerated} \\
= \ & DIC_{sat} + DIC_{disequilib} + DIC_{regenerated}
\end{aligned}
\tag{11}
$$

where the preformed carbon, $DIC_{preformed}$, is the amount of carbon in a water parcel when in the mixed layer at the time of subduction, and the regenerated carbon, $DIC_{regenerated}$, is the amount of dissolved inorganic carbon accumulated below the mixed layer due to biological regeneration of organic carbon. The preformed carbon is affected by the carbonate chemistry and ocean physics. To gain further insight into how close the ocean is to an equilibrium with the atmosphere, the preformed carbon, $DIC_{preformed}$, is further split into saturated, $DIC_{sat}$, and disequilibrium, $DIC_{disequilib}$ components. The saturated component represents the concentration in surface water fully equilibrated with the contemporary atmospheric $CO_2$



concentration. The disequilibrium component represents the extent that surface water is
incompletely equilibrated before subduction, which is affected by the strength of the ocean
circulation altering the residence time in the mixed layer and the ocean ventilation rate. Each of
these components is affected by the increase in atmospheric $CO_2$ and the changes in climate.

The change in the global ocean carbon inventory, $\Delta C_O$, relative to the preindustrial may then be
related to the global volume integral of the change in each of these DIC pools,

$$
\begin{aligned}
\Delta C_O &= \quad \Delta C_{preformed} \qquad\quad + \Delta C_{regenerated} \\
&= \quad \Delta C_{sat} + \Delta C_{disequilib} + \Delta C_{regenerated}
\end{aligned}
\tag{12}
$$

where $\Delta C_{preformed}$ is the preformed carbon inventory, $\Delta C_{sat}$ is the saturated carbon inventory,
$\Delta C_{disequilib}$ is the disequilibrium carbon inventory and $\Delta C_{regenerated}$ is the regenerated carbon
inventory.

The simplified expressions for carbon-cycle feedback parameters (6) and (7) based on the air-sea
flux changes to the ocean may then be approximated by the global ocean carbon inventory
changes,  which may be expressed in terms of  these  different global ocean carbon pools
(Williams et al., 2019):

$$
\begin{aligned}
\beta_O \approx \frac{\Delta C_O^*}{C'} &= \frac{\Delta C_{preformed}}{C'} \qquad\quad + \frac{\Delta C_{regenerated}}{C'} \\
&= \frac{\Delta C_{sat}}{C'} + \frac{\Delta C_{disequilib}}{C'} + \frac{\Delta C_{regenerated}}{C'}
\end{aligned}
\tag{13}
$$



$$\gamma_O \approx \frac{\Delta C_O' - \Delta C_O^*}{T'} \quad = \frac{\Delta C_{preformed}' - \Delta C_{preformed}^*}{T'} \qquad + \frac{\Delta C_{regenerated}' - \Delta C_{regenerated}^*}{T'}$$


$$= \frac{\Delta C_{sat}' - \Delta C_{sat}^*}{T'} + \frac{\Delta C_{disequilib}' - \Delta C_{disequilib}^*}{T'} + \frac{\Delta C_{regenerated}' - \Delta C_{regenerated}^*}{T'}$$

396 (14)

The anomalies for each of these carbon pools are calculated as
$$\Delta DIC_{regenerated} = -R_{CO}\,\Delta AOU + \frac{1}{2}(\Delta Alk - \Delta Alk_{pre} - R_{NO}\,\Delta AOU) \qquad (15)$$
$$\Delta DIC_{sat} = f(p\text{CO}_2^{atm}, T_o, S_o, P, Si, Alk_{pre})_t - f((p\text{CO}_2^{atm}, T_o, S_o, P, Si, Alk_{pre})_{t=0} \qquad (16)$$
$$\Delta DIC_{disequilib} = \Delta DIC - \Delta DIC_{regenerated} - \Delta DIC_{sat} \qquad (17)$$
where $R_{CO}$ and $R_{NO}$ are constant stochiometric ratios, $\Delta AOU$ is the change in apparent oxygen
utilization from its pre-industrial value (where preformed oxygen is assumed to be approximately
saturated with respect to atmospheric oxygen), $\Delta Alk$ is the change in alkalinity, $T_o$ and $S_o$ are the
ocean temperature and salinity, respectively, P and Si are the phosphate and silicate
concentrations, and $\Delta Alk_{pre}$ is the change in preformed alkalinity (Ito and Follows, 2005;
Appendix of Lauderdale et al., 2013; Williams and Follows, 2011). In equation (16), $\Delta DIC_{sat}$ is
calculated using values of $p\text{CO}_2^{atm}$, $T_o$, $S_o$, P, Si, and $Alk_{pre}$ at time $t$ and the pre-industrial values at
time $t$=0. The preformed alkalinity is estimated from a multiple linear regression using salinity
and the conservative tracer PO (PO=$O_2$-$R_{o2:P}$P) (Gruber et al., 1996), with the coefficients of this
regression estimated based on the upper ocean (first 10 meters) alkalinity, salinity, oxygen and
phosphate in each model. The small contribution from minor species (borate, silicate, phosphate)
to the alkalinity is removed from the total alkalinity before using it for estimates of the carbon
system following the algorithm of (Follows et al., 2006). Our diagnostics of the ocean feedbacks



and carbon pools depend primarily upon changes in DIC, the preformed and regenerated pools,
relative to the pre industrial, although differences in the pre-industrial ocean do slightly affect
the saturated DIC due to the non-linearity of the carbonate chemistry.

## 3. Model descriptions

Table 2 summarizes the primary features of the eleven comprehensive ESMs that contributed
results to this study. Brief descriptions of land and ocean carbon cycle components of these ESMs
are provided in the Appendix. The eleven ESMs, in alphabetical order, are the 1) Commonwealth
Scientific and Industrial Research Organisation (CSIRO) ACCESS-ESM1.5, 2) Beijing Climate Centre
(BCC) BCC-CSM2-MR, 3) Canadian Centre for Climate Modelling and Analysis (CCCma) CanESM5,
4) Community Earth System Model, version 2 (CESM2), 5) Centre National de Recherches
Météorologiques (CNRM) CNRM-ESM2-1, 6) Institut Pierre-Simon Laplace (IPSL) IPSL-CM6A-LR,
7) Japan Agency for Marine-Earth Science and Technology (JAMSTEC) in collaboration with the
University of Tokyo and the National Institute for Environmental Studies (Team MIROC) MIROC-
ES2L, 8) Max Planck Institute for Meteorology (MPI) MPI-ESM1.2-LR, 9) Geophysical Fluid
Dynamics Laboratory (GFDL) NOAA-GFDL-ESM4, 10) Norwegian Climate Centre (NCC) NorESM2-
LM, and 11) United Kingdom (UK) UKESM1-0-LL.

In contrast to the A13 study where only two of the eight participating comprehensive ESMs had
terrestrial N cycle implemented and coupled to their C cycle, in this study six of the eleven



participating ESMs represent coupling of terrestrial C and N cycles. These six models are the
ACCESS-ESM1.5, CESM2, MIROC-ES2L, MPI-ESM1.2-LR, NorESM2-LM, and UKESM1-0-LL. Note
that CESM2 and NorESM2-LM employ the same land surface component – the version 5 of the
Community Land Model (CLM5) so we expect the land carbon cycle to respond very similarly in
the two models. Three of the ESMs have land components which dynamically simulated
vegetation cover and competition between their PFTs - NOAA-GFDL-ESM4, MPI-ESM1.2-LR, and
UKESM1-0-LL.

4. Results

4.1. Global surface $CO_2$ fluxes and temperature change

Figure 1 shows the simulated changes in temperature in the three model configurations (COU,
BGC, and RAD) of the 1pctCO2 experiment. The values show the model mean and the range
across the ten participating models, since results from the RAD configuration of the NorESM2-
LM model were not available at the time of writing of this manuscript. Here and in subsequent
figures, model mean results are also shown for the eight comprehensive ESMs that participated
in the A13 study to allow a direct comparison between CMIP5 and CMIP6 models. The eight
models in the A13 study are a subset of eleven models considered in this study although they
have been updated since CMIP5.



As expected, temperature change is higher in the COU and RAD simulations, than in the BGC
simulation, since the radiative forcing responds to increasing [$CO_2$] in these simulations. The small
temperature change in the BGC simulation is due to a number of contributing but also
compensating factors: 1) reduction in transpiration, and hence latent heat flux, due to stomatal
closure in response to increasing [$CO_2$] (Cao et al., 2010), 2) increase in vegetation leaf area index
(LAI), which decreases land surface albedo and hence increases absorbed solar radiation, 3)
increase in vegetation fraction in models that explicitly simulate competition between their plant
functional types (PFTs) over land (NOAA-GFDL-ESM4, MPI-ESM1.2-LR, and UKESM1-0-LL) which
also leads to reduced land surface albedo. As a result, temperature change in the COU simulation
is higher than in the RAD simulation since these biogeochemical processes are active and
contribute to a small additional warming. This is seen in panel (a) for CMIP6 models and panel
(b) for CMIP5 models.

When comparing CMIP5 and CMIP6 models, the CMIP6 models are on average slightly warmer
than CMIP5 models in the COU and RAD simulations. In Figure 1a, the globally-averaged near
surface temperature change at $CO_2$ quadrupling in the fully-coupled simulation is 5.00 °C (4.87
°C when NorESM2-LM is included) in CMIP6 models, compared to 4.74 °C in CMIP5 models. The
globally-averaged temperature change at $CO_2$ quadrupling in the fully-coupled simulation for the
eight models that are common to this (CMIP6) and the A13 (CMIP5) studies, are 4.97 and 4.74
°C, respectively.  The temperature change in the BGC simulation in CMIP6 models (0.24 °C) is,
however, slightly smaller than in the CMIP5 models (0.26 °C). The values in Figure 1 for
participating CMIP5 models are slightly different than those reported in A13 study because those



numbers also included the UVic Earth System Climate Model (an intermediate complexity model)
which we have omitted here to keep the comparison consistent between comprehensive ESMs.
In addition, in contrast to A13, the temperature at the end of a simulation in this study is
calculated after fitting a polynomial to the model mean values rather than using the actual model
mean value at the end of the simulation which can be higher or lower than that calculated using
the polynomial fit due to inter-annual variability.

Figure 2 and 3 show simulated model mean values and the range across models for annual
simulated atmosphere-land and atmosphere-ocean $CO_2$ fluxes and their cumulative values for
participating CMIP6 and CMIP5 models from the fully-, biogeochemically- and radiatively-
coupled configurations of the 1pctCO2 experiment. Here, in contrast to Figure 1, results from all
eleven models are included since model mean cumulative atmosphere-land and atmosphere-
ocean $CO_2$ fluxes are not particularly sensitive to inclusion/exclusion of the NorESM2-LM models
for which results from the RAD simulation were not available. The general results from CMIP6
models are broadly similar in nature to those from CMIP5 models, as would be expected, with
higher annual and cumulative values of atmosphere-land and atmosphere-ocean $CO_2$ fluxes in
the BGC simulation compared to the COU simulation in which the radiative warming caused by
increasing $CO_2$ weakens the land and ocean sinks.  In the RAD simulation, where land and ocean
carbon cycle components do not respond to increasing [$CO_2$], both components lose carbon, for
reasons discussed below.



Over land, the model mean rate of increase of atmosphere-land $CO_2$ flux declines and even
becomes negative in the COU and BGC simulations as the terrestrial $CO_2$ fertilization effect
saturates and the carbon pools build up, which increases the respiratory losses. The biggest
difference between the CMIP5 and CMIP6 models is that the cumulative land carbon uptake in
the COU simulation is about 25 % higher in CMIP6 (635 ± 258 PgC, mean ± standard deviation)
models than in CMIP5 (505 ± 297 PgC) models, although this increase is not statistically significant
across the model ensemble (Mann-Whitney test). The cumulative value of carbon loss in the RAD
simulation is similar in both CMIP6 and CMIP5 models, 250 ± 121 vs. 252 ± 158 PgC, respectively.
This carbon loss occurs due both to increased heterotrophic respiration per unit carbon mass and
reduced GPP (and consequently NPP) in the RAD simulation (not shown).  While NPP declines
globally in response to increase in temperature, mid- to high-latitude net primary production
increases (Qian et al., 2010) so the reduction in global NPP comes largely from the reduction in
the tropics. The large range across land carbon cycle models, seen also in earlier F06 and A13
studies, has not meaningfully declined for CMIP6 models participating in this study and its
implications will be discussed in more detail in Section 5. This is also seen later in Figure 6 which
compares the absolute magnitude and the standard deviation of the strength of the feedback
parameters from CMIP5 and CMIP6 models.

Over the ocean, the response to increasing [$CO_2$] and changing climate remains fairly similar
across CMIP5 and CMIP6 models. The cumulative ocean carbon uptake in the COU simulation is
593 ± 54 and 611 ± 50 PgC in CMIP6 and CMIP5 models, respectively. Unlike the land uptake,
however, the ocean carbon uptake does not saturate over the length of the simulation in the BGC



simulation (Figure 3, panels a and b); it keeps on increasing albeit at a declining rate. The
cumulative ocean carbon loss in the RAD simulation is 23 ± 19 and 37 ± 17 PgC in CMIP6 and
CMIP5 models, respectively, and associated with warmer temperatures which reduce $CO_2$
solubility (Goodwin and Lenton, 2009).

Figure 4 shows results from individual CMIP6 models for which model means and ranges were
shown in Figures 1, 2, and 3. Figure 4 allows identification of models which behave differently
compared to the majority of models. In Figure 4, panels a and c, CanESM5 shows the largest
temperature increase, and NorESM2-LM and MIROC-ES2L the smallest, in response to increase
in [$CO_2$] for the COU and RAD simulations, respectively. For cumulative atmosphere-land $CO_2$ flux
in the COU simulation (panel d), CanESM5 simulates the largest land carbon uptake and ACCESS-
ESM1.5 the smallest. This is not the case for the BGC simulation (panel e) where land carbon
uptake from the BCC-CSM2-MR and CNRM-ESM2.1 are the largest among all models, while land
carbon uptake from the ACCESS-ESM1.5 is the lowest. Finally, in the RAD simulation (panel f) the
loss of carbon from land in response to increasing temperatures is lowest in the MPI-ESM1.2-LR
and largest in the BCC-CSM2-MR. Over the ocean, while most models behave very similarly, the
carbon uptake in the BCC-CSM2-MR, ACCESS-ESM1.5, and NOAA-GFDL-ESM4 are larger than
most models in the COU and BGC simulations. In the RAD simulation, almost all models simulate
a loss of carbon from the ocean, but the CNRM-ESM2.1 shows a small uptake. Reasons for
divergent response of some models are presented later.



As in F06 and A13, the range in cumulative atmosphere-land $CO_2$ fluxes among models at the end
of the simulation, in response to changes in atmospheric $CO_2$ concentration and surface
temperature, is three to four times larger than for the atmosphere-ocean $CO_2$ fluxes.

4.2. Carbon budget terms

Figure 5a shows the carbon budget components of the diagnosed cumulative fossil fuel emissions
at the end of the 140-year period of the 1pctCO2 COU experiment when $CO_2$ concentration
quadruples ($\tilde{E}_{4\times CO2}$ or simply $\tilde{E}$), from CMIP6 models. Cumulative emissions can similarly also
be calculated at 2×CO2 ($\tilde{E}_{2\times CO2}$). The term "carbon budget" in this context refers to the
accounting of carbon internal to individual ESMs. The sum of ocean ($\Delta C'_O$) and land ($\Delta C'_L$) sinks
and the resulting atmospheric $CO_2$ growth rate ($\Delta C'_A$) yields cumulative fossil fuel emissions
which are consistent with the specified $CO_2$ pathway (the 1pctCO2 scenario in this case) as
indicated in the appendix. The corollary to this is that, in a specified emissions simulation, if the
respective fossil fuel emissions were to be used in their models, each model will yield $CO_2$
concentrations that rise at a rate of 1% per year. The term "diagnosed" implies that the
cumulative fossil fuel emissions are calculated after the fact from changes in atmosphere, land
and ocean carbon pools in the specified-concentration 1pctCO2 experiment. In Figure 5a, the
results are arranged in an ascending order according to models' diagnosed cumulative fossil fuel
emissions. Figure 5b shows the terms of the budgets as fractional components for atmosphere
($A$), land ($L$) and ocean ($O$) based on equation (A7), where $f_A$ is the airborne fraction of emissions



and $f_L$ and $f_O$ are the fractions of emissions take up by land and ocean, respectively. More details
are provided in the Appendix.
$$\Delta C'_A + \Delta C'_L + \Delta C'_O = \int_0^t E \, dt = \tilde{E} \tag{18}$$
$$f_A + f_L + f_O = 1 \tag{19}$$
All panels in Figure 5 identify models whose land component includes a representation of the N
cycle – the cumulative land carbon uptake (panels a and c) and fractional emissions taken up by
land (panels b and d) for these models are shown in red. Finally, model mean values are also
shown for all models and for models whose land components include and do not include a
representation of the land N cycle. For comparison, panels c and d in Figure 5 show the same
results but for CMIP5 models reported in A13.

Consistent with Figure 4, and CMIP5 results reported in the A13 study, the differences among
models are primarily due to the diverse response of the land carbon cycle components. While
the model mean cumulative carbon uptake by the ocean is fairly similar between participating
CMIP5 (611 ± 50 PgC) and CMIP6 (593 ± 54 PgC) models, the land uptake is higher in CMIP6 (635
± 258 PgC) compared to CMIP5 (505 ± 297 PgC) models, as mentioned earlier. This is the case
even when the CanESM5, the model with the largest land carbon uptake, is omitted from CMIP6
models (model mean land carbon uptake for the remaining ten models is 578 ± 185 PgC). As a
result, model mean cumulative diagnosed emissions from CMIP6 models (3031 ± 242 PgC) are
about 4% higher than for CMIP5 models (2927 ± 294 PgC). In Figure 5a, the land carbon uptake



in CESM2 (656 PgC) and NorESM2-LM (652 PgC) model are very similar; as noted above these
models employ the same land component.

Model mean estimates that are reported separately for models whose land component do and
do not include a representation of N cycle, for both CMIP5 and CMIP6 models, show that model-
mean land carbon uptake is lower for models that explicitly represent the N cycle. As a
consequence, the airborne fraction of emissions is also higher for models that represent land N
cycle and their diagnosed cumulative fossil fuel emissions are lower (Figure 5).

Figure 5a and 5c allow direct comparison of models from the same modelling group. CanESM2,
from the Canadian Centre for Climate Modelling and Analysis, which had below average land
carbon uptake among CMIP5 models, has evolved to CanESM5, a model with the largest land
carbon uptake among CMIP6 models. The reason for this is an increase in the strength of its $CO_2$
fertilization effect as explained in Arora and Scinocca (2016). CESM1, which had one of the lowest
land carbon uptake among CMIP5 models, because of its apparently excessive nitrogen limitation
effect in CLM4, has evolved to CESM2 (with CLM5 land component) with near average land
carbon uptake among CMIP6 models. The transition of CLM from CLM4 to CLM5, and the
reduction in its nutrient constraints on photosynthesis and the parametric controls on
fertilization responses are discussed in Wieder et al. (2019) and Fisher et al. (2019), respectively.
The land carbon uptake in MIROC-ESM increased from the lowest among CMIP5 models (149
PgC) to 701 PgC for MIROC-ES2L, among CMIP6 models, due to a new terrestrial biogeochemical



component (Ito and Inatomi, 2012). Although the $CO_2$ fertilization effect in this new land model
is weaker likely due to the incorporation of the nitrogen cycle, the model yields relatively higher
NPP (Hajima et al., 2019a), due to a higher $CUE_\Delta$ (as confirmed later in section 4.4.1). The land
carbon uptake in the IPSL-CM5A-LR model decreased from being the second largest in CMIP5
models (741 PgC) to below average for the IPSL-CM6A-LR model (477 PgC) due to implementation
of terrestrial photosynthesis downregulation, as a function of $CO_2$ concentration, which leads to
a decrease in GPP across all latitudes, with the largest decrease in the tropics.
The ocean carbon uptake in the IPSL model decreased from being the largest among CMIP5
models at 670 PgC in IPSL-CM5A-LR to 579 PgC for IPSL-CM6A-LR, and this is attributed to a
greater ocean stratification in the IPSL-CM6A-LR. The annual mean mixed layer depth is 46.7 m
and 40.2 m in IPSL-CM5A-LR and IPSL-CM6A-LR, respectively. While NorESM1-ME was one of the
CMIP5 models with the largest ocean carbon uptake (667 PgC), NorESM2-LM has an ocean
carbon uptake (599 PgC) close to the CMIP6 model mean. This is a consequence of changes in
the simulated (shallower depth and weaker strength) Atlantic meridional overturning circulation
and reduced mixed layer biases particularly at high latitudes (less deep winter mixing). Due to
these modifications, the efficiency of carbon export below the mixed layer in NorESM2-LM is
considerably reduced compared to the NorESM1-ME. This, in turn, leads to less excess carbon
stored in the North Atlantic Deep Water (below 2000 m) as well as in the Antarctic Intermediate
Water. For the MPI ESM, the decrease in land carbon uptake from 825 PgC in MPI-ESM-LR for
CMIP5 to 586 PgC in MPI-ESM1.2-LR for CMIP6 is associated with implementation of nitrogen
cycle model (Goll et al., 2017) and a new soil carbon model YASSO (Goll et al., 2015). Compared
to its predecessor HadGEM2-ES, UKESM1 represents a prognostic treatment of terrestrial



nitrogen including its impact on carbon storage in vegetation biomass and soil organic matter.
Limitation on terrestrial productivity from available nitrogen is the main reason for reduced land
carbon storage in UKESM1-0-LL (408 PgC) compared to HadGEM2-ES (768 PgC).

Figure A1 in the appendix shows the version of Figure 5 but at the time of $CO_2$ doubling (at year
70). Interestingly, the ordering of the models according to their diagnosed cumulative emissions
at 2×$CO_2$ is different from that at 4×$CO_2$. As expected, however, the model mean fractional
emissions taken up by land and ocean at 2×$CO_2$ are higher than at 4×$CO_2$, because both land and
ocean carbon sinks relatively weaken as $CO_2$ continues to increase.

## 4.3. Feedback parameters
Figure 6, panels a and b, compares the carbon-concentration ($\beta_L$) and carbon-climate feedback
($\gamma_L$) parameters over land from participating CMIP6 models. The plots show feedback parameters
from different models as coloured dots but also their mean ± 1 standard deviation as a box. The
feedback parameters are calculated using all of the four approaches that are summarized in Table
1 to illustrate their sensitivity to the approach used. In addition, models whose land component
includes a representation of the N cycle are identified by an additional circle around their
coloured dots.  Figure 6 also shows the mean ± 1 standard deviation values separately for models
that do and do not include a representation of the land N cycle using the BGC-COU approach, in
an attempt to understand the reason for the diverse responses of the land models. Results from
CMIP5 models in the A13 study are shown in a similar format for comparison in panels c and d.






Three primary observations can be made from Figure 6. First and foremost, the spread in the
magnitude of carbon-concentration and carbon-climate feedback over land in CMIP6 models is
of similar magnitude to that of CMIP5 models. Second, the carbon-climate feedback ($\gamma_L$) is more
sensitive to the approach used (and hence the type of simulations used) to derive its value than
the carbon-concentration feedback ($\beta_L$). Third, in the model mean sense, the absolute strength
of the feedback parameters is weaker for models that include a representation of the N cycle, for
both CMIP5 and CMIP6 models. Both the carbon gain due to increase in atmospheric $CO_2$
concentration and the carbon loss due to increase in globally average temperature in models
with representation of land N cycle is much lower than models that do not include the N cycle.
This response is most likely explained by the N limitation of photosynthesis as $CO_2$ increases and
additional release of N from dead organic matter as warming increases which boosts productivity
thereby compensating for carbon lost due to increased respiratory losses, as also discussed in
A13. The values of the feedback parameters, however, overlap between models that do and do
not include a representation of the N cycle, given the wider spread in the feedback parameter
values among models that do not include a representation of land N cycle, compared to models
that do.

Figure 7, panels a and b, compare the carbon-concentration ($\beta_O$) and carbon-climate feedback
($\gamma_O$) parameters over the ocean from participating CMIP6 models. As in Figure 6, the feedback
parameters are calculated using all of the four approaches that are summarized in Table 1 and





results from CMIP5 models are shown for comparison in panels c and d. For both CMIP5 and
CMIP6 models, the absolute spread in the magnitude of the feedback parameters across the
participating models is an order of magnitude smaller for the ocean C cycle component compared
to the land C cycle component, as was also seen in F06 and A13. Similar to the land, the calculated
values of the ocean carbon-climate feedback ($\gamma_O$) are more sensitive to the approach used (and
hence the type of simulations used) than the ocean carbon-concentration feedback ($\beta_O$). In
agreement with Schwinger et al. (2014), the absolute values of $\gamma_O$ are 2-3 times larger when
calculated using the COU and BGC simulations, compared to cases when RAD simulation is used,
for reasons mentioned earlier. Figures 6 and 7 show also that while the strength of the carbon-
concentration feedback is similar over land and ocean, the strength of the carbon-climate
feedback parameter over ocean is much weaker than over land.

Section A2 in the appendix discusses how Figures 6 and 7 and corroborate existing studies for the
preferred use of the BGC and COU simulations for finding the feedback parameters. Figure 6 and
7 also show that the effect of assuming T* (the temperature change in the BGC simulation) zero
is around 1% for the calculated value of the carbon-concentration feedback parameter ($\beta_X, X =$
$L, O$) and around 5% for the carbon-climate feedback parameter ($\gamma_X, X = L, O$). This small effect
of T* on the calculated global values of the feedback parameter allows investigation of the
reasons for differences among model by using simplified forms of $\beta_X$ and $\gamma_X$ as presented in
equations (6) and (7).



For completeness, Table A1 in the appendix summarizes the values of feedback parameters for
both land and ocean from CMIP6 and CMIP5 models (corresponding to Figures 6 and 7) at $4\times CO_2$
but also at $2\times CO_2$. Table A1 also shows the value of parameter $\alpha$, the linear transient climate
sensitivity to $CO_2$, following F06 (their equation 6) which is calculated as

693                                        $$T' = \alpha\, c'$$                    (20)

at 4 x$CO_2$.

4.4. Reasons for differences among models
4.4.1 Land
Equations (8) and (9) in Section 2.1.1 are used to gain insight into reasons for differing responses
of land models. In the BGC-COU approach and assuming T*=0 (equation 8), the carbon uptake in
the BGC simulation is used to calculate the carbon-concentration feedback parameter ($\beta_L$).
Figure 8 shows how this carbon uptake over land is separated into vegetation and soil+litter
components both in absolute (panel a) and fractional terms (panel b). The models are arranged
from lowest to highest in terms of their land carbon uptake in the BGC simulation. The
partitioning into vegetation and soil+litter components is not shown for the BCC-CSM2-MR
model because total land carbon uptake in this model exceeded the sum of changes in the
vegetation and soil+litter carbon pools by more than 10% likely because of incomplete
accounting of pool sizes. Figure 8b shows that models vary widely in terms of how the carbon
uptake over land is split into vegetation and soil+litter components. The model mean values





indicate that slightly more of the carbon sequestered is allocated to vegetation (55%) than to the
soil+litter pools (45%).

Figure 9 shows the individual components of equation (8) which contribute to terms
corresponding to changes in vegetation ($\Delta C_V$) and soil+litter ($\Delta C_S$) carbon pools. Panel (a) of
Figure 8 is repeated in Figure 9 for easy correspondence of individual terms with their models.
The model mean values of individual terms do not take into account the results from the BCC-
CSM2-MR model. In essence, the terms in Figure 9 are emergent properties of the land models
of the individual ESMs and result from their multiple interacting processes. The comparison of
the individual terms of equation (8) provides additional insight into the reasons for differences in
land models. For example, the CNRM-ESM2-1 model has the highest land carbon uptake among
all models in the BGC simulation. However, this is not caused by a strong $CO_2$ fertilization effect
(the $\frac{\Delta GPP}{c'}$ term), but rather by the relatively high $\tau_{veg\Delta}$ and $\tau_{soil\Delta}$ values. The $CO_2$ fertilization
effect is strongest for the three models that simulate vegetation cover dynamically ($\frac{\Delta GPP}{c'} =$
$0.141, 0.128,$ and $0.117$ PgC yr$^{-1}$ ppm$^{-1}$ for NOAA-GFDL-ESM4, MPI-ESM1.2-LR, and UKESM1-0-
LL, respectively) since the $\frac{\Delta GPP}{c'}$ term also implicitly includes the effect of increasing vegetation
cover as $CO_2$ increases. The tree cover in the NOAA-GFDL-ESM4 model, for example, increases in
the BGC simulation – particularly in dry, high-latitude regions above 50° N (not shown). However,
these models do not simulate the largest land carbon uptake because of their lower than average
$\tau_{veg\Delta}$ and $\tau_{soil\Delta}$ values. The $\frac{\Delta GPP}{c'}$ term is unable to capture the $CO_2$ fertilization effect separately



from increasing vegetation cover and this illustrates the challenge in comparing models that do
and do not simulate vegetation cover dynamically. The CanESM5 model exhibits higher than
average land carbon uptake despite its near average strength of the $CO_2$ fertilization effect, and
$\tau_{veg\Delta}$ and $\tau_{soil\Delta}$ values. However, its $CUE_\Delta$ is the highest and therefore a much larger fraction of
GPP is converted to NPP. Although $CUE_\Delta$ is not the same as $CUE$, we found that $CUE_\Delta$ and $CUE$
(calculated at the end of the 1pctCO2 simulation at 4xCO$_2$) are strongly correlated with a
correlation of around 0.90 (not shown). Similarly, $\tau_{veg\Delta}$ is strongly correlated, with a correlation
of 0.96, to $\tau_{veg} = C_V/NPP$ calculated at the end of the simulation. The ACCESS-ESM1.5 model
exhibits the lowest land carbon uptake because of its weak $CO_2$ fertilization effect and the lowest
$CUE_\Delta$ of all models. Finally, the $\frac{\Delta R_h}{\Delta LF}$ term shows the least variability across models, which is
reflective of the fact that the magnitude of the heterotrophic respiration flux is dominated by
NPP inputs into the dead carbon pools (Koven et al., 2015). Several of these individual terms are
strongly correlated. The $\frac{\Delta GPP}{c'}$ and $\frac{\Delta LF}{c'}$ terms have a correlation of 0.77, and $CUE_\Delta\frac{\Delta GPP}{c'}$ and
$\frac{\Delta LF}{c'}$ have a correlation of 0.94, since a stronger $CO_2$ fertilization effect also implies a larger litter
fall flux per unit $CO_2$. Surprisingly, $CUE_\Delta$ and $\tau_{veg}$ are negatively correlated (correlation = –0.49)
across models indicating that models which retain a higher fraction of GPP as NPP typically get
rid of vegetation carbon sooner via litter fall as indicated by a faster turnover of vegetation (lower
$\tau_{veg}$), there by partially compensating for higher $CUE_\Delta$.



While Figure 9 investigates reasons for differences among models that lead to different values of
their carbon-concentration feedback over land ($\beta_L$), Figure 10 investigates the reasons for varying
magnitudes of the carbon-climate feedback over land ($\gamma_L$). In equation (9), $\gamma_L$ is a function of
change in land carbon (divided into vegetation and soil+litter components) in the COU relative to
the BGC simulation and the temperature change in the COU simulation ($T'$). Over land, the higher
temperatures in the COU relative to the BGC simulation affect both autotrophic and
heterotrophic respiratory fluxes, from live and dead vegetation pools, respectively, but also gross
photosynthesis rates. The primary effect of this temperature change in COU versus the BGC
simulation is the loss of carbon from the soil+litter carbon pool (hence the negative sign of $\gamma_L$ for
most models, Figure 6b and 6d) but changes in the vegetation carbon pool also occur. Although
$\gamma_L$ also depends on $T'$, Figure 10 arranges models in order from largest to smallest loss of land
carbon in COU relative to the BGC simulation to illustrate the varying response of the models.
This ordering of models changes slightly if the carbon loss (or gain in the CanESM5 model) is
divided by the temperature change  $T'$ in the COU simulation (yielding the value of $\gamma_L$ which
assumes T*=0 as in equation 9).

As shown in Figure 10, all models lose carbon from the soil+litter carbon pool but with widely
varying magnitudes. Although typically smaller than the change in soil+litter carbon pool, the
change in the vegetation carbon pool in the COU relative to the BGC simulation is not of the same
sign across models. Six of the eleven participating models lose carbon in the vegetation pool in
the COU relative to the BGC simulation thereby contributing to increasing the absolute
magnitude of $\gamma_L$, while the remaining five exhibit an increase in the vegetation carbon pool





thereby decreasing the absolute magnitude of $\gamma_L$. The largest increase in the vegetation carbon
pool is seen in the CanESM5 model that more than compensates for the carbon loss from the
soil+litter carbon pool yielding a positive value of $\gamma_L$ in contrast to other models. This is one of
the few times a positive value of $\gamma_L$ is seen in an Earth system model. Preliminary analysis of
CanESM5 data shows the increase in vegetation carbon, in the COU relative to the BGC
simulation, is caused by the increase in GPP and the resulting vegetation growth at mid-to-high
latitudes in response to warming temperatures and increasing $CO_2$. Interestingly, this doesn't
happen at 2×$CO_2$ (see Table A1 in the Appendix). At 2×$CO_2$ $\gamma_L$ is still negative for CanESM5.

The loss in land carbon in the COU relative to the BGC simulation (except the CanESM5 model
that gains carbon), indicated by the orange bar in Figure 10, is strongly correlated with the carbon
gain in the BGC simulation (Figure 4e) (correlation is 0.59 for all models and 0.87 when CanESM5
is excluded) but not with the absolute amount of total land carbon. Figure A2 in the appendix
shows the absolute amount of carbon in soil+litter and vegetation pools, and their change from
the beginning, for the BGC simulation. The models vary widely in terms of the absolute size of
the carbon pools, especially for the soil+litter pool. There are two implications of models losing
more carbon in the COU relative to BGC simulation when they take up more carbon in the BGC
simulation alone. First, the transient behaviour of a model is determined primarily by its response
of $CO_2$ and temperature perturbations and not by the absolute amount of land carbon. Second,
that carbon-concentration ($\beta_X$) and carbon-climate ($\gamma_X$) feedback parameters must be correlated
as well. Indeed, this is the case over land for both CMIP5 and CMIP6 models, but also true for
ocean feebacks although the correlations are somewhat weaker over the ocean. These



correlations are shown in Table 3 and are negative since higher positive values of $\beta_X$ are
correlated with higher negative values of $\gamma_X$ indicating that models that take up more carbon
with increasing $CO_2$ also release more carbon when they "see" the associated higher
temperatures.

4.4.2 Ocean
The time-integrated air-sea flux of carbon provides the dominant contribution to the increase in
the global ocean carbon through changes in the DIC inventory. However, the global ocean carbon
inventory is also affected by the land to ocean carbon flux from river runoff, and the carbon burial
in ocean sediments (see Table A2 in the appendix).

Ocean carbon cycle feedbacks are defined in terms of ocean carbon inventory changes for the
COU simulation, and the differences in COU relative to the BGC simulation. To fully understand
the ocean carbon-cycle feedbacks, it is necessary to understand the ocean carbon distributions
for the preindustrial and then analyze the carbon anomalies relative to the preindustrial for these
climate model experiments.

4.4.2.1 Ocean carbon distribution



The ocean dissolved inorganic carbon distribution, DIC, is controlled by a combination of physical,
chemical and biological processes. For the preindustrial period, there is less DIC in warmer waters
of the upper ocean and more DIC in colder mid-depth and bottom waters (Figure 11a, 12a);
illustrated here for UKESM1-0-LL as a representative example and Figs S1 to S7 show similar
distributions for all the diagnosed Earth system models. The vertical extent of the low DIC follows
the undulations of the thermocline, which is defined by strong vertical temperature and density
gradients, and is deeper over the subtropical gyres at 30°N and 30°S, and shallower in the
equatorial zone  and at high latitudes. The greater DIC at depth is a consequence of greater
solubility in colder waters and the accumulation of DIC from the regeneration of  organic matter.

To gain insight into how the ocean carbon distribution is controlled, the DIC is separated into
three pools, $DIC_{sat}$, $DIC_{disequilib}$, and $DIC_{regenerated,}$ as defined earlier.  The DIC distribution for both
the preindustrial period and after 140 years in the 1pctCO2 simulation reveal the following key
features for each of these carbon pools (Figures 11a,b and 12a,b):
• The saturated carbon pool provides the dominant contribution to the DIC, holding more than
2.15 mol C m$^{-3}$ , particularly within cooler waters below the thermocline;
• The regenerated carbon pool enhances the carbon stored below the surface waters, typically
providing an additional 0.2 mol C m$^{-3}$ within the Southern Ocean and older waters spreading
from the Southern Ocean into the Atlantic and below the thermocline in the Pacific;
• The disequilibrium carbon is small close to the surface, representing waters close to an
equilibrium with the atmosphere.  There  is sometimes a positive disequilibrium of up to 0.05



mol C m$^{-3}$ in some surface waters, which is associated with upwelling transferring carbon-rich
deeper waters to the surface. The disequilibrium carbon is more strongly negative below the
thermocline, typically reaching -0.1 mol C m$^{-3}$ in the Atlantic and        -0.02 mol C m$^{-3}$ in the
Southern Ocean and Pacific. In the preindustrial, the undersaturation in carbon below the
thermocline is due to the subduction of cold waters at high latitudes that have not
equilibrated fully with the atmosphere, which then spread by advection along density
surfaces. In the model integrations reaching year 140,  the carbon below the thermocline
become further undersaturated relative to the contemporary atmosphere due to the rapid
rise in [$CO_2$].

Next we consider the anomalies in the DIC at year 140 in the COU configurations of the 1pctCO2
simulation calculated relative to the preindustrial period.  The carbon anomaly, $\Delta DIC$, in the
COU configuration is positive over the upper thermocline over the Atlantic and Pacific basins,
reaching +0.3 mol C m$^{-3}$, coinciding with regions that are well ventilated. This gain in carbon is
made up of an increase in the saturated carbon over all depths due to higher atmospheric $CO_2$.
There is a dipole in the disequilibrium anomaly (Figures 11b,c and 12 b,c), generally weakly
positive in the upper ocean and more strongly negative in deeper waters below the thermocline
reaching up to -0.2 mol C m$^{-3}$. This negative disequilibrium anomaly in deeper waters is smallest
in the relatively well-ventilated mid-depth waters of the North Atlantic, but extends over nearly
all of the more poorly ventilated mid-depth waters of the Pacific (Figures 11b and 12b).

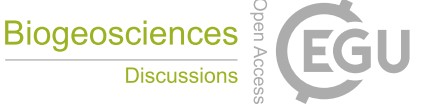

The regenerated carbon anomaly is relatively small in magnitude reaching less than 0.05 mol C
$m^{-3}$ and varies regionally, enhanced within much of the North Atlantic and the thermocline of the
Pacific, but with little change in the deep waters of the Pacific (Figures 11b and 12b). The increase
in regenerated carbon is due to a weakening of ocean overturning leading to an increase in
residence time and an associated accumulation of DIC from the regeneration of biologically-
cycled carbon (Bernardello et al., 2014; Schwinger et al., 2014). The regenerated carbon signal
does not change  in the mid depths and deep Pacific as 140 years is too short an integration
timescale for any effect to be detected.

To diagnose the carbon-cycle feedback parameters, the ocean carbon response needs to be
considered for the BGC configuration where there is no additional warming from the increase in
atmospheric $CO_2$ and limited change in climate and ocean circulation.     The resulting DIC
anomalies are generally very similar to those for the COU configuration (Figures 11b,c and 12b,c),
which is to be expected as the dominant effect for the ocean carbon response is the enhanced
ocean uptake of carbon in response to the increase in [$CO_2$].  There is a weakening in ventilation
in the COU configuration due to the additional radiative forcing. In comparison, in the BGC
configuration, there is no change in the circulation as there is no radiative warming effect, so that
there is slightly more carbon uptake in the northern North Atlantic, such as revealed at around
50°N, compared with the COU configuration. For the BGC configuration, the saturated carbon
pool is slightly greater at depth due to the water masses being cooler than in the COU
configuration, the disequilibrium anomaly shows a less negative anomaly in the northern North





Atlantic because there is little or no change in ventilation, and there are only slight differences in
the regenerated pool.

The climate response to rising [$CO_2$] is now considered in terms of the difference in the COU and
BGC configurations, which includes the combined effects of warming and circulation changes
(Figures 11d and 12d). The surface warming drives a decrease in solubility, an increase in
stratification and a reduction in ventilation, which  leads to an overall decrease in carbon uptake
over the Southern Ocean and Pacific basins, and much of the Atlantic basin.  There is a decrease
in the saturated carbon pool associated with the warming acting to inhibit carbon uptake. The
regenerated carbon anomaly is enhanced in the deep northern North Atlantic and in the
Southern Ocean. The regenerated carbon anomaly for this climate response is very similar to that
for the COU configuration, suggesting that the regenerated carbon anomaly is mainly due to
circulation changes: the gain in regenerated carbon anomaly is consistent with the expected
longer residence time from a weaker overturning and ventilation. There is a more negative
disequilibrium anomaly in the deep waters of the North Atlantic, which is a consequence of
weaker ventilation.

To gain more insight into the disequilibrium response, the ocean DIC response is also considered
for the radiatively-coupled integration (RAD), where there is no increase in [$CO_2$]. The additional
warming leads to a weakening in the overturning, which enhances the residence time in the
surface waters and so generally decreases the magnitude of the disequilibrium anomaly in the



North Atlantic (Figure S8), making the disequilibrium less negative relative to the preindustrial
and so forming a positive disequilibrium anomaly at year 140. In comparison the COU-BGC
captures the effect of the warming under rising [CO2]  leading to the disequilibrium anomaly
instead becoming more negative at depth, since the weakening in the ventilation leads to more
of the anthropogenic carbon remaining at the surface rather than being transferred into the
deeper ocean (Schwinger et al., 2014).

## 4.4.2.2 Changes in ocean carbon pools for diagnosing feedback parameters
The ocean carbon-concentration feedback parameter, $\beta_O$,  is diagnosed from the changes in the
ocean carbon inventories for the BGC configuration, which does not include radiative warming
due to increasing [CO$_2$] (equation 13). There is a consistent increase in ocean carbon storage
across all models with a model mean value of around 670 PgC (Figure 13, light blue bars). This
increase in ocean carbon storage is made up of an increase in the saturated carbon inventory,
$\Delta C_{sat}$, by about 3100 PgC from the increase in [CO$_2$] (Figure 13, red bars). This increase is partly
offset by a more negative disequilibrium carbon, $\Delta C_{disequilib}$, of typically -2500 PgC (Figure 13,
dark blue bars), representing how the ocean carbon uptake cannot keep up with the rate of [CO$_2$]
increase. There is relatively little change in the regenerated carbon inventory, $\Delta C_{regenerated}$. The
resulting $\beta_O$ is positive and mainly explained by the chemical response involving the rise in ocean
saturation with no significant biological changes, although the physical uptake of carbon within
the ocean is unable to keep pace with the rise in [CO$_2$].



The ocean carbon-climate feedback parameter, $\gamma_O$, is diagnosed from the difference between
the COU model configuration and the BGC configuration, and so includes the effect of an
increasing surface warming under rising [$CO_2$] (equation 14). There is a broadly consistent
response across models, with a model mean decrease in carbon inventory of around 80 PgC due
to the additional warming in the COU configuration relative to the BGC configuration (Figure 14,
light blue bars). The effect of this additional warming and the associated climate change leads to
a decrease in both the saturated carbon and disequilibrium carbon of typically -60 and -70 PgC
(Figure 14, orange and dark blue bars), representing the decrease in solubility and decreased
ocean ventilation. There is an increase in the regenerated carbon of typically 50 PgC (Figure 14,
green bars), which is due to a weaker circulation leading to a longer residence time of
thermocline and deep waters, so that there is more time for the accumulation of regenerated
carbon below the mixed layer. The resulting $\gamma_O$ is negative, indicating that the ocean takes up
less carbon in response to the combination of surface warming and a weakening in ocean
ventilation. This response involves a combination of chemical, physical and biological changes
where the warming reduces the solubility of the carbon in the ocean and a weakening in the
circulation decreases the disequilibrium pool, but lengthens the residence time and so increases
the regenerated pool.

Overall, the ocean carbon inventory increases in the BGC configuration by 666±53 PgC (model
mean ± ensemble standard deviation), and decreases in COU relative to BGC by -80±15 PgC. The
resulting $\beta_O$ is very similar across all the models (0.78±0.06 PgC ppm$^{-1}$), reflecting the strong
control of carbonate chemistry by rising atmospheric $CO_2$ (Katavouta et al., 2018). The dominant



contributions are composed of a positive contribution from the saturated carbon (3.66±0.16 PgC
ppm$^{-1}$) and a negative contribution from the disequilibrium carbon (-2.98±0.16 PgC ppm$^{-1}$ ) (see
Table A3 in the Appendix); these inter-model differences are relatively small with ratios of the
standard deviation to model mean of only 0.05 and 0.06 respectively. The regenerated
contribution is over two orders of magnitude smaller than the sum of the saturated and
disequilibrium contributions, and so may be neglected for evaluating $\beta_O$.

The values of $\gamma_O$ differ more strongly across the models (-16.95±5.62 PgC °C$^{-1}$) and  arise from
differences in the extent of the surface warming and the dynamical changes in the ocean
circulation and resulting changes in ventilation, residence time and biological regeneration (Table
A3).  The contributions to $\gamma_O$ include negative contributions from the saturated  (-12.78±2.50 PgC
°C$^{-1}$) and disequilibrium (-16.36±5.31 PgC °C$^{-1}$) components, which are partly opposed by a
positive contribution from the regenerated component (12.25±8.53 PgC °C$^{-1}$). The largest
intermodel differences are in the regenerated and disequilibrium responses and a relatively small
spread in the saturated response, with the ratios of the standard deviation to the model mean
are 0.70, 0.33 and 0.20 respectively (Table A3).

4.5. Transient climate response (TCR) and transient climate response to cumulative
emissions (TCRE)



Other than the feedbacks associated with the coupled carbon cycle and climate system, the
idealized 1pctCO2 simulation is also used for calculating two other climate metrics routinely. The
first is the transient climate response (TCR) which is defined as the temperature change, relative
to the preindustrial state, at the time of $CO_2$ doubling ($\Delta T_{2 \times CO2}$), that occurs at 70 years after the
start of the simulation. The second is the transient climate response to cumulative emissions
(TCRE) which is defined as ratio of TCR to diagnosed cumulative fossil fuel emissions also at the
time of $CO_2$ doubling ($\tilde{E}_{2 \times CO2}$) (Matthews et al., 2009) typically expressed in units of °C/EgC (1
EgC = 1000 PgC).

$$TCRE = \frac{\Delta T_{2 \times CO2}}{\tilde{E}_{2 \times CO2}} \tag{21}$$

It has been shown that TCRE is approximately constant over a wide range of cumulative emissions
and emission pathways (e.g. see review by MacDougall, 2016). Therefore, although non-$CO_2$
GHGs and other climate forcings (e.g. aerosols and land use change) also affect the realized
warming, TCRE is a considered to be a straightforward measure of peak warming caused by
anthropogenic $CO_2$ emissions.

We do not discuss here TCR and TCRE in detail since the focus of our study is on carbon feedbacks.
However, both these quantities are readily calculated using results presented in this study. Table
A4 in the appendix lists TCR, $\tilde{E}_{2 \times CO2}$, and TCRE from the eleven CMIP6 models considered in this
study. The mean ± standard deviation range for TCR, $\tilde{E}_{2 \times CO2}$, and TCRE from the eleven CMIP6
models considered here are 1.99 ± 0.44 °C, 1121 ± 73 PgC, and 1.78 ± 0.41 °C EgC$^{-1}$, respectively.



For fifteen CMIP5 models, Gillett et al. (2013) calculated the mean ± standard deviation range for
TCRE to be 1.63 ± 0.48 °C EgC$^{-1}$ and a 5%-95% range for its observationally constrained value as
0.7-2.0 °C EgC$^{-1}$. The TCRE metric has gained significant policy relevance (Frame et al., 2014;
Millar et al., 2016) and it is used to calculate the remaining allowable carbon emissions to reach
a specified temperature change target above the preindustrial level (Millar et al., 2017; Rogelj et
al., 2019).

The uncertainties in TCRE stem from uncertainties both in TCR and $\tilde{E}_{2\times CO2}$ which is directly
affected by land and ocean carbon uptake. A large fraction of uncertainty in $\tilde{E}_{2\times CO2}$ comes from
the diverse response of land carbon cycle models and the results presented here indicate that
representation of the nitrogen cycle is helpful in reducing this uncertainty, as indicated by the
spread across land models. For the results reported here from eleven CMIP6 models, however,
the uncertainty in TCR (mean ± standard deviation = 1.99 ± 0.44 °C) is much greater than the
uncertainty in $\tilde{E}_{2\times CO2}$ (1121 ± 73 PgC) so that TCR contributes about 90% of the total uncertainty
in the calculated TCRE value (1.78 ± 0.41 °C EgC$^{-1}$) (see section A6 in the Appendix).

The TCRE may also be expressed in terms of a product of a thermal contribution from the
dependence of surface warming on radiative forcing  and a carbon contribution from the
dependence of radiative forcing on cumulative carbon emissions (Williams et al., 2016; Katavouta
et al., 2018), as





$$TCRE = \frac{\Delta T_{2\times CO2}}{\Delta R_{2\times CO2}} \frac{\Delta R_{2\times CO2}}{\tilde{E}_{2\times CO2}} \tag{22}$$

where $\Delta R_{2\times CO2}$ is the change in radiative forcing relative to the preindustrial period. For a suite of ten CMIP5 models, Williams et al. (2017) show that the inter-model spread in the TCRE calculated from the 1pctCO2 experiment, is again dominated by the inter-model differences in the thermal contribution, $\frac{\Delta T_{2\times CO2}}{\Delta R_{2\times CO2}}$, due to climate feedback and ocean heat uptake over the first few decades, but the inter-model differences in the carbon contribution, $\frac{\Delta R_{2\times CO2}}{\tilde{E}_{2\times CO2}}$, due to land and ocean carbon uptake become of comparable importance after 80 years.

Although a large fraction of uncertainty in TCRE is contributed by physical climate system processes that determine TCR and not the biogeochemical processes that determine $\tilde{E}_{2\times CO2}$, reducing the uncertainty in land and ocean carbon uptake across models will still contribute to reducing the uncertainty in the estimates of TCRE on centennial timescales.

## 5. Summary and conclusions

Model intercomparison projects offer several benefits including calculation of model mean response, quantification of the uncertainty based on the spread across models, and how this uncertainty changes over time that allows modellers to evaluate how their model's response is different from others'. The carbon feedbacks analysis presented here based on the C⁴MIP





protocol of experiments (Jones et al., 2016) allows to investigate how feedback strengths have
evolved since CMIP5 and also attempts to understand the reasons behind the spread in models.

The carbon uptake over land and ocean, in response to increasing atmospheric $CO_2$
concentration, is well known to be dominated by the positive contribution from the carbon-
concentration feedback (Arora et al., 2013a; Gregory et al., 2009). The strength of this feedback
is of comparable magnitudes over land (mean ± standard deviation = 0.97±0.40 PgC ppm$^{-1}$) and
ocean (0.79±0.07 PgC ppm$^{-1}$) although the feedback is much more uncertain over land as
indicated by the standard deviation across the eleven models considered here. This dominant
positive contribution from the carbon-concentration feedback is, however, opposed by the
weaker negative carbon-climate feedback that is associated with the climate change that results
due to increasing atmospheric $CO_2$. The absolute magnitude of this weaker negative feedback is
about three times larger, but an order of magnitude more uncertain, over land (-45.1±50.6 PgC
$°C^{-1}$) than over ocean (-17.2±5.0 PgC $°C^{-1}$). Model estimates of the ocean carbon-concentration
feedback are very consistent with each other, reflecting the strong control of how carbonate
chemistry alters with rising atmospheric $CO_2$. There is a relatively wider range in the model
estimates of the ocean carbon-climate feedback, particularly in terms of how changes in ocean
circulation alter the disequilibrium and regeneration terms. Over land, however, since the
carbon-concentration and carbon-climate feedbacks are determined entirely by biological
process, which are much less understood, the resulting uncertainty is much higher across land
models than across the ocean models. This uncertainty in the strength of carbon-concentration
and carbon-climate feedbacks over land is well known (Arora et al., 2013b; Friedlingstein et al.,



2006). The inclusion of N cycle results in lower absolute strength of the feedback parameters
over land but also a reduced spread across the land models. While the uncertainty in TCRE is
dominated by physical processes affecting the thermal response involving climate feedbacks and
heat uptake on decadal timescales, a reduction in the uncertainty in land and ocean carbon
uptake across models will reduce the uncertainty in the TCRE on centennial timescales.

The additional analyses that we have performed to gain further insight into the reasons for
differences among models provide insight into their diverse response, especially for land models.
Over land, the diverse response of models is found to be primarily due to the wide range of the
strength of the $CO_2$ fertilization effect, the fraction of GPP that is converted to NPP, and the
residence times of carbon in the live (vegetation) and dead (litter plus soil) carbon pools across
models. There is more consistency in the response of the ocean models, although inter-model
differences arise from differences in the ventilation and residence time altering the ocean
disequilibrium and regenerated carbon.

Finally, the decision to use fully- and biogeochemically coupled configurations of the 1pctCO2
experiment as the standard simulations to diagnose carbon cycle and climate system feedbacks
from should provide consistency and continuity for future versions of Earth system models to be
compared against their predecessors.





**Table 1**: The values of the carbon-concentration ($\beta$) and carbon-climate ($\gamma$) feedback parameters
can be solved using results from any two combinations of the RAD, BGC and COU versions of an
experiment as shown in equation (1). In addition, when using results from the BGC and COU
simulations the effect of temperature change in the BGC simulation ($T^*$) can be neglected, as was
done in the F06 study, yielding approximate values for $\beta_X$ and $\gamma_X$.

| Approach | $\gamma_X$ | $\beta_X$ |
|---|---|---|
| The RAD-BGC approach | $\gamma_X = \frac{\Delta C_X^+}{T^+}$ | $\beta_X = \frac{\Delta C_X^*}{c'} - \frac{\gamma_X T^*}{c'}$ |
| The RAD-COU approach | $\gamma_X = \frac{\Delta C_X^+}{T^+}$ | $\beta_X = \frac{\Delta C_X'}{c'} - \frac{\gamma_X T'}{c'}$ |
| The BGC-COU approach | $\gamma_X = \frac{\Delta C_X' - \Delta C_X^*}{T' - T^*}$ | $\beta_X = \frac{1}{c'}\left(\frac{\Delta C_X^* T' - \Delta C_X' T^*}{T' - T^*}\right)$ |
| The BGC-COU approach with $T^* = 0$ | $\gamma_X = \frac{\Delta C_X' - \Delta C_X^*}{T'}$ | $\beta_X = \frac{\Delta C_X^*}{c'}$ |










| Modelling group | CSIRO | BCC | CCCma | CESM | CNRM | GFDL |
|---|---|---|---|---|---|---|
| ESM | ACCESS-ESM1.5 | BCC-CSM2-MR | CanESM5 | CESM2 | CNRM-ESM2-1 | GFDL-ESM4 |
| Atmosphere resolution | 1.875°x1.25°, L38 | 1.125°x1.125°, L46 | 2.81° ×2.81°, L49 | 0.9°x1.25° | T127 (1.4°x1.4°) L91 | Cube-sphere C96 (1-degree) |
| Ocean resolution | 1° but finer between 10S-10N and in the Southern Ocean, L50 | 1° but becoming finer to 1/3° within 30°N - 30°S, L40 | 1° but becoming finer to 1/3° within 20°N - 20°S, L45. | gx1v7 displaced pole grid (384 x 320 lat x lon) | 1°but becoming 0.3° in the Tropics, L75 | 0.5 degree tri-polar grid |
| Land carbon/biogeochemistry component | | | | | | |
| Model name | CABLE2.4 with CASA-CNP | BCC-AVIM2 | CLASS-CTEM | CLM5 | ISBA-CTRIP | LM4p1 |
| Number of live carbon pools | 3 | 3 | 3 | 22 | 6 | 6 |
| Number of dead carbon pools | 6 | 8 | 2 | 7 | 7 | 4 |
| Number of plant functional types (PFTs) | 13 | 16 | 9 | 22 | 16 | 6 |
| Fire | No | No | No | Yes | yes | Yes |
| Dynamic vegetation cover | No | No | No | No | no | Yes |
| Nitrogen cycle | Yes (and phosphorus) | No | No | Yes | No (implicit, derived from Yin 2002) | No |
| Ocean carbon/biogeochemistry component | | | | | | |
| Model name | WOMBAT | MOM4_L40, Ocean carbon cycle follows OCMIP2 | CMOC (biology), carbonate chemistry follows OMIP protocol. | MARBL | PISCESv2-gas | COBALTv2 |
| Number of phytoplankton types | 1 | 0 | 1 | 3 | 2 | 2 |
| Number of zooplankton types | 1 | 0 | 1 | 1 | 2 | 3 |
| Explicit nutrients considered | Phosphorus, Iron | Phosphorus | Nitrogen | Nitrogen, Phosphorus, Silica, Iron | Nitrogen, Phosphorus, Silica, Iron | Nitrogen, Phosphorus, Silica, Iron |










| Modelling group | IPSL | JAMSETC (Team MIROC) | MPI | NCC | UK |
|---|---|---|---|---|---|
| ESM | IPSL-CM6A-LR | MIROC-ES2L | MPI-ESM1.2-LR | NorESM2-LM | UKESM1-0-LL |
| Atmosphere resolution | 2.5°x.3°, L79 | 2.81x2.81, L40 | T63, 1.8°x1.8°. L47 | 1.9°x2.5°, L32 | 1.875° x1.25°, L85 |
| Ocean resolution | 1°-0.3° in the Tropics L75 | Almost 1° but becoming finer to North pole and equator (Tripolar system: 360x256), L62 | GR1.5 (1.5°, finer close to Antarctica and Greenland), L40 | 1° with enhanced meridional resolution near the Equator, L53 | 1° |
| Land carbon/biogeochemistry component | | | | | |
| Model name | ORCHIDEE, branch 2.0 | MATSIRO (physics) VISIT-e (BGC) | JSBACH3.2 | CLM5 | JULES-ES-1.0 |
| Number of live carbon pools | 8 | 3 | 3 | 22 | 3 |
| Number of dead carbon pools | 3 | 6 | 18 | 7 | 4 |
| Number of plant functional types (PFTs) | 15 | 13 | 13 | 22 | 13 |
| Fire | No | No | Yes | Yes | No |
| Dynamic vegetation cover | No | No | Yes | No | Yes |
| Nitrogen cycle | No | Yes | Yes | Yes | Yes |
| Ocean carbon/biogeochemistry component | | | | | |
| Model name | PISCES-v2 | OECO2 | HAMOCC6 | Modified HAMOCC5.1 | MEDUSA-2.1 |
| Number of phytoplankton types | 2 | 2 (non-diazotroph and diazotroph) | 2 | 1 | 2 |
| Number of zooplankton types | 2 | 1 | 1 | 1 | 2 |
| Explicit nutrients considered | Nitrogen, Phosphorus, Silica, Iron | Nitrogen, Phosphorus, Iron | Nitrogen, Phosphorus, Silica, Iron | Nitrogen, Phosphorus, Silica, Iron | Nitrogen, Silica, Iron |

**Table 2**: Primary features of the physical atmosphere and ocean components, and land and ocean carbon cycle components of the eleven participating models in this study.






**Table 3**: Correlation between carbon-concentration ($\beta_X$) and carbon-climate ($\gamma_X$) feedback
parameters over land and ocean across comprehensive ESMs from the CMIP5 intercomparison
in the A13 study and CMIP6 intercomparison in this study. For land correlation is also shown
when CanESM5 is excluded from CMIP6 models.

| Land | Ocean | |
|------|-------|---|
| −0.69 <br> −0.92 (excluding CanESM5) | −0.64 | **CMIP6** (11 models) |
| −0.82 | −0.75 | **CMIP5** (8 models) |










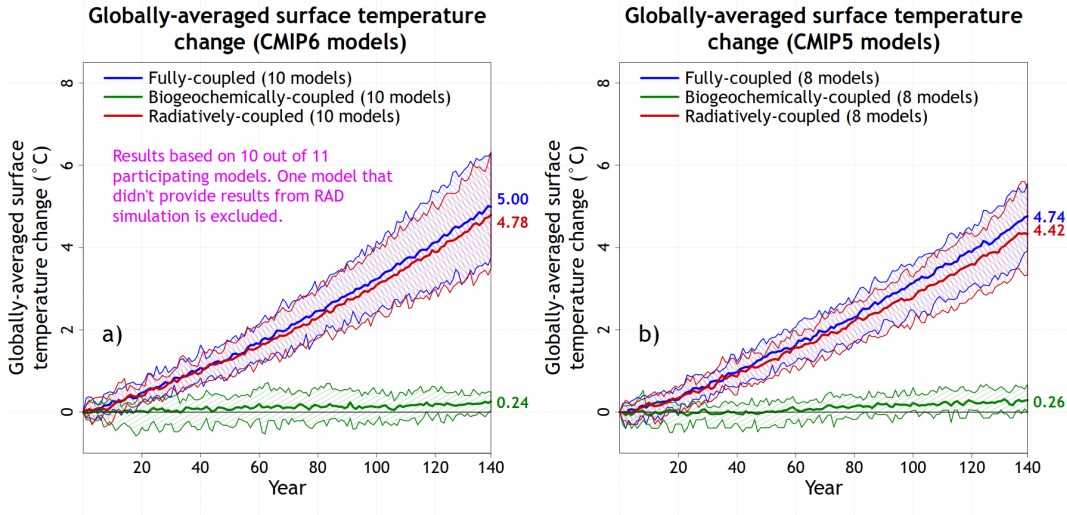


Figure 1: Temperature changes in the fully-, biogeochemically- and radiatively-coupled configurations of the 1pctCO2 experiment across participating CMIP6 (panel a) and CMIP5 (panel b) comprehensive ESMs that participated in this and the Arora et al. (2013) study, respectively. Model mean is indicated by the solid lines and the range across the models is indicated by shading around the solid lines. Individual model results are shown in Figure 4.

1134

1135

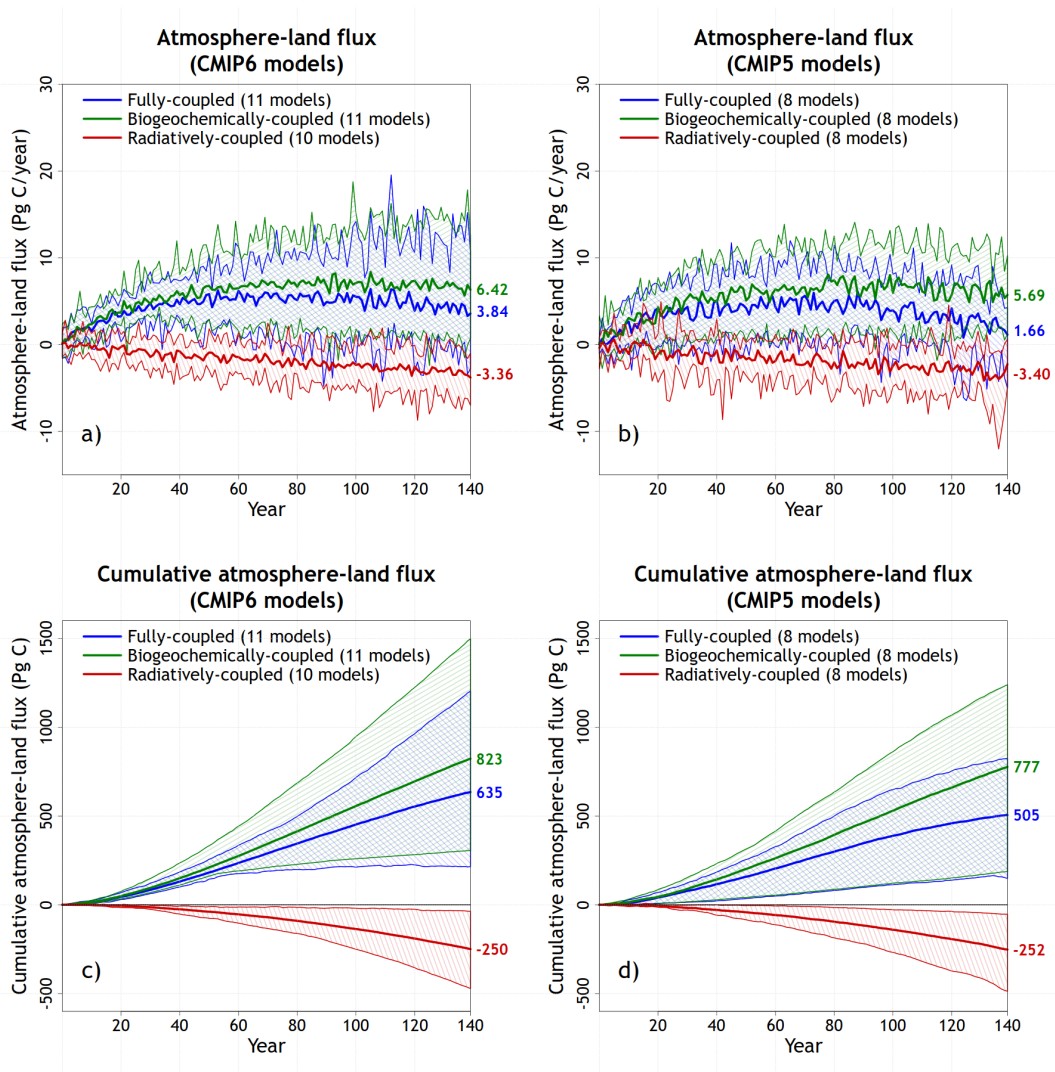

Figure 2: Model mean values and the range across models for annual simulated atmosphere-land $CO_2$ flux (top row) and their cumulative values (bottom row) for participating CMIP6 (left column) and CMIP5 (right column) models from the fully-, biogeochemically- and radiatively-coupled versions of the 1pctCO2 experiment. Individual model results are shown in Figure 4.



1142

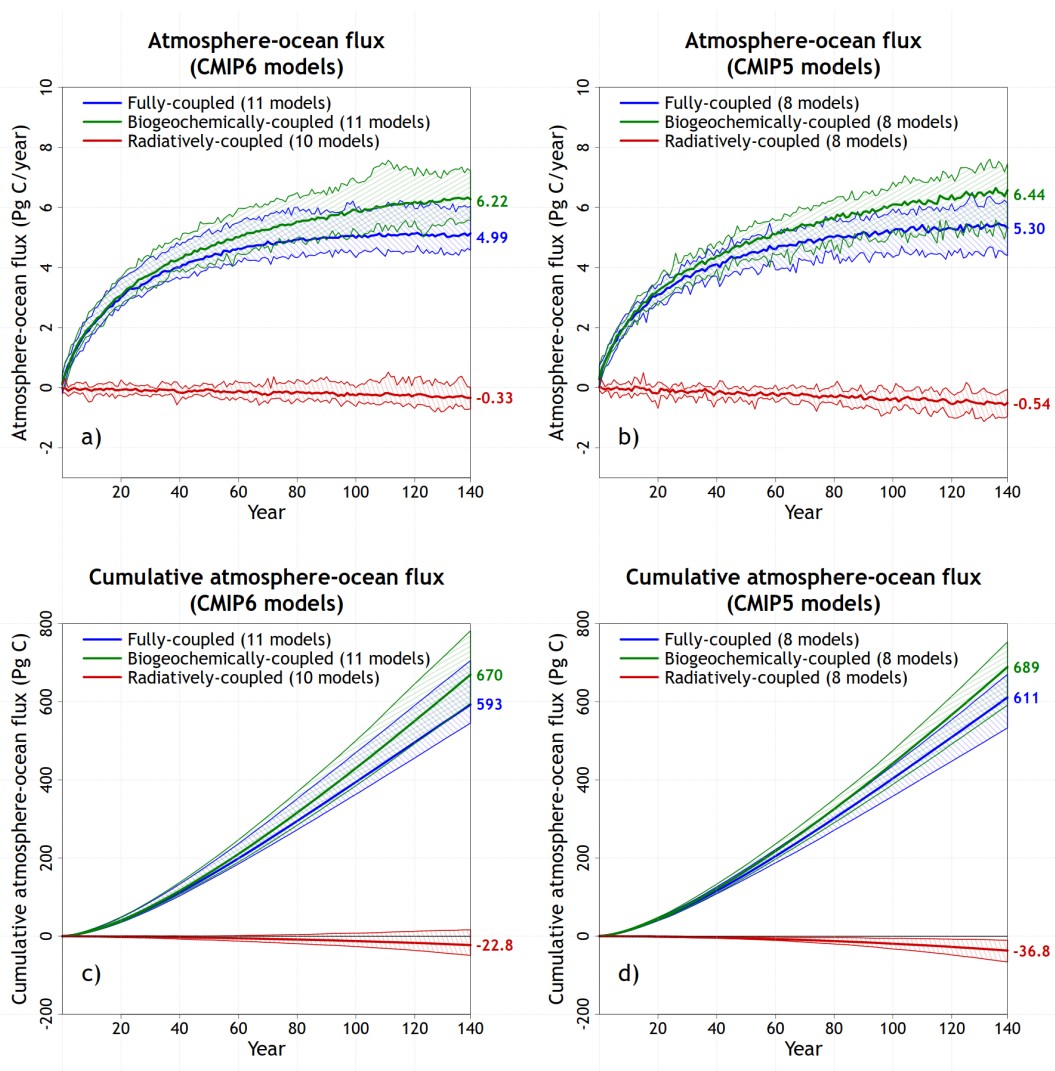

1143

Figure 3: Model mean values and the range across models for annual simulated atmosphere-ocean $CO_2$ flux (top row) and their cumulative values (bottom row) for participating CMIP6 (left column) and CMIP5 (right column) models from the fully-, biogeochemically- and radiatively-coupled versions of the 1pctCO2 experiment. Individual model results are shown in Figure 4.

1148




Figure 4: Individual model values from CMIP6 models for globally-averaged surface temperature change (top row), cumulative atmosphere-land CO$_2$ flux (middle row), and cumulative atmosphere-ocean CO$_2$ flux (bottom row) from the fully-, biogeochemically- and radiatively-coupled versions of the 1pctCO2 experiment. Results from the radiatively-coupled configuration were not available from NorESM2-LM models at the time of writing.


1154

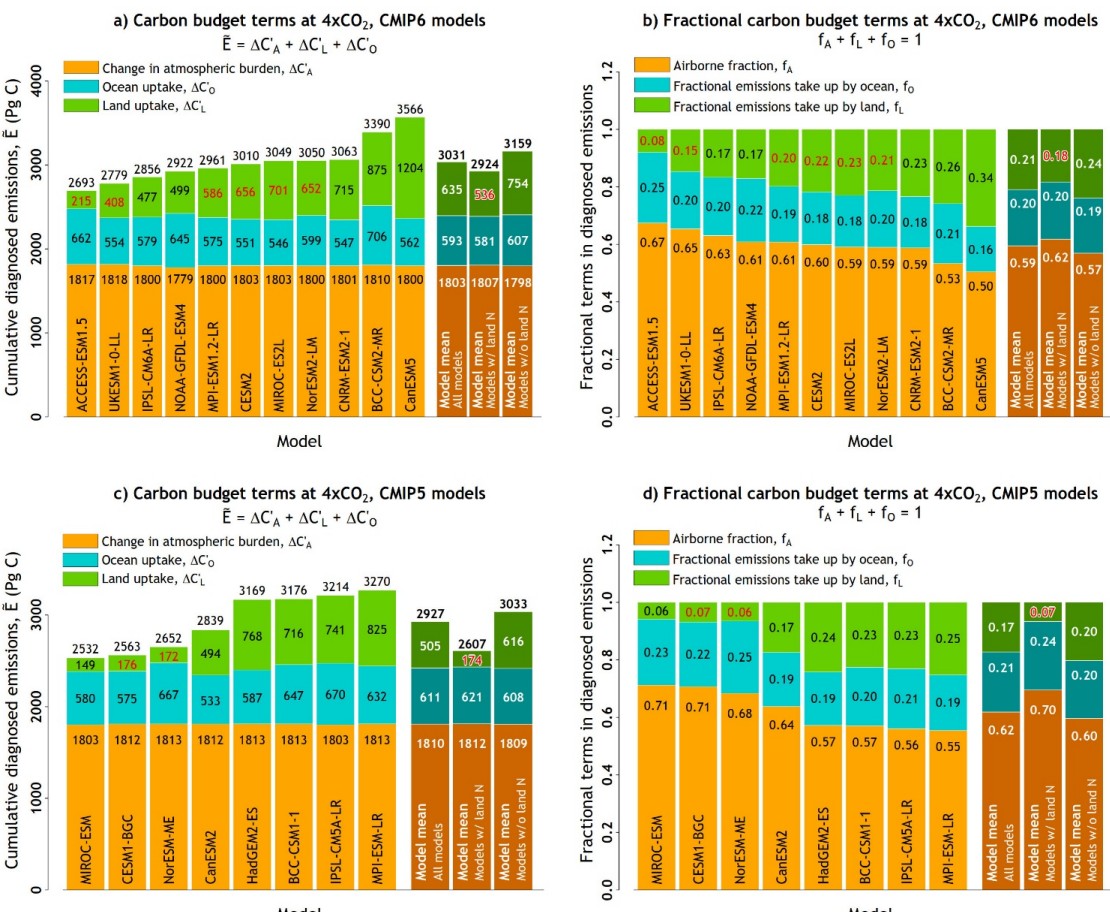

Figure 5: Components of the carbon budget terms in cumulative emissions from the eleven participating CMIP6 models based on equation (15) in panel (a) and equation (16) in panel (b) using results from the fully-coupled 1pctCO2 simulation. The models are arranged in an ascending order based on their cumulative emissions values. Results from participating CMIP5 models in the A13 study are shown in panels c and d. In addition, ESMs whose land component includes a representation of N cycle are identified by red font colour for cumulative land carbon uptake (panels a and c) and fractional emissions taken up by land (panels b and d). Model mean is shown for all models but also separately for models whose land components include or do not include a representation of the N cycle.

1164

1165

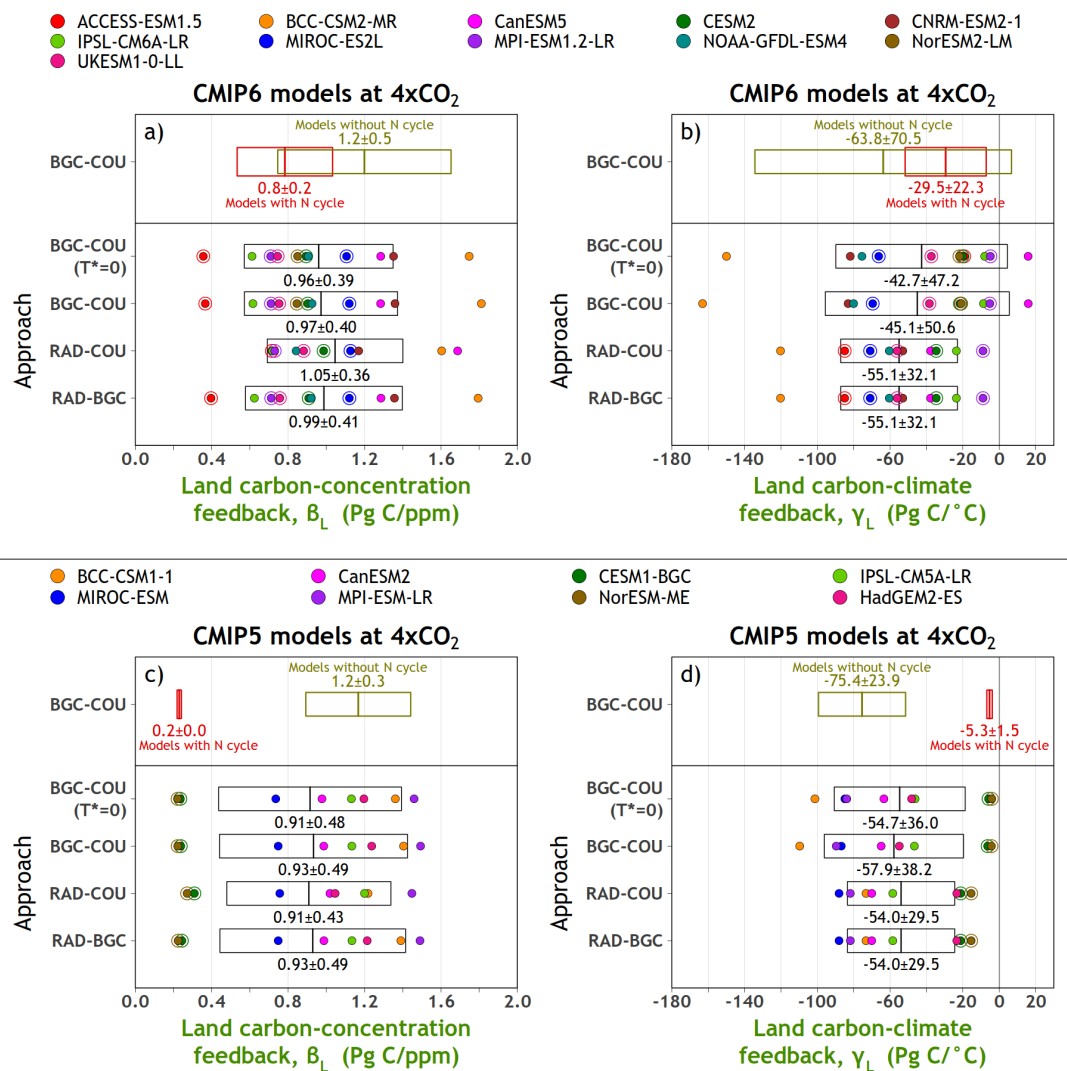

1166

Figure 6: Carbon-concentration (panel a) and carbon-climate (panel b) feedback parameters
over land from participating CMIP6 models calculated using the approaches summarized in
Table 1. The boxes show the mean ± 1 standard deviation range and the individual coloured
dots represent individual models. Models which include a representation of land nitrogen cycle
are identified with a circle around their dot. Model-mean ± 1 standard deviation range of
feedback parameters is also separately shown for models which do and do not represent land
nitrogen cycle using the BGC-COU approach. Results from participating CMIP5 models in the
A13 study are shown in panels c and d. Note that among CMIP6 models results from NorESM2-
LM were not available for the RAD simulation at the time of writing.


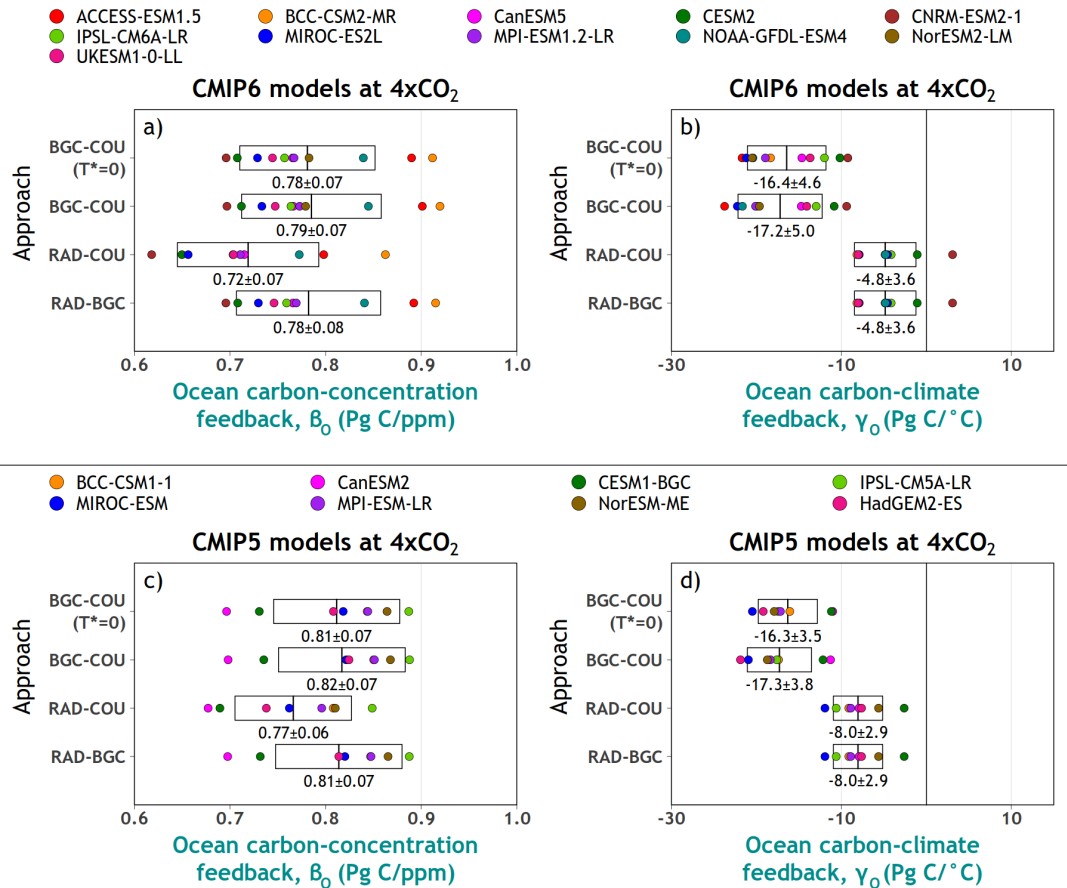


Figure 7: Carbon-concentration (panel a) and carbon-climate (panel b) feedback parameters
over ocean from participating CMIP6 models calculated using the approaches summarized in
Table 1. The boxes show the mean ± 1 standard deviation range. Results from participating
CMIP5 models in the A13 study are shown in panels c and d. Note that among CMIP6 models
results from NorESM2-LM were not available for the RAD simulation at the time of writing.


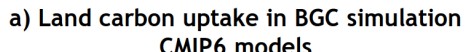

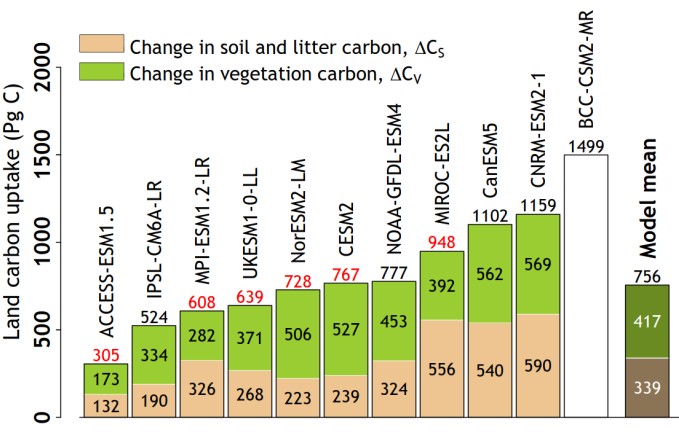

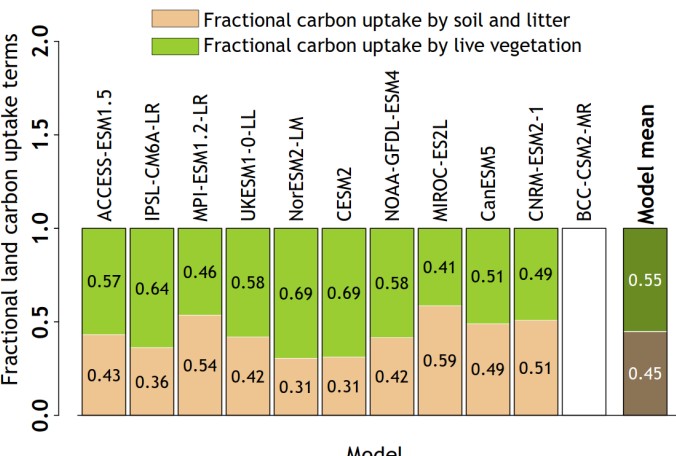


Figure 8: Carbon uptake over land in the BGC simulation, used to calculate land carbon-concentration feedback ($\beta_L$) and its partitioning into vegetation and soil+litter carbon pools across the participating CMIP6 models (panel a). Panel (b) shows the fractional land carbon uptake by vegetation and soil+litter carbon pools in the BGC simulation. No partitioning is shown for the BCC-CSM2-MR model because total land carbon uptake in this model exceeded the sum of changes in the vegetation and soil+litter carbon pools by more than 10%. Total land carbon uptake in models which include a representation of the N cycle is shown in red color. The results from the BCC-CSM2-MR model are not used in calculating the model-mean values.





1192

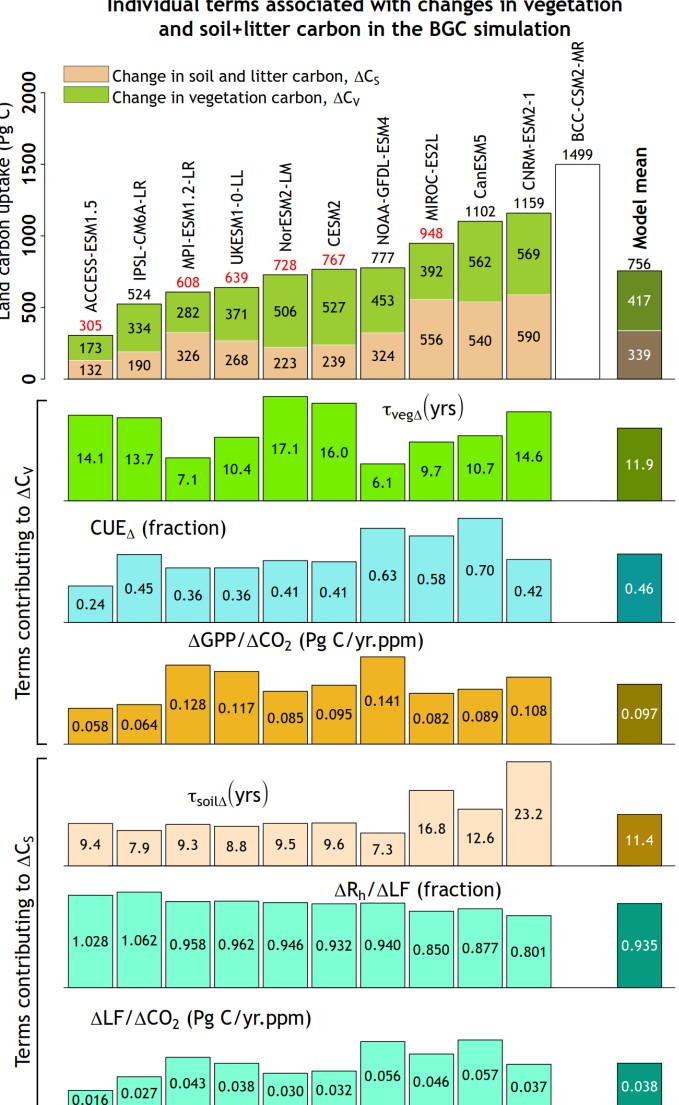

Figure 9: Individual terms of equation (8) which contribute to changes in vegetation ($\Delta C_V$) and litter+soil ($\Delta C_S$) carbon pools. Values from the BCC-CSM2-MR model are not used in calculating the model-mean.










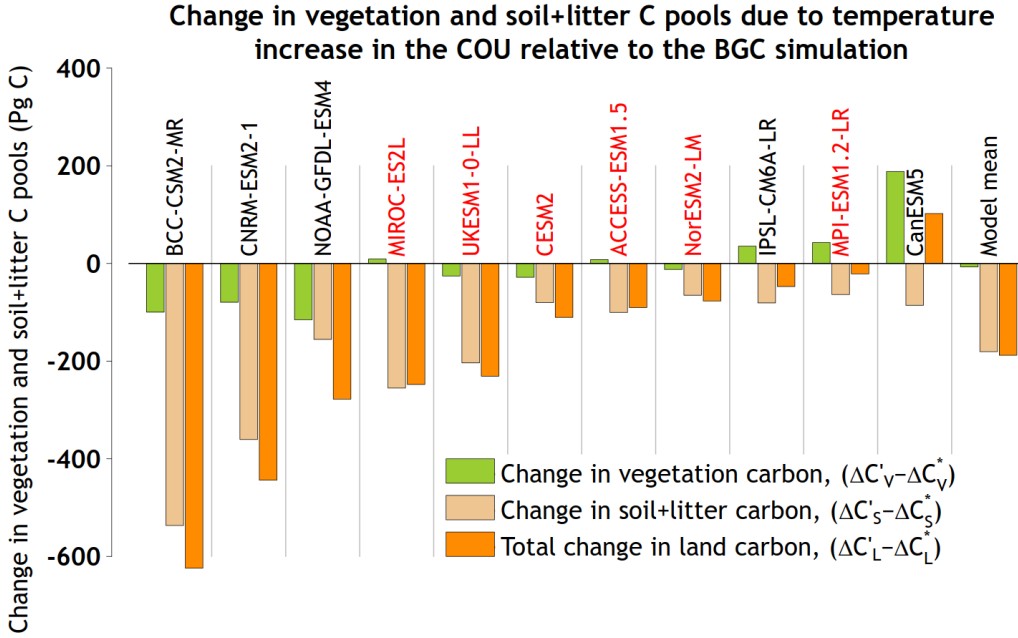

 Figure 10: The changes in vegetation and soil+litter carbon pools in the COU relative to the BGC
 simulation, as shown in equation (9), which contribute to the calculation of carbon-climate
 feedback over land ($\gamma_L$) in the BGC-COU approach.





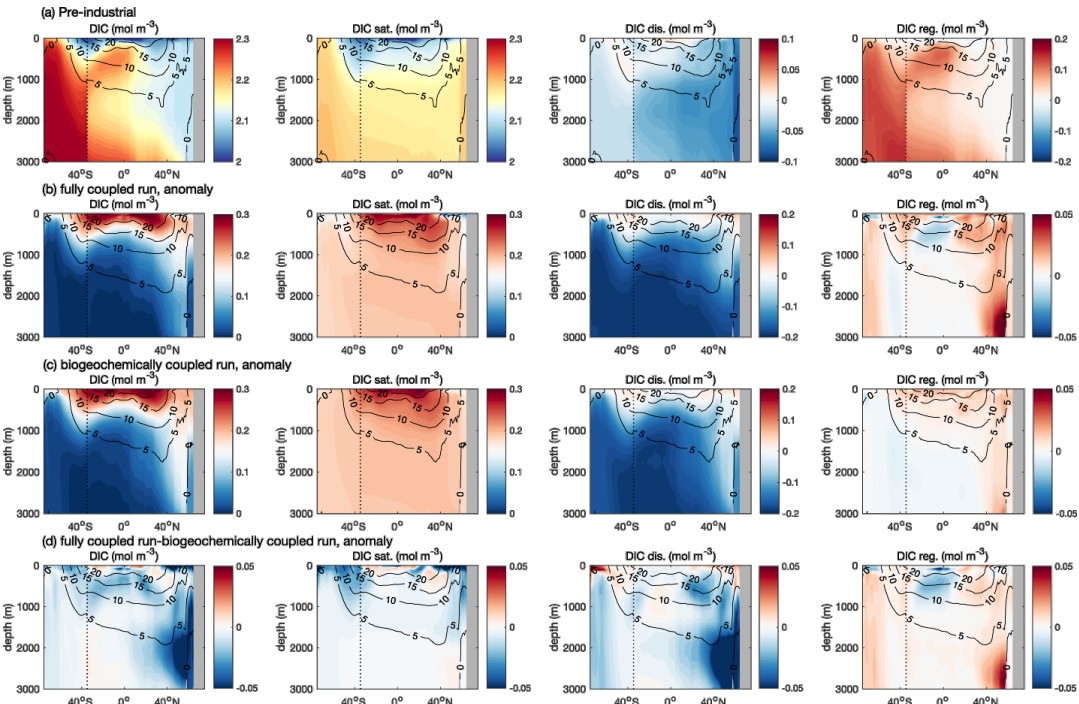

1229

1230

Figure 11. Meridional section of the dissolved inorganic carbon, $DIC$ (mol m$^{-3}$), and constituent carbon pools in UK-ESM1-0-LL for the zonally-averaged Atlantic and Southern Ocean: (a) the preindustrial absolute concentrations, and the anomalies relative to the preindustrial state at year 140 for (b) the COU configuration, (c) the BGC configuration and (d) the COU minus the BGC configuration. The $DIC$ is separated into saturated carbon, $DIC_{sat}$, the disequilibrium carbon, $DIC_{disequilib}$, and the regenerated carbon, $DIC_{regenerated}$. The Atlantic and Southern Ocean domains are separated by a black vertical line.

1238



1239

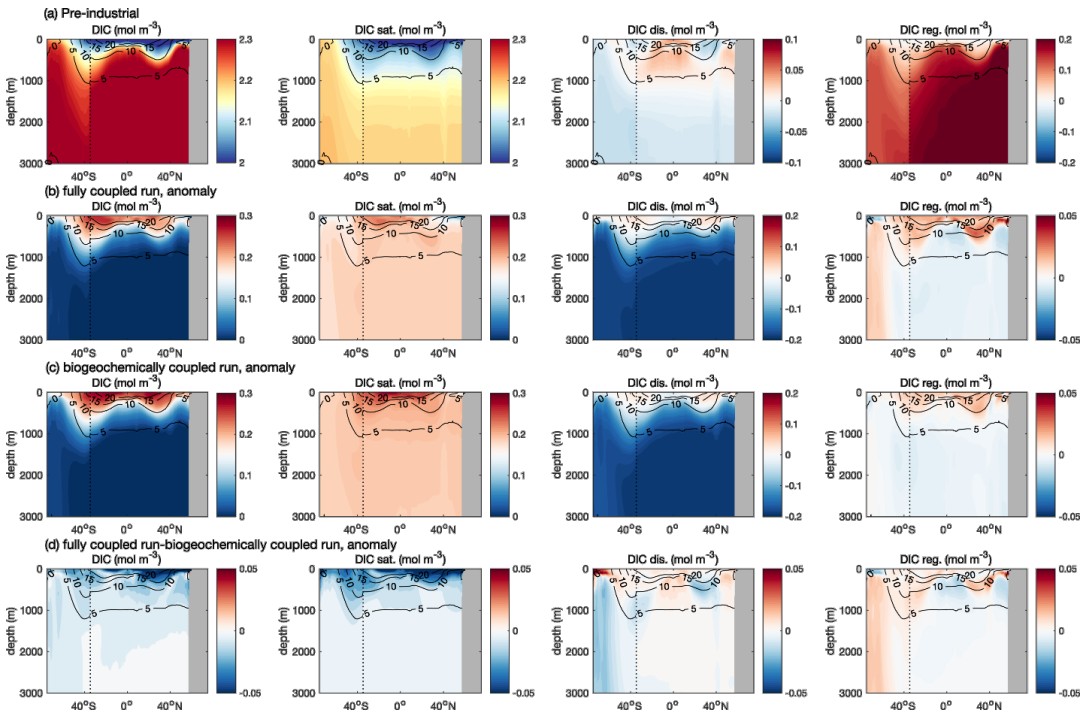

1240

Figure 12. Meridional section of the dissolved inorganic carbon, *DIC* (mol m$^{-3}$), and constituent

carbon pools in UK-ESM1-0-LL for the zonally-averaged Pacific and Southern Ocean: (a) the

preindustrial absolute concentrations, and the anomalies relative to the preindustrial state at

year 140 for (b) the COU configuration, (c) the BGC configuration and (d) the COU minus the

BGC configuration. The *DIC* is separated into saturated carbon, $DIC_{sat}$, the disequilibrium

carbon, $DIC_{disequilib}$, and the regenerated carbon, $DIC_{regenerated}$. The Pacific and Southern Ocean

domains are separated by a black vertical line.





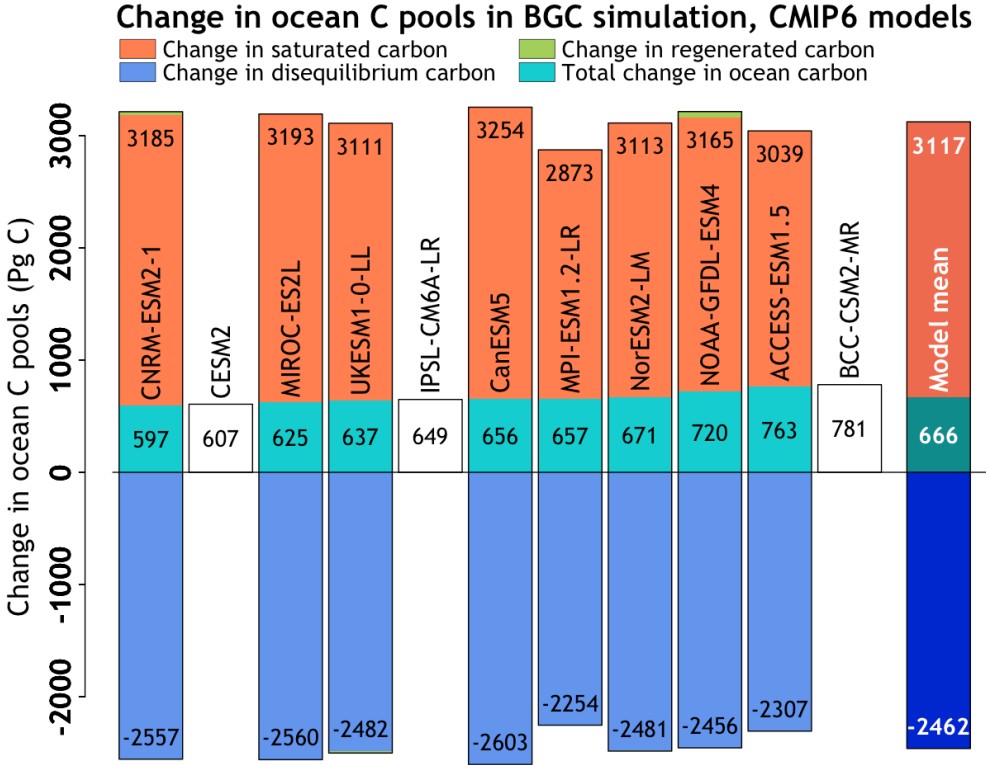


Figure 13. Carbon uptake over the ocean in the biogeochemically-coupled simulation, used to
calculate ocean carbon-concentration feedback and its partitioning into saturated, disequilibrium
and regenerated carbon pools across the participating CMIP6 models (left panels) using equation
(12). No partitioning is shown for models for which 3D ocean fields were not available and the
results of these models are not used in calculating the model mean values (right panel).  The sum
of the partitions does not exactly match the total ocean uptake diagnosed from the air-sea fluxes
due to land-ocean interactions involving storage in sediments and river inputs.



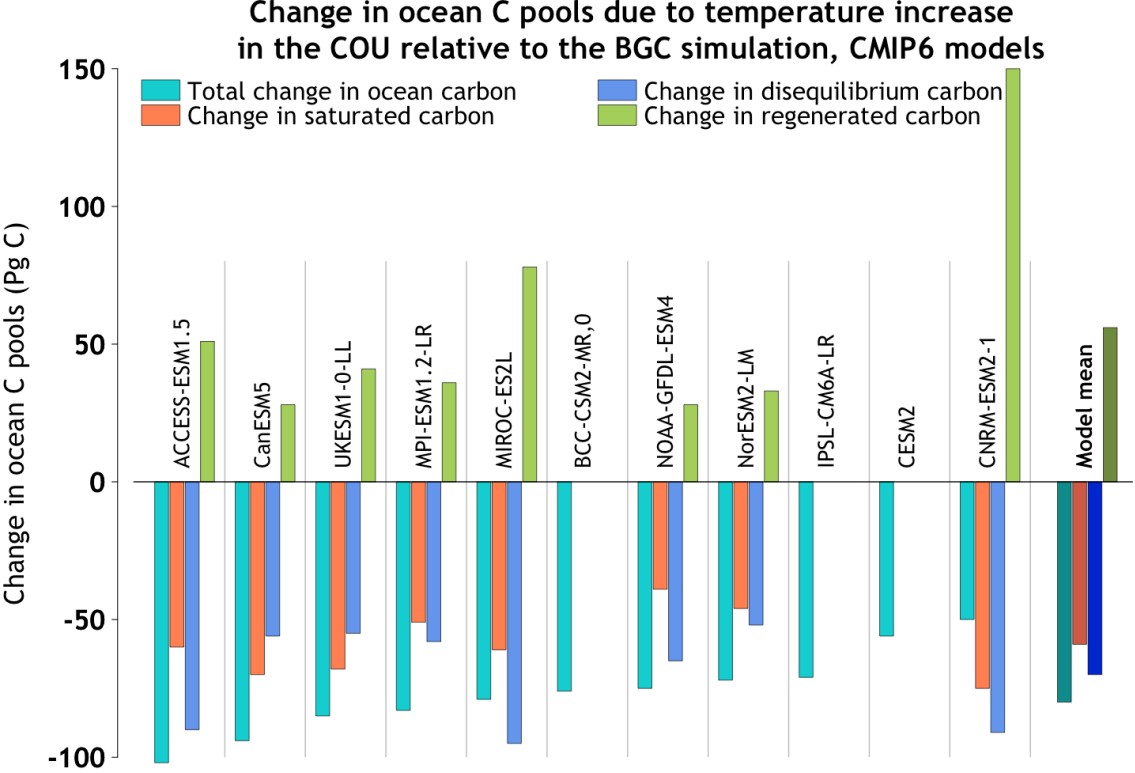


Figure 14.  Change in saturated, disequilibrium and regenerated carbon pools in the fully coupled
minus the biogeochemical simulation using equation (12), which contribute to the calculation of
carbon-concentration feedback over the ocean. The sum of the partitions does not exactly match
the total ocean uptake diagnosed from the air-sea fluxes due to land-ocean interactions involving
storage in sediments and river inputs.




**Appendix**

**A1. The climate carbon cycle feedbacks framework**

The rate of change of carbon in the combined atmosphere-land-ocean system is written as

$$\frac{dC_G}{dt} = \frac{dC_A}{dt} + \frac{dC_L}{dt} + \frac{dC_O}{dt} = E \qquad \text{(A1)}$$


where the *G*lobal carbon pool $C_G = C_A + C_L + C_O$ is the sum of carbon in the *A*tmosphere, *L*and
and *O*cean components (PgC), and *E* is the rate of anthropogenic $CO_2$ emission (PgC/yr) into the
atmosphere. The equations for the atmosphere, land and ocean are

$$
\begin{aligned}
\frac{dC_A}{dt} &= F_A(T,c) + E \\
\frac{dC_L}{dt} &= F_L(T,c) \\
\frac{dC_O}{dt} &= F_O(T,c)
\end{aligned}
\qquad \text{(A2)}
$$


where $(F_L + F_O) = -F_A$ are the fluxes between the atmosphere and the underlying land and
ocean, taken to be positive into the components. The fluxes *F* are expressed as functions of
surface temperature T and the surface atmospheric $CO_2$ concentration *c*. Here and subsequently,
uppercase *C* denotes carbon pools and lowercase *c* denotes atmospheric $CO_2$ concentration.



In the fully- , biogeochemically-, and radiatively-coupled versions of the 1pctCO2 experiments
analyzed here, the rate of change of atmospheric carbon $dC_A/dt$ is specified in equations (A1)
and (A2). The uptake or release of $CO_2$ by the underlying land and ocean yields an effective
emission *E* which serves to maintain the budget.

The changes in atmosphere carbon budgets, from the pre-industrial control simulation, in the
differently coupled simulations are represented as

Radiatively-coupled $$\frac{dC_A'}{dt} - E^+ = F_A^+ = -F_L^+ - F_O^+ = \Gamma_A T^+ \qquad \text{(A3a)}$$
Biogeochemically-coupled $$\frac{dC_A'}{dt} - E^* = F_A^* = -F_L^* - F_O^* = \Gamma_A T^* + B_A c' \qquad \text{(A3b)}$$
Fully-coupled $$\frac{dC_A'}{dt} - E = F_A' = -F_L' - F_O' = \Gamma_A T' + B_A c' \qquad \text{(A3c)}$$


which serve to define the instantaneous carbon-concentration ($B_A$) and carbon-climate ($\Gamma_A$)
feedback parameters and assume linearization of the globally integrated surface-atmosphere
$CO_2$ flux in terms of global mean temperature and concentration change. In equation (A3), $F^+$,
$F^*$, and $F'$ are the flux changes and $T^+$, $T^*$, and $T'$ the temperature changes in the radiatively-,
biogeochemically- and fully-coupled simulations, and $E^+$, $E^*$, and $E$ are the resulting implicit
emissions. $c'$ is the specified $CO_2$ concentration change above its pre-industrial level in the
1pctCO2 simulations. In the biogeochemically-coupled simulation there is no radiative forcing
due to increasing $CO_2$ so $T^*$ is small, although not zero and exhibits a distinct spatial pattern. The



assumption made in equation (A3) is that the feedback parameters are the same in the three
cases.

Carbon budget changes for the land component parallel (A3) but without the emissions terms as
Radiatively-coupled
$$\frac{dC'_L}{dt} = F^+_L = \Gamma_L T^+ \tag{A4a}$$

Biogeochemically-coupled
$$\frac{dC^*_L}{dt} = F^*_L = \Gamma_L T^* + B_L c' \tag{A4b}$$

Fully-coupled
$$\frac{dC^*_L}{dt} = F'_L = \Gamma_L T' + B_L c' \tag{A4c}$$


and similarly for the ocean component. Since $F_A = -(F_L + F_O)$ it follows that $\Gamma_A = -(\Gamma_L + \Gamma_O)$
and $B_A = -(B_L + B_O)$. There are no terms involving $c'$ in the radiatively-coupled simulation
(equations 3a and 4a) since the pre-industrial value of atmospheric $CO_2$ concentration is
prescribed for the biogeochemistry components so $c' = 0$ and does not affect the flux.

The instantaneous feedback parameters ($B_L$ and $\Gamma_L$) differ from that in the integrated flux
approach of Friedlingstein et al. (2006) who express time integrated flux changes (i.e. change in
pool or reservoir sizes) as functions of temperature and $CO_2$ concentration changes with
Radiatively-coupled
$$\int F^+_L = \Delta C^+_L = \gamma_L T^+ \tag{A5a}$$

Biogeochemically-coupled
$$\int F^*_L = \Delta C^*_L = \gamma_L T^* + \beta_L c' \tag{A5b}$$

Fully-coupled
$$\int F'_L = \Delta C'_L = \gamma_L T' + \beta_L c' \tag{A5c}$$




and similarly for the ocean component, with the assumption that the $\Delta C'_O$ term includes changes
in the carbon amount of ocean sediment as well.

The units of instantaneous and integrated flux based parameters are different ($\Gamma$ - PgC yr$^{-1}$ °C$^{-1}$,
B - PgC yr$^{-1}$ ppm$^{-1}$ and $\gamma$ - PgC °C$^{-1}$, $\beta$ - PgC ppm$^{-1}$). Arora et al. (2013) show how the
instantaneous and integrated flux based feedback parameters are related to each other

Integrating equations (A1) and (A2) from initial time to $t$ gives
$$\Delta C'_A + \Delta C'_L + \Delta C'_O = \int_0^t E\, dt = \tilde{E} \qquad \text{(A6)}$$

Where $\Delta C'_A = 2.12\,(c(t) - c(0))$ is the change in atmospheric carbon burden (the factor 2.12
converts atmospheric CO$_2$ concentration from ppm to atmospheric burden in PgC) and $\Delta C'_X =$
$\int_0^t F'_X\, dt, X = L, O$ is the cumulative flux equal to the change in the land or ocean carbon pool
for the fully-coupled simulation. The terms in equation (A6) indicate the contribution of changes
in atmosphere, land and ocean carbon pools to cumulative emissions $\tilde{E}$. Finally, division by the
cumulative emissions term in equations (A6) gives all the terms in a fractional form as

$$f_A + f_L + f_O = 1 \qquad \text{(A7)}$$




where $f_A$ is the airborne fraction of cumulative emissions and $f_L$ and $f_O$ are fractional emissions
taken up by the land and ocean. These components are evaluated at the time of $CO_2$ quadrupling.

**A2. Justification for using BGC and COU simulations for finding feedback parameters**
Figures 6 and 7 provide justification for using the BGC-COU approach, over the RAD-BGC and
RAD-COU approaches, in calculating the feedback parameters as discussed below. In Figure 7,
the absolute magnitude of $\gamma_O$ when using the BGC-COU approach is about twice in CMIP5 models
(and more than three times in CMIP6 models) compared to its model-mean value calculated using
the RAD-BGC and RAD-COU approaches. The reason for this is that the RAD simulation misses the
suppression (due to weakening of the ocean circulation) of carbon drawdown to the deep ocean.
This is because there is no buildup of a strong carbon gradient from the atmosphere to the deep
ocean in the RAD simulation. This process is important when climate change is forced by
increasing atmospheric $CO_2$, and therefore feedback parameters calculated using the BGC-COU
approach are more likely to include all processes relevant to application for realistic scenarios. In
Figure 6, although the carbon-climate feedback parameter over land ($\gamma_L$) is larger in absolute
amount, it is comparatively less sensitive to the approach used, than over ocean, because over
land an increase in temperature not only increases the respiratory losses but also affects
photosynthetic processes especially in conjunction with increasing $CO_2$. Warmer temperatures
increase photosynthesis over mid to high latitude regions where photosynthesis is currently
limited by temperature and more so with increasing $CO_2$, but decrease photosynthesis over
tropical regions where the temperatures are already too warm for optimal photosynthesis. The



net result of these compensating processes plays out very differently in different models and in
the model-mean sense this results in less sensitivity of the calculated value of carbon climate
feedback parameter over land ($\gamma_L$) to the different approaches than over ocean. This is seen in
both CMIP5 and CMIP6 models. When $\gamma_L$ is calculated using the RAD-BGC and RAD-COU
approaches, it is exclusively calculated using results from the RAD simulation. However, since
over land photosynthesis is also affected by temperature in addition to respiration (with widely
varying responses between models) the $\gamma_L$ values vary widely between models between the RAD-
BGC/RAD-COU approach and the BGC-COU approach. This is seen, for example, for ACCESS-
ESM1.5, IPSL, and CanESM5 models in Figure 6b. The very different values of $\gamma_L$ for individual
models, when using different approaches to calculate them, are the result of the differing
responses of the vegetation and soil+litter carbon pools, in the RAD and COU simulations, and
this is supported by results that were presented in Section 4.3.2.

In Figure 7 value of $\gamma_O$ changes sign for the CNRM-ESM2-1 model from positive when calculated
using the RAD-BGC or RAD-COU approaches to negative when calculated using the BGC-COU
approach and this further illustrates the sensitivity of feedback parameters to the approach used
to calculate them. This non-linear behaviour for a previous version of the CNRM model has been
document in Schwinger et al. (2014) and caused by the large increase in regenerated DIC in the
RAD simulation, similar to the increase in the COU relative to the BGC simulation, as shown in
Figure 14 for the CNRM-ESM2-1 model. This non-linear behaviour is stronger in CNRM-ESM2-1,
compared to CNRM-ESM1, its previous version (Séférian et al., 2016), most likely due to a new
parameterization for N fixation which increases ocean NPP and a revised parameterization for


organic matter remineralization in the model's ocean biogeochemistry component (PISCESv2-
gas). A contribution to a positive $\gamma_O$ is also made by declining sea ice in the RAD simulation which
leads to changes in the sign of the air-sea carbon exchange in the Southern Ocean. The vertical
profile of dissolved inorganic carbon in the Southern Ocean in BGC and COU simulations (with
rising $[CO_2]$) is different from that in the RAD simulation (for the preindustrial $[CO_2]$) and this
leads to additional non-linearities.





**A3. Additional Figures**

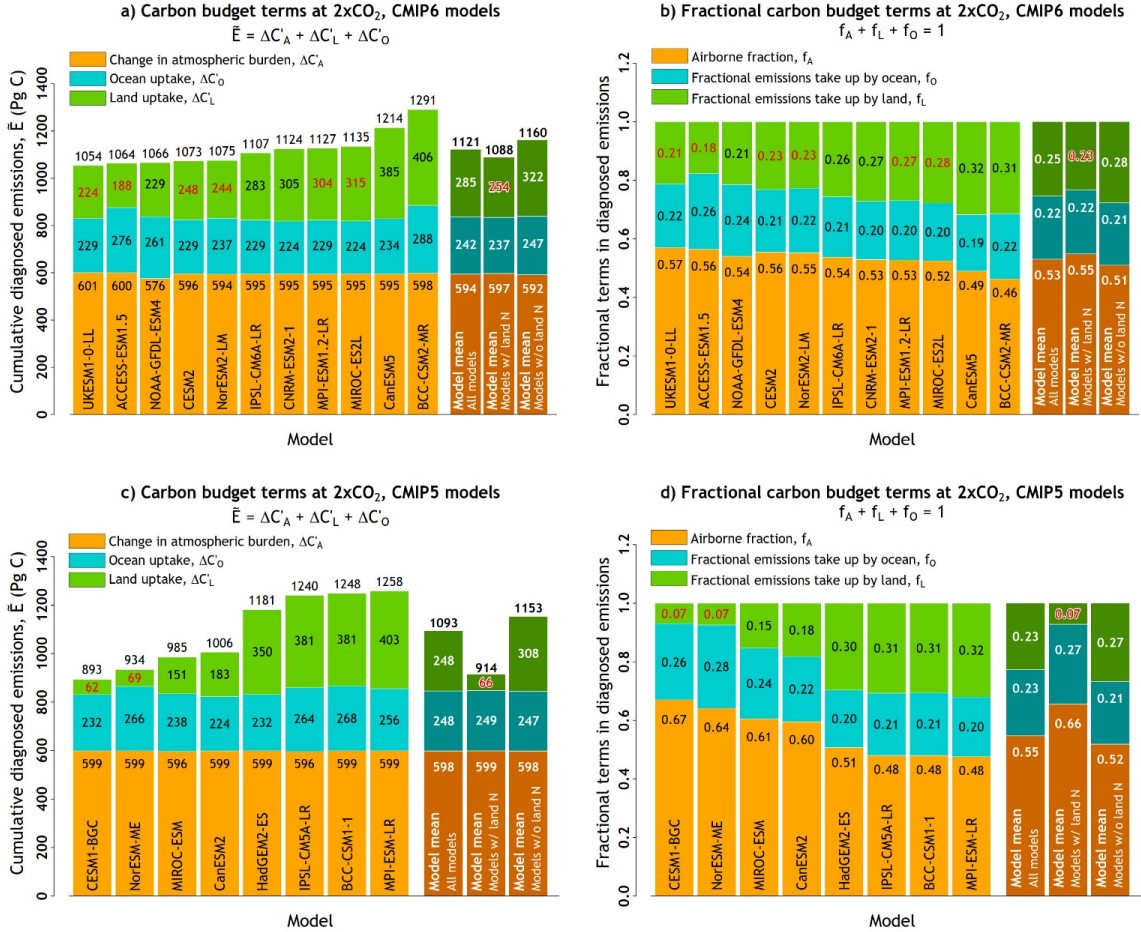

Figure A1: Components of the carbon budget terms in cumulative emissions from the eleven
participating CMIP6 models based on equation (15) in panel (a) and equation (16) in panel (b)
using results from the fully-coupled 1% per year increasing $CO_2$ simulation but at $2\times CO_2$ (year 70)
in contrast to Figure 5 which showed these results at $4\times CO_2$. The models are arranged in an
ascending order based on their cumulative emissions values. Results from participating CMIP5
models in the A13 study are shown in panels c and d. In addition, ESMs whose land component
includes a representation of N cycle are identified by red font colour for cumulative land carbon
uptake (panels a and c) and fractional emissions taken up by land (panels b and d). Model mean
is shown for all models but also separately for models whose land components include or do not
include a representation of the N cycle.






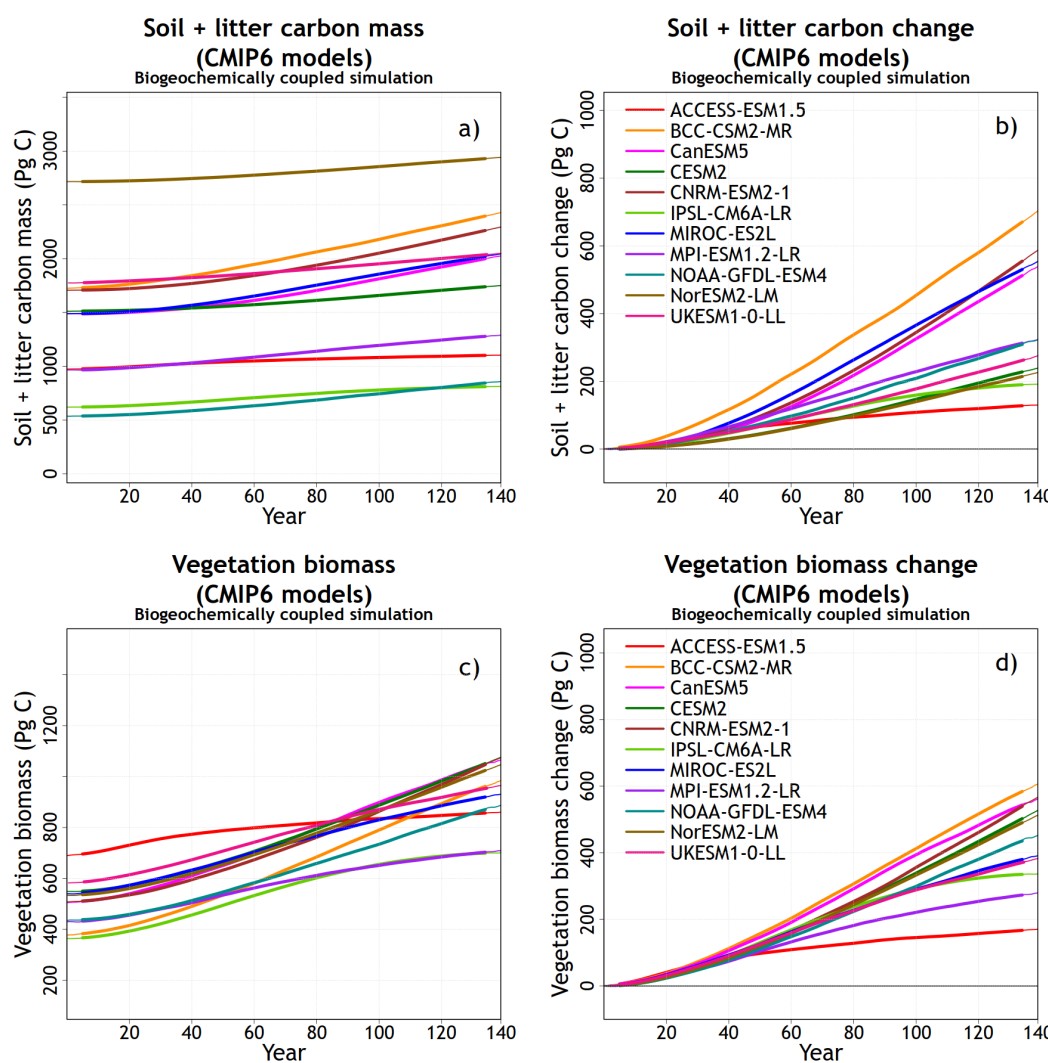


Figure A2: Absolute amounts and the change from the beginning of the BGC simulation for carbon
in soil+litter (panels a and b) and vegetation (panels c and d) pools.







**A4. Additional tables**


**Table A1**: Values of carbon-concentration and carbon-climate feedback parameters for land and
ocean calculated using the B-C approach (using results from the COU and BGC simulations), and
the linear transient climate sensitivity to $CO_2$, from CMIP6 and CMIP5 models at $4\times CO_2$ (i.e. at
the end of the 1pctCO2 simulation) and $2\times CO_2$.


| CMIP6 models at 4×CO₂ | | | | | |
|---|---|---|---|---|---|
| | Land | | Ocean | | |
| | Carbon-climate feedback, $\gamma_L$ | Carbon-concentration feedback, $\beta_L$ | Carbon-climate feedback, $\gamma_O$ | Carbon-concentration feedback, $\beta_O$ | Climate sensitivity, $\alpha$ |
| | PgC °C⁻¹ | PgC ppm⁻¹ | PgC °C⁻¹ | PgC ppm⁻¹ | °C ppm⁻¹ |
| ACCESS-ESM1.5 | -21.1 | 0.37 | -23.75 | 0.9 | 0.00546 |
| BCC-CSM2-MR | -163.1 | 1.81 | -19.94 | 0.92 | 0.00485 |
| CanESM5 | 15.95 | 1.28 | -14.72 | 0.77 | 0.00751 |
| CESM2 | -21.6 | 0.9 | -10.85 | 0.71 | 0.00637 |
| CNRM-ESM2-1 | -83.11 | 1.36 | -9.38 | 0.7 | 0.00632 |
| IPSL-CM6A-LR | -8.67 | 0.62 | -12.97 | 0.76 | 0.00687 |
| MIROC-ES2L | -69.57 | 1.12 | -22.25 | 0.73 | 0.00436 |
| MPI-ESM1.2-LR | -5.17 | 0.71 | -20.11 | 0.77 | 0.00512 |
| NOAA-GFDL-ESM4 | -80.06 | 0.93 | -21.65 | 0.84 | 0.00430 |
| NorESM2-LM | -20.95 | 0.85 | -19.64 | 0.78 | 0.00410 |
| UKESM1-0-LL | -38.4 | 0.75 | -14.07 | 0.75 | 0.00721 |
| **Model mean** | **-45.07** | **0.97** | **-17.21** | **0.78** | **0.00568** |
| **Standard deviation** | **48.24** | **0.38** | **4.72** | **0.07** | **0.00118** |


| CMIP6 models at 2×CO₂ | | | | | |
|---|---|---|---|---|---|
| | Land | | Ocean | | |
| | Carbon-climate feedback, $\gamma_L$ | Carbon-concentration feedback, $\beta_L$ | Carbon-climate feedback, $\gamma_O$ | Carbon-concentration feedback, $\beta_O$ | Climate sensitivity, $\alpha$ |
| | | PgC °C⁻¹ | PgC ppm⁻¹ | PgC °C⁻¹ | PgC ppm⁻¹ |
| ACCESS-ESM1.5 | -12 | 0.75 | -11.72 | 1.06 | 0.00750 |
| BCC-CSM2-MR | -132.84 | 2.22 | -12.38 | 1.09 | 0.00592 |
| CanESM5 | -6.22 | 1.42 | -7.71 | 0.9 | 0.00950 |
| CESM2 | -12.76 | 0.98 | -4.24 | 0.84 | 0.00789 |
| CNRM-ESM2-1 | -44.51 | 1.37 | -3.58 | 0.81 | 0.00650 |
| IPSL-CM6A-LR | -12.24 | 1.11 | -7.37 | 0.87 | 0.00876 |
| MIROC-ES2L | -63.36 | 1.45 | -10.44 | 0.85 | 0.00530 |
| MPI-ESM1.2-LR | -0.81 | 1.08 | -11.4 | 0.88 | 0.00636 |
| NOAA-GFDL-ESM4 | -50.69 | 1.08 | -8.97 | 0.97 | 0.00543 |
| NorESM2-LM | -15.61 | 0.94 | -9.34 | 0.88 | 0.00509 |
| UKESM1-0-LL | -24.01 | 1 | -7.35 | 0.88 | 0.00885 |
| **Model mean** | **-34.10** | **1.22** | **-8.59** | **0.91** | **0.00701** |
| **Standard deviation** | **36.61** | **0.38** | **2.76** | **0.09** | **0.00150** |






| CMIP5 models at 4×CO$_2$ | | | | | |
|---|---|---|---|---|---|
| | Land | | Ocean | | |
| | Carbon-climate feedback, $\gamma_L$ | Carbon-concentration feedback, $\beta_L$ | Carbon-climate feedback, $\gamma_O$ | Carbon-concentration feedback, $\beta_O$ | Climate sensitivity, $\alpha$ |
| | | PgC °C$^{-1}$ | PgC ppm$^{-1}$ | PgC °C$^{-1}$ | PgC ppm$^{-1}$ |
| BCC-CSM1-1 | -109.7 | 1.4 | -17.4 | 0.85 | 0.00511 |
| CanESM2 | -64.9 | 0.99 | -11.28 | 0.7 | 0.00623 |
| CESM1-BGC | -6.39 | 0.24 | -12.16 | 0.74 | 0.00481 |
| IPSL-CM5A-LR | -46.65 | 1.13 | -17.6 | 0.89 | 0.00559 |
| MIROC-ESM | -86.82 | 0.75 | -20.94 | 0.82 | 0.00660 |
| MPI-ESM-LR | -89.64 | 1.49 | -18.36 | 0.85 | 0.00582 |
| NorESM-ME | -4.3 | 0.22 | -18.72 | 0.87 | 0.00441 |
| HadGEM2-ES | -54.94 | 1.24 | -21.88 | 0.82 | 0.00607 |
| **Model mean** | **-57.92** | **0.93** | **-17.29** | **0.82** | **0.00558** |
| **Standard deviation** | **35.77** | **0.46** | **3.54** | **0.06** | **0.00070** |


| CMIP5 models at 2×CO$_2$ | | | | | |
|---|---|---|---|---|---|
| | Land | | Ocean | | |
| | Carbon-climate feedback, $\gamma_L$ | Carbon-concentration feedback, $\beta_L$ | Carbon-climate feedback, $\gamma_O$ | Carbon-concentration feedback, $\beta_O$ | Climate sensitivity, $\alpha$ |
| | | PgC °C$^{-1}$ | PgC ppm$^{-1}$ | PgC °C$^{-1}$ | PgC ppm$^{-1}$ |
| BCC-CSM1-1 | -57.61 | 1.75 | -11.06 | 1.03 | 0.00676 |
| CanESM2 | -48.13 | 1.05 | -6.64 | 0.85 | 0.00830 |
| CESM1-BGC | -5.02 | 0.25 | -4.41 | 0.86 | 0.00603 |
| IPSL-CM5A-LR | -37.28 | 1.58 | -8.88 | 0.99 | 0.00609 |
| MIROC-ESM | -64.79 | 1.04 | -12.36 | 0.94 | 0.00778 |
| MPI-ESM-LR | -62.52 | 1.86 | -11.24 | 0.99 | 0.00686 |
| NorESM-ME | 1.02 | 0.24 | -9.53 | 1 | 0.00506 |
| HadGEM2-ES | -21.78 | 1.43 | -11.27 | 0.92 | 0.00836 |
| **Model mean** | **-37.01** | **1.15** | **-9.42** | **0.95** | **0.00690** |
| **Standard deviation** | **24.17** | **0.59** | **2.53** | **0.06** | **0.00110** |






**Table A2**: Estimate of the change in the ocean carbon inventory (PgC) expected from a time integral of the global air-sea carbon flux into the ocean versus the volume integral of the change in the dissolved inorganic carbon, together with the small residual. The time integral of the air-sea carbon flux provides the dominant contribution to the change in the ocean carbon inventory, although there is a small mismatch due to the land to ocean carbon flux from river runoff and the ocean to land carbon flux from carbon burial in ocean sediments.

| Model | Time integral of the global air-sea carbon flux into the ocean (PgC) | Global ocean volume integral of $\Delta$DIC (PgC) | Residual (PgC) |
|---|---|---|---|
| ACCESS-ESM1.5 | 763 | 736 | 27 |
| CanESM5 | 656 | 651 | 5 |
| CNRM-ESM2-1 | 597 | 658 | -61 |
| MIROC-ES2L | 625 | 632 | -7 |
| MPI-ESM1.2-LR | 657 | 621 | 36 |
| NOAA-GFDL-ESM4 | 720 | 759 | -39 |
| NorESM2-LM | 671 | 628 | 43 |
| UKESM1-0-LL | 637 | 609 | 28 |
| Model mean ($\bar{x}$) | 666 | 662 | |
| Standard deviation ($\sigma_x$) | 53 | 55 | |
| Coefficient of variation ($\sigma_x/|\bar{x}|$) | 0.08 | 0.08 | |






**Table A3**: Carbon-cycle feedback parameters for the ocean, $\beta_O$ and $\gamma_O$, diagnosed from the air-
sea carbon fluxes and separately diagnosed for the ocean carbon inventory and its separate
ocean saturated, disequilibrium and regenerated DIC pools for the subset of eight CMIP6 models
for which 3D ocean data were available; their sum does not exactly match the diagnostics from
the air-sea fluxes due to land-ocean interactions involving storage in sediments and river inputs.

| | Carbon-concentration feedback (PgC ppm$^{-1}$) | | | | Carbon-climate feedback (PgC °C$^{-1}$) | | | |
|---|---|---|---|---|---|---|---|---|
| | $\beta_O$ | $\beta_{sat}$ | $\beta_{dis}$ | $\beta_{reg}$ | $\gamma_O$ | $\gamma_{sat}$ | $\gamma_{dis}$ | $\gamma_{reg}$ |
| ACCESS-ESM1.5 | 0.90 | 3.54 | -2.69 | 0.005 | -23.75 | -13.60 | -20.47 | 11.52 |
| CanESM5 | 0.77 | 3.83 | -3.06 | -0.001 | -14.72 | -10.72 | -8.62 | 4.29 |
| CNRM-ESM2-1 | 0.70 | 3.75 | -3.01 | 0.03 | -9.38 | -14.56 | -17.66 | 29.27 |
| MIROC-ES2L | 0.73 | 3.76 | -3.01 | -0.001 | -22.25 | -16.48 | -25.50 | 21.08 |
| MPI-ESM1.2-LR | 0.77 | 3.34 | -2.62 | 0.002 | -20.11 | -14.37 | -15.37 | 8.40 |
| NorESM2-LM | 0.78 | 3.67 | -2.92 | -0.004 | -19.64 | -12.91 | -14.44 | 9.19 |
| UKESM1-0-LL | 0.75 | 3.62 | -2.88 | -0.02 | -14.07 | -8.87 | -11.04 | 6.56 |
| NOAA-GFDL-ESM4 | 0.84 | 3.77 | -2.93 | 0.05 | -21.65 | -10.75 | -17.77 | 7.7 |
| Model mean ($\bar{x}$) | 0.78 | 3.66 | -2.89 | -0.003 | -16.95 | -12.78 | -16.36 | 12.25 |
| Standard deviation ($\sigma_x$) | 0.06 | 0.16 | 0.16 | 0.009 | 5.62 | 2.50 | 5.31 | 8.53 |
| Coefficient of variation ($\sigma_x/|\bar{x}|$) | 0.08 | 0.05 | 0.06 | 3.00 | 0.33 | 0.20 | 0.33 | 0.70 |


**Table A4**: Transient Climate Response (TCE, $\Delta T_{2\times CO2}$), diagnosed cumulative emissions at
2×CO$_2$ ($\tilde{E}_{2\times CO2}$), and transient climate response to cumulative emissions (TCRE) for the eleven
CMIP6 models considered in this study.

| CMIP6 model | TCR (°C) | Cumulative diagnosed emissions (PgC) | TCRE (°C EgC$^{-1}$) |
|---|---|---|---|
| ACCESS-ESM1.5 | 2.13 | 1064 | 2.00 |
| BCC-CSM2-MR | 1.68 | 1291 | 1.30 |
| CanESM5 | 2.69 | 1214 | 2.21 |
| CESM2 | 2.24 | 1073 | 2.08 |
| CNRM-ESM2-1 | 1.84 | 1124 | 1.64 |
| IPSL-CM6A-LR | 2.48 | 1107 | 2.24 |
| MIROC-ES2L | 1.50 | 1135 | 1.32 |
| MPI-ESM1.2-LR | 1.80 | 1127 | 1.60 |
| NOAA-GFDL-ESM4 | 1.54 | 1066 | 1.44 |
| NorESM2-LM | 1.44 | 1075 | 1.34 |
| UKESM1-0-LL | 2.51 | 1054 | 2.38 |
| Mean | 1.99 | 1121 | 1.78 |
| Standard deviation | 0.42 | 70 | 0.39 |







**A5. Model descriptions**
**A5.1. Commonwealth Scientific and Industrial Research Organisation (CSIRO) ACCESS-ESM1.5**
The Australian Community Climate and Earth System Simulator ACCESS-ESM1.5 (Ziehn et al.,
2017; Ziehn et al., 2019, The Australian Earth System Model: ACCESS-ESM1.5, in prep) is
comprised of a number of component models. The atmospheric model is the UK Met Office
Unified Model at version 7.3 (Martin et al., 2010, 2011) with their land surface model replaced
with the Community Atmosphere Biosphere Land Exchange (CABLE) model (Kowalczyk et al.,
2013). The ocean component is the NOAA/GFDL Modular Ocean Model (MOM) at version 5
(Griffies, 2014) with the same configuration as the ocean model component of ACCESS1.0 and
ACCESS1.3 (Bi et al., 2013). Sea ice is simulated using the LANL CICE4.1 model (Hunke and
Lipscomb, 2010). Coupling of the ocean and sea-ice to the atmosphere is through the OASIS-MCT
coupler(Valcke, 2013). The physical climate model configuration used here is very similar to the
version (ACCESS1.3) that contributed to the Coupled Model Intercomparison Project Phase 5
(CMIP5) (Bi et al., 2013). The carbon cycle is included in ACCESS through the CABLE land surface
model and its biogeochemistry module, CASA-CNP (Wang et al., 2010), and through the World
Ocean Model of Biogeochemistry and Trophic-dynamics (WOMBAT) (Oke et al., 2013).

The WOMBAT model is based on a NPZD (nutrient-phosphate, phytoplankton, zooplankton and
detritus) model with the additions of bio-available iron limitation, dissolved inorganic carbon,
calcium carbonate, alkalinity and oxygen.  Productivity drives uptake and formation of carbon
and oxygen which exchange with the atmosphere.  Sinking and remineralization of detritus





carries biogeochemical tracers to the deep ocean.  Iron is supplied by dust deposition, continental
shelves and background ocean values.

The Australian community model CABLE simulates the fluxes of momentum, heat, water and
carbon at the surface. The biogeochemistry module CASA-CNP simulates the flow of carbon and
nutrients such as nitrogen and phosphorus between three plant biomass pools (leaf, wood, root),
three litter pools (metabolic, structural, coarse woody debris) and three organic soil pools
(microbial, slow, passive) plus one inorganic soil mineral nitrogen pool and three phosphorus soil
pools.

In the CABLE configuration applied here we use 10 vegetated types and 3 non-vegetated types.
CABLE calculates gross primary production (GPP) and leaf respiration at every time step using a
two-leaf canopy scheme (Wang and Leuning, 1998) as a function of the leaf area index (LAI). This
set-up uses a simulated (prognostic) LAI based on the size of the leaf carbon pool and the specific
leaf area. Daily mean GPP and leaf respiration values are then passed onto CASA-CNP to calculate
daily respiration fluxes and the flow of carbon and nutrients between the pools. Similar to the
previous version, ACCESS-ESM1 (Law et al., 2017; Ziehn et al., 2017), the model is run with
nitrogen and phosphorus limitation enabled.



**A5.2. Beijing Climate Center (BCC) Climate System Model version 2 with Medium Resolution**
**(BCC-CSM2-MR)**
BCC-CSM2-MR (Wu et al., 2019) is the second generation of the BCC model with medium
resolution that was released to run CMIP6 simulations. It is a fully-coupled global climate model
and updated from its previous version of BCC-CSM1.1 used for CMIP5 (Wu et al., 2013). The
atmospheric component of BCC-CSM2-MR is the BCC Atmospheric General Circulation Model
version 3 (BCC-AGCM3-MR, Wu et al., 2019). The land component is the BCC Atmosphere and
Vegetation Interaction Model version 2.0 (BCC-AVIM2, Li et al., 2019) with terrestrial carbon
cycle.  The oceanic component is the Modular Ocean Model version 4 with 40 levels (hereafter
MOM4-L40). The sea ice component is Sea Ice Simulator (SIS). These components are physically
coupled through fluxes of momentum, energy, water, and carbon at their interfaces. The
coupling was realized with the flux coupler version 5 developed by the National Center for
Atmosphere Research (NCAR).

The atmospheric component of BCC-CSM2-MR has a horizontal resolution of T106 approximately
1.125° and 46 vertical levels in a hybrid sigma/pressure vertical coordinate system with the top
level at 1.459 hPa. The ocean component resolution of BCC-CSM2-MR is 1° longitude by 1/3°
latitude between 30°S and 30°N ranged to 1° latitude at 60°S and 60°N and nominally 1°
polarward with tripolar coordinates, and there are 40 z-levels in the vertical.



The atmospheric component model BCC-AGCM3-MR in BCC-CSM2-MR is developed from its
previous CMIP5 version (Wu et al., 2008). The main updates include a modification of deep
convection parameterization, a new scheme for cloud fraction, indirect effects of aerosols
through clouds and precipitation, and the gravity wave drag generated by deep convection (Wu
et al., 2019).Atmospheric $CO_2$ concentration in BCC-AGCM3-MR for this work is a prognostic
variable and calculated through a budget equation which considered advective transport in the
atmosphere, anthropogenic $CO_2$ emissions, and interactive $CO_2$ fluxes at the interfaces with land
and ocean. But chemical processes are not taken into account. The terrestrial carbon cycle in
BCC-AVIM2 (Li et al., 2019) operates through a series of biochemical and physiological processes
on photosynthesis and respiration of vegetation, and takes into account carbon loss due to
turnover and mortality of vegetation, and $CO_2$ release into atmosphere through soil respiration.
The vegetation litter to the ground surface and into the soil is divided into eight terrestrial carbon
pools (surface structural, surface metabolic, surface microbial, soil structural, soil metabolic, soil
microbial, slow, and passive carbon pools) according to the timescale of the decomposition of
carbon in each pool and transfers between different pools. Allocation to and from the three
vegetation biomass pools (leaf, stem, root) leads to dynamic vegetation that in turn produces
litter fall and ultimate transfer to soil organic carbon. The allocation of carbon to the three
vegetation biomass pools is dependent on light availability, water stress and phenology stages of
the canopy and follows the formulations of Arora and Boer (2005).

The biogeochemistry module to simulate the ocean carbon cycle in MOM4_L40 is based on the
protocols from the Ocean Carbon Cycle Model Intercomparison Project–Phase 2 (OCMIP2,



http://www.ipsl.jussieu.fr/OCMIP/phase2/). The OCMIP biogeochemistry module parameterizes
the process of marine biology in terms of geochemical fluxes without explicit representation of
the marine ecosystem and food web processes, and includes five prognostic variables:
phosphate, dissolved organic phosphorus, dissolved oxygen, dissolved inorganic carbon, and
alkalinity. Ocean carbon cycle processes in BCC-CSM2-MR follow OCMIP, except for
parameterizing the export of organic matter from surface waters to deep oceans (Wu et al.,

1536    2013).


**A5.3. Canadian Centre for Climate Modelling and Analysis (CCCma) fifth generation Earth**
**System Model, CanESM5**
CanESM5 has evolved from its predecessor CanESM2 (Arora et al., 2011) that was used in the
Coupled Model Intercomparison Project phase 5 (CMIP5). CanESM5 represents a major update
to CanESM2 and described in detail in Swart et al. (2019). The major changes relative to CanESM2
are the implementation of completely new models for the ocean, sea-ice, marine ecosystems,
and a new coupler. The resolution of CanESM5 (T63 or ~2.8° in the atmosphere and ~1° in the
ocean) remains similar to CanESM2, and is at the lower end of the spectrum of CMIP6 models.
The atmospheric component of CanESM5 is represented by version 5 of the Canadian
Atmospheric Model (CanAM5) has several improvements relative to its predecessor, CanAM4
(von Salzen et al., 2013) including changes to aerosol, clouds, radiation, land surface and lake
processes. CanAM5 uses a triangular spectral truncation in the model dynamical core, with an
approximate horizontal resolution of 2.8 degrees in latitude/longitude. It uses a hybrid vertical

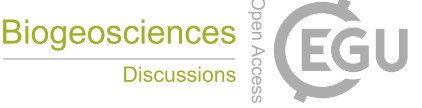

coordinate system with 49 levels between the surface and 1 hPa, with a vertical resolution of
about 100 m near the surface. Relative to the 35 levels used in CanESM2 most of the additional
14 levels were added in the upper troposphere and stratosphere.

The land surface in CanESM5 is modelled using the Canadian Land Surface Scheme (CLASS;
Verseghy, 2000) and the Canadian Terrestrial Ecosystem Model (CTEM; Arora and Boer, 2005,
2010) which together form the land component of CanESM5. CLASS-CTEM simulate the physical
and biogeochemical land surface processes, respectively, and together they calculate fluxes of
energy, water, $CO_2$ and wetland $CH_4$ emissions at the land-atmosphere boundary. Over land,
three permeable soil layers are used with default thicknesses of 0.1, 0.25, and 3.75 m for which
liquid and frozen soil moistures and temperature are prognostically calculated. The depth to
bedrock is specified on the basis of the global data set which reduces thicknesses of the
permeable soil layers where soil depth is less than 4.1 meters. Snow is represented using one
layer whose snow water equivalent and temperature are modelled prognostically. The
introduction of dynamic wetlands and their methane emissions is a new biogeochemical process
added since the CanESM2 (Arora et al., 2018). Nitrogen cycle over land is not represented but
the effect of photosynthesis down-regulation as $CO_2$ increases is represented. The magnitude of
the parameter representing this down-regulation is increased in CanESM5, compared to
CanESM2, following Arora and Scinocca (2016) who found best value of this parameter that
reproduced various aspects of the historical carbon budget for CanESM4.2 (a model version more
similar to CanESM2 than CanESM5). Other than wetlands, and the changes to the strength of the





CO₂ fertilization effect, the remaining terrestrial ecosystem processes are represented the same
as in CanESM2.

The physical ocean component of CanESM5 is based on NEMO version 3.4.1. It is configured on
the tripolar ORCA1 C-grid with 45 z-coordinate vertical levels, varying in thickness from ~6 m near
the surface to ~250 m in the abyssal ocean. The horizontal resolution is based on a 1° Mercator
grid, varying with the cosine of latitude, with a refinement of the meridional grid spacing to 1/3°
near the equator. Two modifications have been introduced to the NEMO's mesoscale and small-
scale mixing physics in CanESM5 and these are detailed in Swart et al. (2019). Sea ice is
represented using the LIM2 sea ice model (Bouillon et al., 2009; Fichefet and Morales Maqueda,
1997), which is run within the NEMO framework.

Ocean carbon cycle is represented using the Canadian Model of Ocean Carbon (CMOC) which
was developed for earlier versions of CanESM (Arora et al., 2011; Christian et al., 2010), and
includes carbon chemistry and biology. The biological component is a simple Nutrient-
Phytoplankton-Zooplankton-Detritus (NPZD) model, with fixed Redfield stoichiometry, and
simple parameterizations of iron limitation, nitrogen fixation, and export flux of calcium
carbonate.

**A5.4. Community Earth System Model, version 2 (CESM2)**



The CESM2 (Danabasoglu et al., 2019: The Community Earth System Model version 2 - CESM2,
in preparation) contains substantial improvements since CESM1. The resolution remains the
same as in CESM1 (0.9° latitude x 1.25° longitude for the atmosphere and land with 32 vertical
atmospheric levels and 25 ground levels and ~1° for the ocean). The Community Atmosphere
Model version 6 (Neale, R. B. et al., 2019: The NCAR Community Atmosphere Model version 6
(CAM6): Scientific configuration and simulation fidelity, in preparation) includes many changes
to the representation of physical processes with the primary change being the inclusion of the
Cloud Layers Unified By Binormals (CLUBB) unified turbulence scheme.

The CESM2 ocean component (POP2) is largely the same as that used in CESM1 except with a
new parameterization for mixing effects in estuaries along with several other numerical and
physics improvements. The sea ice model is CICE version 5.1.2 (CICE5; (Hunke et al., 2015) .
Ocean biogeochemistry is represented by the Marine Biogeochemistry Library (MARBL). MARBL
represents multiple nutrient co-limitation (N, P, Si, and Fe). It includes three explicit
phytoplankton functional groups (diatoms, diazotrophs, and pico/nano phytoplankton), one
implicit phytoplankton group (calcifiers) and one zooplankton group. MARBL includes prognostic
carbonate chemistry and simulates sinking particulate organic matter. Major updates relative to
CESM1 include a representation of subgrid-scale variations in light and variable C:P
stoichiometry. Atmospheric deposition of iron is computed prognostically in CESM2 as a function
of dust and black carbon deposition simulated by CAM6. Riverine nutrient, carbon, and alkalinity
fluxes are supplied to the ocean from a dataset.




The land component is the Community Land Model version 5 (CLM5, Lawrence et al., 2018) which
simulates land water, energy, momentum, carbon and nitrogen cycling.  CLM5 includes an
extensive suite of new and updated processes and parameterizations that collectively improve
the model's hydrological, biogeochemical and ecological realism and enhance the representation
of anthropogenic land use activities on climate and the carbon cycle. The primary updates are as
follows with details, references, and additional updates described and listed in (Lawrence et al.,
2018): (1) updated parameterizations and structure for hydrology and snow (spatially explicit soil
depth, dry surface layer, revised groundwater scheme, revised canopy interception and canopy
snow processes, updated fresh snow density, and inclusion of the Model for Scale Adaptive River
Transport); (2) a plant hydraulics scheme to more mechanistically represent plant water use and
limitation; (3) vertically-resolved soil biogeochemistry with base organic matter decomposition
rates varying with depth and modified by soil temperature, water, and oxygen limitation and
nitrification and denitrification updated as in Century model; (4) a methane production,
oxidation, and emissions model; (5) improved representation of plant N dynamics to address
deficiencies in CLM4 through introduction of flexible plant carbon : nitrogen (C:N) stoichiometry
which avoids the problematic CLM4 separation of potential and actual plant productivity,
explicitly simulating photosynthetic capacity response to environmental conditions through the
Leaf Utilization of Nitrogen for Assimilation (LUNA) module, and accounting for how N availability
affects plant productivity through the Fixation and Uptake of Nitrogen (FUN) module which
determines the C costs of N acquisition; methane emissions and oxidation from natural land
processes; (6) a global active crop model with six crop types and time-evolving irrigated areas





and industrial fertilization rates; (7) updated canopy processes including a revised canopy
radiation scheme and canopy scaling of leaf processes, co-limitations on photosynthesis and
updated stomatal conductance; (8) a new fire model that includes representation of natural and
anthropogenic ignition sources and suppression along with agricultural, deforestation, and peat
fires; and (9) inclusion of carbon isotopes.

**A5.5. Centre National de Recherches Météorologiques (CNRM)  CNRM-ESM2-1**
CNRM-ESM2-1 is the second generation Earth System model developed by CNRM-CERFACS for
CMIP6 (Séférian et al., 2019).

The atmosphere component of CNRM-ESM2-1 is based on version 6.3 of the global spectral model
ARPEGE-Climat (ARPEGE-Climat_v6.3). ARPEGE-Climat resolves atmospheric dynamics and
thermodynamics on a T127 triangular grid truncation that offers a spatial resolution of about 150
km in both longitude and latitude. CNRM-ESM2-1 employs a ''high-top'' configuration with 91
vertical levels that extend from the surface to 0.01 hPa in the mesosphere; 15 hybrid σ-pressure
levels are available below 1500 m.

The surface state variables and fluxes at the surface-atmosphere interface are simulated by the
SURFEX modeling platform version 8.0 over the same grid and with the same time-step as the
atmosphere model. SURFEXv8.0 encompasses several submodules for modeling the interactions



between the atmosphere, the ocean, the lakes and the land surface. Over the land surface,
CNRM-ESM2-1 uses the ISBA-CTRIP land surface modeling system (http://www.umr-
cnrm.fr/spip.php?article1092&lang=en) to solve energy, carbon and water budgets at the land
surface (Decharme et al., 2019; Delire et al., 2019). Its physical core explicitly solves the one-
dimensional Fourier and Darcy laws throughout the soil, accounting for the hydraulic and thermal
properties of soil organic carbon. It uses a 12-layer snow model of intermediate complexity that
allows separate water and energy budgets for the soil and the snowpack. It accounts for a
dynamic river flooding scheme in which floodplains interact with the soil and the atmosphere
through free-water evaporation, infiltration and precipitation interception and a two-
dimensional diffusive groundwater scheme to represent unconfined aquifers and upward
capillarity fluxes into the superficial soil. More details on these physical aspects can be found in
Decharme et al. (2019).

To simulate the land carbon cycle and vegetation-climate interactions, ISBA-CTRIP simulates
plant physiology, carbon allocation and turnover, and carbon cycling through litter and soil. It
includes a module for wild fires, land use and land cover changes, and carbon leaching through
the soil and transport of dissolved organic carbon to the ocean. Leaf photosynthesis is
represented by the semi-empirical model proposed by Goudriaan et al. (1985). Canopy level
assimilation is calculated using a 10-layer radiative transfer scheme including direct and diffuse
radiation. Vegetation in ISBA is represented by 4 carbon pools for grasses and crops (leaves, stem,
roots and a non-structural carbohydrate storage pool) with 2 additional pools for trees
(aboveground wood and coarse roots). Leaf phenology results directly from the carbon balance



of the leaves. The model distinguishes 16 vegetation types (10 tree and shrub types, 3 grass types
and 3 crop types) alongside desert, rocks and permanent snow. In the absence of nitrogen cycling
within the vegetation, an implicit nitrogen limitation scheme that reduces specific leaf area with
increasing $CO_2$ concentration was implemented in ISBA following the meta-analysis of Yin (2002).
Additionally, there is an ad-hoc representation of photosynthesis down-regulation. The litter and
soil organic matter module is based on the soil carbon part of the CENTURY model (Parton et al.,
1988). The 4 litter and 3 soil carbon pools are defined based on their location above- or below-
ground and potential decomposition rates. The litter pools are supplied by the flux of dead
biomass from each biomass reservoir (turnover). Decomposition of litter and soil carbon releases
$CO_2$ (heterotrophic respiration). During the decomposition process, some carbon is dissolved by
water slowly percolating through the soil column. This dissolved organic carbon is transported by
the rivers to the ocean. A detailed description of the terrestrial carbon cycle can be found in
Delire et al. (2019).

The ocean component of CNRM-ESM2-1 is the Nucleus for European Models of the Ocean
(NEMO) version 3.6 (Madec et al., 2017) which is coupled to both the Global Experimental Leads
and ice for ATmosphere and Ocean (GELATO) sea-ice model (Salas Mélia, 2002) version 6 and
also the marine biogeochemical model Pelagic Interaction Scheme for Carbon and Ecosystem
Studies version 2-gas (PISCESv2-gas). NEMOv3.6 operates on the eORCA1L75 grid (Mathiot et al.,
2017) which offers a nominal resolution of 1° to which a latitudinal grid refinement of 1/3° is
added in the tropics; this grid describes 75 ocean vertical layers using a vertical z*-coordinate
with partial step bathymetry formulation (Bernard et al., 2006).

Biogeosciences 
Discussions


The atmospheric chemistry scheme of CNRM-ESM2-1 is Reactive Processes Ruling the Ozone
Budget in the Stratosphere version 2 (REPROBUS-C_v2). This scheme resolves the spatial
distribution of 63 chemistry species but does not represent the low troposphere ozone non-
methane hydrocarbon chemistry. CNRM-ESM2-1 also includes an interactive tropospheric
aerosol scheme included in the atmospheric component ARPEGE-Climat. This aerosol scheme,
named Tropospheric Aerosols for ClimaTe In CNRM (TACTIC_v2), represents the main
anthropogenic and natural aerosol species of the troposphere.

The ocean biogeochemical component of CNRM-ESM2-1 uses the Pelagic Interaction Scheme for
Carbon and Ecosystem Studies model volume 2 version trace gases (PISCESv2-gas), which derives
from PISCESv2 as described in Aumont et al. (2015). PISCESv2-gas simulates the distribution of
five nutrients (from macronutrients: nitrate, ammonium, phosphate, and silicate to
micronutrient: iron) which regulate the growth of two explicit phytoplankton classes
(nanophytoplankton and diatoms). Dissolved inorganic carbon (DIC) and alkalinity (Alk) are
involved in the computation of the carbonate chemistry, which is resolved by "Model the Ocean
Carbonate SYstem" version 2 (MOCSY 2.0,Orr & Epitalon, 2015) in PISCESv2-gas. MOCSY 2.0
enables a better and faster resolution of the ocean carbonate chemistry at thermodynamic
equilibria. Oxygen is prognostically simulated using two different oxygen-to-carbon ratios, one
when ammonium is converted to or mineralized from organic matter, the other when oxygen is
consumed during nitrification. Their values have been set respectively to 131/122 and 32/122.






At ocean surface, PISCESv2-gas exchanges carbon, oxygen, dimethylsulfide (DMS) and nitrous
oxide ($N_2O$) tracers with the atmosphere using the revised air-sea exchange bulk as published by
Wanninkhof (2014). PISCESv2-gas uses several boundary conditions which represent the supply
of nutrients from five different sources: atmospheric deposition, rivers, sediment mobilization,
sea-ice and hydrothermal vents.

**A5.6. Institut Pierre Simon Laplace (IPSL) IPSL-CM6A-LR**
IPSL-CM6A-LR is the coupled climate model of the Institut Pierre Simon Laplace (Servonnat et al.,
2019, in preparation). It results from the integration of the following components: the LMDZ
atmospheric general circulation model (version 6A-LR, Hourdin et al., 2019), the NEMO oceanic
model (version 3.6, Aumont et al., 2015; Madec et al., 2017; Rousset et al., 2015; Vancoppenolle
et al., 2009) and the ORCHIDEE land surface model (version 2.0, Peylin et al., 2019, in
preparation).

The atmospheric general circulation model LMDZ6A-LR builds onto its previous version that has
notably incorporated advances in the parameterization of turbulence, convection, and clouds.
More specifically, LMDZ6A-LR includes a turbulent scheme based on the prognostic equation for
the turbulent kinetic energy that follows Yamada (1983), a mass flux representation of the
organized structures of the convective boundary layer called "Thermal Plume Model" (Hourdin



et al., 2002; Rio et al., 2010; Rio and Hourdin, 2008), and a parameterization of the cold pools or
wakes created below cumulonimbus by the evaporation of convective rainfall (Grandpeix et al.,
2010; Grandpeix and Lafore, 2010). It is based on a regular horizontal grid with 144 grid points
regularly spaced in longitude and 142 in latitude, corresponding to a resolution of 2.5° × 1.3°, and
79 vertical layers.

IPSL-CM6A-LR further includes NEMO (Nucleus for European Models of the Ocean), which is itself
composed of three major building blocks: the ocean physics NEMO-OPA (Madec et al., 2017), the
sea-ice dynamics and thermodynamics NEMO-LIM3 (Rousset et al., 2015; Vancoppenolle et al.,
2009), and the ocean biogeochemistry NEMO-PISCES (Aumont et al., 2015). The grid used has a
nominal resolution of 1° in the zonal and meridional directions with a latitudinal grid refinement
of 1/3° in the Tropics. Vertical discretization uses a partial step formulation (Bernard et al., 2006),
which ensures a better representation of bottom bathymetry, with 75 levels. The initial layer
thicknesses increase non-uniformly from 1 m at the surface to 10 m at 100 m depth, and reaches
200 m at the bottom, and are subsequently time-dependent. NEMO-PISCES (Aumont et al., 2015)
models the lower trophic levels of marine ecosystem (phytoplankton, microzooplankton and
mesozooplankton) and the biogeochemical cycles of carbon and of the main nutrients (P, N, Fe,
and Si). This model is also able to compute air-sea carbon fluxes.

Finally, IPSL-CM6A-LR includes ORCHIDEE, a global process-based terrestrial biosphere model
Krinner et al. (2005); Peylin et al., 2019, in preparation) that calculates carbon, water and energy



fluxes between the land surface and the atmosphere. Photosynthesis and all components of the
surface energy and water budgets are calculated at a half-hourly resolution while the dynamics
of the carbon storage (including carbon allocation in plant reservoirs, soil carbon dynamics, and
litter decomposition) are resolved on a daily basis. Photosynthesis depends on light availability
and $CO_2$ concentration, soil moisture and temperature and is parameterized based on Farquhar
et al. (1980) and Collatz et al. (1992) for $C_3$ and $C_4$ plants, respectively. This latest version of
ORCHIDEE includes a downregulation capability that models a reduction of the terrestrial
photosynthesis rates as a function of $CO_2$ concentration. In ORCHIDEE, the spatial distribution of
vegetation is represented using 15 plant functional types (PFTs) (Cramer, 1997; Prentice et al.,
1992; Wullschleger et al., 2014). More precisely these PFTs are decomposed into 3 groups
according to their physiological behavior under similar climate conditions: tall vegetation
(forests) is represented by 8 PFTs, short vegetation (grasses and crops) is represented by 6 PFTs,
and bare soil. The fractional coverage of these PFTs vary geographically. A soil type is associated
with each one of these 3 PFT groups. This 3-group partitioning allows for dividing each grid box
into 3 tiles for which an independent hydrological budget is calculated, using the 11-layer
physically based hydrology scheme. In ORCHIDEE the wood harvest product from the LUHv2h
database is used in addition to the annual land cover maps.

**A5.7. Team MIROC (Japan Agency for Marine-Earth Science and Technology / the University of**
**Tokyo / the National Institute for Environmental Studies) MIROC-ES2L**



MIROC-ES2L (Hajima et al., 2019a) is based on the global climate model MIROC5.2 (Tatebe et al.,
2018), which is a minor updated version of MIROC5 used for CMIP5 (Watanabe et al., 2010). The
physical core shares almost same structure and characteristics with the latest model MIROC6
(Tatebe et al., 2019), except for the atmospheric spatial resolution and treatment of cumulus
clouds. This model interactively couples an atmospheric general circulation model (CCSR-NIES
AGCM, Tatebe et al., 2019) including an on-line aerosol component (SPRINTARS, Takemura et al.,
2000), an ocean GCM with sea-ice component (COCO, Hasumi, 2015), and a land physical surface
model (MATSIRO, Takata et al., 2003). The land and ocean biogeochemical components are
represented by VISIT (Ito and Inatomi, 2012) and OECO2 (Hajima et al., 2019a), respectively,
which are interactively coupled to the atmospheric component. There exists another branched
version that has atmospheric chemistry component with finer atmospheric grid (MIROC-ES2H),
but not used in this study.

The atmospheric grid resolution is approximately 2.81° with 40 vertical levels between the
surface and about 3 hPa. For the ocean, the model employs tripolar coordinate system with 62
vertical levels. To the south of 63°N, the ocean model has longitudinal grid spacing of about 1°,
while the meridional grid spacing varies from about 0.5° near the equator to 1° in the mid-
latitudes. Over the Arctic ocean the grid resolution is even finer following the tripolar coordinate
system. The physical terrestrial component resolves vertical soil profile with 6 layers down to
14m depth, with two types of land-use tiles (agriculture and non-agriculture). Terrestrial
biogeochemical component considers two layered soil organic matter (the upper litter layer and



the lower humus layer), with 5 types of land-use tiles (primary vegetation, secondary vegetation,
urban, crop, and pasture).

The terrestrial biogeochemical component covers major processes relevant to global carbon
cycle, with vegetation (leaf, stem, and root), litter (leaf, stem, and root), and humus (active,
intermediate, and passive) pools and with a static biome distribution. Details on carbon cycle
processes in the model can been found in (Ito and Oikawa, 2002). N cycle is simulated with N
pools of vegetation (canopy and structural), organic soil (litter, humus, and microbe), and
inorganic nitrogen (ammonium and nitrate). The model considers two major nitrogen influxes
into ecosystem (biological nitrogen fixation and external nitrogen inputs). Fluxes out of land
ecosystem in the model are $N_2$/$N_2O$ emissions, leaching, $NH_3$ emission, and other emission like
volatilization from land-use product pools. For installing into MIROC-ES2L, the terrestrial
ecosystem processes were modified such that photosynthetic capacity is controlled by leaf N
concentration. Processes associated with land-use change are also modified to take full
advantage of CMIP6 LUC forcing dataset. Further details can be found in (Hajima et al., 2019a).

The new ocean biogeochemical component model, OECO2, is a NPZD-type model and modified
from the previous model (Watanabe et al., 2011). The biogeochemical compartments of OECO2
are nitrate, phosphate, dissolved iron, dissolved oxygen, two types of phytoplankton (non-
diazotroph and diazotroph), zooplankton, and particulate detritus. There exist other
compartments of dissolved inorganic carbon (DIC), total alkalinity, calcium, calcium carbonate,



and $N_2O$. All organic materials have identical elemental stoichiometric ratio. The model considers
external nutrient inputs (atmospheric N/Fe deposition, inorganic N/P from rivers, biological N
fixation, Fe input from ocean bottom/shelf) and nutrient loss (denitrification for N and loss into
sediment for N, P, and Fe). The emission, transportation and deposition processes of iron are
explicitly simulated by the atmospheric aerosol component.

**A5.8. Max Planck Institute for Meteorology (MPI) MPI-ESM1.2-LR**
The MPI-ESM1.2-LR model (Mauritsen et al., 2019) consists of ocean, atmosphere, land and sea-
ice components which are connected via a coupler analogous to the predecessor MPI-ESM
versions (Giorgetta et al., 2013). The atmosphere model, ECHAM6.3, at the LR resolution has a
spectral truncation at T63 or approximately 200-km grid spacing with 47 vertical levels. It is
directly coupled to the land model, JSBACH3.2, through surface exchange of mass, momentum,
and heat. The ocean general circulation model, MPIOM1.6 in MPI-ESM1.2-LR runs on a  bi-polar
grid GR1.5 and has 40 unevenly placed levels. It computes transport of tracers of the ocean
biogeochemistry model HAMOCC6 (Ilyina et al., 2013; Paulsen et al., 2017). The MPI-ESM-LR
configuration computes 45–85 model years per physical day enabling new simulations which
were not feasible previously, such as for instance, large ensemble simulations (Maher et al.,
2019) or millennial-scale simulations with interactive carbon cycle (Brovkin et al., 2019).

Terrestrial vegetation in JSBACH includes vegetation dynamics which interacts with land use
changes (Reick et al., 2013), accounting for the latest changes in the land use harmonization



dataset by Hurtt et al. (2006). The new SPITFIRE model simulates burned area and carbon
emissions to atmosphere due to wildfires and anthropogenic fires (Lasslop et al., 2014), replacing
old global fire parameterization used in the CMIP5 model. Soil carbon model YASSO simulates
dynamics of 4 fast soil carbon pools which are different for leaf and woody litter types, plus a
slow humus pool (Goll et al., 2015). Nitrogen and carbon pools are coupled based on CO2-induced
nitrogen limitation (Goll et al., 2017).

The ocean biogeochemistry model HAMOCC6 has been extended as compared to the previous
version described in Ilyina et al. (2013) to explicitly resolve nitrogen-fixing cyanobacteria as an
additional prognostic phytoplankton class (Paulsen et al., 2017). This allows to capture the
response of $N_2$ fixation and ocean biogeochemistry to changing climate conditions. Additionally,
updates of existing processes have been performed. This includes for instance the addition of a
vertically varying settling rate for detritus following the formulation by Martin et al. (1987).
Finally some empirical relationships in the parameterized processes have been updated to follow
recommendations of the C4MIP and OMIP protocols (Jones et al., 2016; Orr et al., 2017). The full
overview of changes in HAMOCC is given in Mauritsen et al. (2019).

**A5.9. Geophysical Fluid Dynamics Laboratory (GFDL) NOAA-GFDL-ESM4**



GFDL-ESM4.1 is a comprehensive, fully-coupled Earth System Model developed by NOAA's
Geophysical Dynamics Laboratory with a fully-interactive carbon cycle and interactive
atmospheric chemistry (Dunne et al., 2019, in prep., The GFDL Earth System Model version 4.1
(GFDL-ESM4.1): Model description and simulation characteristics) that builds on previous
generation modeling efforts of the carbon cycle (ESM2-series) (Dunne et al., 2012, 2013) and
atmospheric chemistry (CM3) (Donner et al., 2011) along with increased resolution and improved
numerics and physics akin to GFDL's 4th generation coupled climate model (CM4.0; Held et al.,
2019, in preparation), and representation of additional Earth System Processes.

The atmospheric component, GFDL AM4.1, is based on the third generation finite volume cube-
sphere dynamical core (FV3) (Lin, 2004) with a 1° horizontal resolution and 49 vertical levels. The
model top is located at ~0.1 hPa to resolve the stratosphere. AM4.1 shares the critical
developments in model physics with the AM4.0 model (Zhao et al., 2018) including radiation,
convection, and clouds. AM4.1 differs from the AM4.0 model in its enhanced vertical resolution
and its more explicit representation of atmospheric chemistry that motivated a separate
radiative and gravity wave tuning.

AM4.1 includes interactive tropospheric and stratospheric gas-phase and aerosol chemistry
represented through 56 prognostic (transported) tracers and 36 diagnostic (non-transported)
chemical tracers. The tropospheric chemistry includes reactions for the oxidation of methane
among other volatile organic compounds. The stratospheric chemistry accounts for the major



ozone loss cycles and heterogeneous reactions on liquid and solid stratospheric aerosols. Details
on the base chemical mechanism including improvements relative to the previous generation
model (AM3) are included in Horowitz et al. (2019, in prep).

Land hydrology and ecosystem dynamics are represented by the GFDL Land Model version 4.1
(LM4p1; Shevliakova et al., 2019, in prep) and builds on the previous generation LM3.1 model
(Milly et al., 2014). Soil carbon dynamics and biogeochemistry represented through the CORPSE
model (Sulman et al., 2019) with an explicit treatment of soil microbes. LM4.1 also includes a
new fire model FINAL (Rabin et al., 2018). Vegetation dynamics represented by the second
generation age-height structured approach the Perfect Plasticity Approximation (PPA) (Weng et
al., 2015, Martinez Cano et al., 2019, in prep). There are 6 carbon pools in LM4.1 representing
leaves, fine roots, heartwood, sapwood, seeds, and non-structural carbon (i.e. sugars). Litter is
broken into leaf and coarse wood categories as well into fast and slow timescale partitions. Soil
has 20 vertical levels each with its own prognostic state for energy, water and soil carbon
variables. There are 5 types of vegetation forms in LM4.1 representing $C_3$ grass, $C_4$ grass, tropical
trees, temperate deciduous trees, cold evergreen trees. A combination of these vegetation types
could coexist in some location. The model also includes a new treatment of stomatal conductance
and plant hydraulics. The vegetation state is used to drive a dust emission model that is coupled
with the atmosphere for transport (Ginoux et al., 2019, in prep.). ESM4 implementation of LM4.1
does not include an interactive nitrogen cycle.





The ocean biogeochemical component of ESM4 is version 2 of the Carbon, Ocean
Biogeochemistry and Lower Trophics (COBALTv2) model (Stock et al., 2014b). COBALTv2 uses 33
tracers to represent carbon, alkalinity, oxygen, nitrogen, phosphorus, iron, silica, calcite and
lithogenic mineral cycling within the ocean.    Relative to previous generation ocean
biogeochemistry models developed at GFDL, COBALTv2 includes an enhanced representation of
plankton food web dynamics to resolve the flow of energy from phytoplankton to fish (Stock et
al., 2014a) and enhance the model's capacity to resolve linkages between food webs and
biogeochemical cycles.  COBALTv2 explicitly includes small, large (split into diatoms and non-
diatoms), and diazotrophic phytoplankton groups, three zooplankton groups, bacteria and three
labilities of dissolved organic matter. Other updates include a temperature-dependence to
sinking organic matter remineralization (Laufkötter et al., 2017), the addition of semi-labile
dissolved organic material, carbonate chemistry calculations based on the open source Model of
the Ocean Carbonate SYstem version 2.0 (Orr and Epitalon, 2015).

Data from the NOAA-GFDL-ESM4 model used in the analysis presented in this paper are
accessible via the Earth System Grid Federation (ESGF) for 1pctCO2 (Krasting et al., 2019b)
simulation and for its radiatively- and biogeochemically-coupled configurations (Krasting et al.,
2019a).

**A5.10. Norwegian Climate Centre (NCC) NorESM2-LM**



The NorESM2-LM is based on the latest release of the Community Earth System Model
(CESM2.1), whose development is supported by the National Center for Atmospheric Research
at the United States. NorESM2 keeps the original land and sea-ice components of CESM2.1 (i.e.,
CLM5, and CICE5, respectively). The atmospheric component is CAM6 (as in CESM), but with
modifications regarding the energy and angular momentum conservation. Further, the
atmospheric chemistry module of CAM6 has been replaced by the scheme developed by the
Norwegian Meteorological Institute. The ocean physical and biogeochemical components of
NorESM2 are the  isopycnal ocean circulation and carbon cycle components updated from
NorESM1 version (Schwinger et al., 2016; Tjiputra et al., 2013)

The CLM5 (Community Land Model version 5) prognostically simulates the carbon and nitrogen
cycles, which include natural vegetation, crops, and soil biogeochemistry. The carbon and
nitrogen budgets comprise leaf, live stem, dead stem, live coarse root, dead coarse root, fine-
root, and grain pools. Each of these pools has short-term and long-term storage of non-structural
carbohydrates and labile nitrogen. In addition to the vegetation pools, CLM includes a series of
decomposing carbon and nitrogen pools as vegetation successively breaks down to coarse woody
debris, and/or litter, and subsequently to soil organic matter. Details on the CLM5 models are
available in Lawrence et al. (2018).

Similar to the earlier version, the ocean carbon cycle component in NorESM2 is based on the
Hamburg Oceanic Carbon Cycle (HAMOCC; Maier-Reimer et al., 2005) model, which has been



adopted to the isopycnic ocean general circulation model. The current version includes new
processes, refined parameterizations, as well as new diagnostic tracers. The ecosystem model is
based on an NPZD-type model with multi nutrient limitation in its phytoplankton growth
formulation. Riverine fluxes of inorganic and organic carbon as well as nutrients are now
implemented. Unlike the earlier version, the sea-to-air dimethyl sulfate (DMS) fluxes alter the
atmospheric radiative forcing and hence the climate carbon cycle feedback. More details on the
ocean carbon cycle of NorESM2 are available in Tjiputraet al. (2019, in preparation).

**A5.11. The United Kingdom Community Earth System Model, UKESM1-0-LL**
UKESM1-0-LL (Sellar et al., 2019) is based upon the HadGEM3-GC3.1 (Williams et al., 2018) global
climate model which includes coupled ocean, atmosphere, land and sea-ice components. The
atmosphere component is the Unified Model with a resolution of 1.875˚ by 1.25˚ with 85 vertical
levels up to a model top of 90 km (Walters et al., 2019) and includes a modal aerosol scheme
(Mann et al., 2010). The ocean component uses the NEMO dynamical ocean at 1˚ resolution with
75 vertical levels (Storkey et al., 2018). The sea-ice component uses CICE on the same grid as the
ocean with 5-ice thickness categories (Ridley et al., 2018). The land component uses the JULES
land surface model (Wiltshire et al., in preparation), however, the land surface configuration is
substantially updated for UKESM. The primary differences between the physical and earth system
models is the inclusion of a terrestrial carbon and nitrogen cycle (Wiltshire et al., in preparation),
ocean biogeochemistry (Yool et al., 2013) and tropospheric-stratospheric chemistry model.
Atmospheric chemistry in UKESM1 is simulated by the UKCA chemistry and aerosol model with



the specific configuration a combination of tropospheric (O'Connor et al., 2014) and
stratospsheric chemistry (Morgenstern et al., 2009, 2017).

Terrestrial biogeochemistry is represented by the JULES-ES model cycle (Wiltshire et al., in
preparation). The land surface is represented by 13 plant functional types (PFTs) including 4
managed crop and pasture land types. The height, leaf area index and spatial distribution of the
PFTs are dynamic simulated by TRIFFID dynamic global vegetation model (Cox, 2001). Soil carbon
is represented by the 4 pool Roth-C scheme (Coleman and Jenkinson, 1999). Terrestrial carbon
uptake may be limited by the availability of nitrogen. Nitrogen does not directly affect
photosynthetic capacity through leaf N concentrations but acts indirectly by controlling the
biomass and leaf area index within the TRIFFID DGVM. A second mechanism acts through soil
carbon by limiting the decomposition of litter into soil carbon in the RothC model. The vegetation
model includes retranslocation of Nitrogen during senescence of leaves and roots into a labile
pool to supply nutrients for the following seasonal leaf out. The soil model simulates
mineralisation and immobilisation with mineralised nitrogen becoming available for plant uptake
and ecosystem loss. Inorganic Nitrogen is represented by a single gridbox pool from which all
PFTs have equal access. Nitrogen deposition is prescribed from ancillary data.

Land-use change is represented by the application of time-varying fields of crop and pasture to
the DGVM, which allocates space dynamically to $C_3$ and $C_4$, crop and pasture types. Pasture is
represented as natural grass whereas crops include a harvest parameterization and are fertilized.



Biogenic Volatile Organic Compound (BVOC) emissions from vegetation are simulated and affect
the formation of secondary organic aerosols. Mineral dust is emitted from bare soil and acts as
both an aerosol and a fertiliser to the ocean.

Ocean biogeochemistry is represented by MEDUSA-2 (Yool et al., 2013) which resolves a dual
size-structured ecosystem of small (nanophytoplankton and microzooplankton) and large
(microphytoplankton and mesozooplankton) components. This explicitly includes the
biogeochemical cycles of nitrogen, silicon and iron nutrients as well as the cycles of carbon,
alkalinity and dissolved oxygen. Large phytoplankton are treated as diatoms and utilise silicic acid
in addition to nitrogen, iron and carbon. Like the living components, the detrital components are
split into two size classes. At the seafloor, MEDUSA-2 resolves 5 reservoirs to temporarily store
sinking organic material reaching the sediment. The model's nitrogen, silicon and alkalinity cycles
are closed and conservative (e.g. no riverine inputs), while the other three cycles (carbon, iron,
oxygen) are open. The ocean's iron cycle includes aeolian (land derived dust) and benthic sources,
and is depleted by scavenging. The ocean's carbon cycle exchanges CO2 with the atmosphere.
The ocean's oxygen cycle exchanges with the atmosphere, and dissolved oxygen is additionally
created by primary production and depleted by remineralisation. Ocean biogeochemistry also
feeds back on the atmosphere through the production of marine DMS and marine organic
aerosols.

**A6. Contribution of uncertainties in $\Delta T_{2\times CO2}$ and $\tilde{E}_{2\times CO2}$ to TCRE.**






The uncertainty in TCRE, as indicated by its standard deviation ($\sigma_{TCRE}$), can be represented in
terms of the standard deviation of $\Delta T_{2 \times CO2}$ ($\sigma_{\Delta T}$), standard deviation of $\tilde{E}_{2 \times CO2}$ ($\sigma_E$), and their
means $\overline{\Delta T}$ and $\overline{E}$ across the eleven CMIP6 models. Since $\Delta T_{2 \times CO2}$ and $\tilde{E}_{2 \times CO2}$ are nearly
independent (correlation between these two quantities is only 0.02 across the eleven CMIP6
models considered here), we can write
$$\sigma_{TCRE} = \overline{TCRE} \cdot \sqrt{\left(\frac{\sigma_{\Delta T}}{\overline{\Delta T}}\right)^2 + \left(\frac{\sigma_E}{\overline{E}}\right)^2}$$
(A8)

which allows to calculate to contributions of $\left(\frac{\sigma_{\Delta T}}{\overline{\Delta T}}\right)^2$ and $\left(\frac{\sigma_E}{\overline{E}}\right)^2$ to $\sigma_{TCRE}$.













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
