# Peer review of "Carbon-concentration and carbon-climate feedbacks in CMIP6 models, and their comparison to CMIP5 models"

_Biogeosciences, 2019_

## Referee Comment (RC1) · Anonymous Referee #1 · 30 Dec 2019

I did not find any obvious errors or omissions in the manuscript, but I believe that the abstract should report the mean values of the concentration-carbon and carbon-climate feedback parameters for land and for the ocean calculated using the fully-coupled configuration.

Besides, I would recommend to revise the first abstract of the 'Introduction' if the authors would have some spare time. I think it would be better to replace it with a brief historical overview of the climate feedbacks problem citing some seminal work on this subject.

---

## Referee Comment (RC2) · Kirsten Zickfeld (Referee) · 11 Feb 2020

Review of "Carbon-concentration and carbon-climate feedbacks in CMIP6 models, and their comparison to CMIP5 models" by Arora et al.

General comments:

This manuscript provides an update on carbon cycle feedback parameters in Earth system models based on simulations conducted under the CMIP6 framework. This is a useful manuscript that demonstrates how feedback metrics have evolved from CMIP5 to CMIP6 models and provides explanations for these differences. However, the manuscript is very long, and inconsistent in its quality. A few sections are merely descriptive and do not offer much insight into the reasons for inter-model differences, whereas other parts are missing important details (see specific comments). Also, I am not convinced of the usefulness for this manuscript of the analysis that decomposes feedback parameters into various terms (section 4.4). The decomposition has meaning to experts in the respective land and ocean biogeochemistry disciplines, but in my view has little value to readers not familiar with the concepts of one or either discipline. I suggest that the authors consider removing these analyses from the manuscript and preparing separate disciplinary submissions. This would free room to strengthen other aspects of the manuscript. For instance, the abstract concludes that the approach of calculating the feedback parameters from COU and BGC simulations is "most relevant", yet there is no discussion of this point in the main manuscript. Also, an interesting result is that despite the fact that several models now include N limitation, the model-mean land carbon uptake has increased relative to CMIP5. This is an important result, which in my view warrants more attention in the results and conclusion sections.

Specific comments:

Abstract, l. 48-49: How are they different?

Abstract, l. 50-51: This conclusion is not supported by the discussion in the manuscript. There is discussion on this point in the Appendix that should be elevated to the main manuscript if this point is to be kept in the abstract.

Abstract, l. 51-55: Report interesting findings rather than methodological approaches.

p. 4, l. 97: "Offers several benefits": which specifically?

p. 4, l. 116: "Comparison is useful". I don't think comparison between A13 and F06 is particularly useful. The two studies differ with regard to a number of key assumptions that make resulting feedback metrics hard to compare. In addition to different scenarios these include emissions versus concentration-driven runs and different methods to compute gamma.

p. 6, l. 142: Section 2 title: I suggest to rename the title to "Feedbacks *metrics* in the coupled climate-carbon system". The current title suggests a section on processes, which it is not.

p. 8, l. 183: Even though c' does not appear in Eq. 1a I suggest to mention that c' is the same for RAD as well.

p. 9, l. 192-193: Unclear what is meant by "evolve over time" (from one year to another or from CMIP5 to CMIP6?).

p. 10, l. 225: Include reference to Zickfeld et al. (2011). The paper provides a detailed analysis of non-linearities in the coupled climate-carbon cycle system.

p. 11, l. 234-235: First part of the sentence is repetitive.

p. 11, l. 237: How is it different, i.e. is it larger or smaller?

p. 13, l. 276-277, "explicitly considering...": contradictory. Needs clarification.

p. 17, l. 361-362: Need to say that changes in biological carbon inventory are assumed to be small.

p. 19, l. 407: How is the function f defined?

p. 20, l. 415-416, "Do slightly affect": Can this be quantified?

p. 21, l. 450-452. This statement is confusing. After reading it I thought that the additional figure panels show CMIP6 results for the subset of models that was used in CMIP5, but from the figure captions I gather that those panels show the CMIP5 results from A13. Please clarify.

p. 22, L. 470-472: Mention that the CMIP6 model ensemble includes some high climate sensitivity models.

p. 23, l. 481-482: "fitting a polynomial"? Justify why you chose to do this. Fitting procedure needs to be described in the Methods section.

p. 23, l. 489-490: It is not intuitive why temperature in the RAD simulation is sensitive to inclusion of NorESM2-LM whereas land and ocean carbon fluxes are not. Please explain.

p. 24, l. 512, "has not meaningfully declined". The bottom panels in Fig. 2 suggest that it has actually increased.

p. 25, l. 523-524: How about changes in ocean circulation?

p. 25, l. 526-540: I don't find this pargarph particularly useful as it merely describes what is evident in the figure. I suggest to either include an explanation for inter-model differences, or delete the paragraph. The figure could then be moved to the Appendix.

p. 26, l. 543: Which simulation – RAD? The factor is lower (~two) for the COU and BGC simulations.

p. 26, l. 552-553: Need to clarify that DC' refers to a change in a reservoir. As such, DC'_A is not the atmospheric growth rate (PgC/yr) but the change in atmospheric carbon burden (PgC).

p. 26, l. 555: Which equation/section of the Appendix? Could also refer to Eq. (18).

p. 27, l. 564: Which equation/section of the Appendix?

p. 28, l. 588: It should be emphasized that the difference between models with and without representation of the N cycle is much smaller than in CMIP5.

p. 28, l. 595-596: It would he helpful to have a brief explanation of the increase in $CO_2$ fertilization effect in CanESM5.

p. 28, l. 602-603: I suggest to remove quantitative information from this and the subsequent paragraph (not needed).

p. 29, l. 611 – p. 30 l. 628: The discussion would be easier to follow if differences in land models were first discussed, followed by a discussion of differences in ocean models.

p. 28, l. 592 – p. 30 l. 628: It would be worth emphasizing (here and in the conclusions) that with implementation of N limitation in several models land carbon uptake increased in CMIP6 relative to CMIP5.

p. 30, l. 641-646: Avoid repeating information from the figure legend in the text.

p. 31, l. 650-652: This is not immediately evident from the figure (e.g. beta_L calculated with RAD-COU differs from that calculated with other approaches for CMIP6 models). What measure was used to quantify the sensitivity?

p. 31, l. 656-658: It could again be noted that difference between models with and without N limitation is smaller than for CMIP5.

p. 31, l. 663: It is worth mentioning in my view that the spread in feedback parameters for models with and without N cycle has widened compared to CMIP5.

p. 32, l. 680, "existing studies": Provide references.

p. 33, l. 703-707: Avoid repeating information from the figure legend in the text.

p. 32, l. 680-681: The preferred use of COU and BGC over other approaches to calculate the feedback parameters is a conclusion highlighted in the abstract, yet the discussion is relegated to the Appendix. If the conclusion is to be kept in the abstract, the text in A2 should be elevated to the main manuscript.

p. 34, l. 722-724: Suggest to delete quantitative information in parenthesis (not needed).

p. 37, l. 772-773, "This is one of the few times…": Include references.

p. 38, l. 798-801: Unclear why this needs to be stated upfront.

p. 46, l. 961: climate response to cumulative *carbon* emissions

p. 47, l. 977-978: Is the increase in the mean value of the TCRE since CMIP5 due to changes in TCR, diagnosed emissions or both?

p. 47, l. 986-987, "representation of the nitrogen cycle is helpful in reducing this uncertainty": unclear what results this statement is based on.

p. 47, l. 993-p. 48, l. 1003: Several studies have explored the decomposition of TCRE into various terms and their contribution to TCRE uncertainty. Given that a comprehensive discussion of this literature is out of scope here I suggest to delete this paragraph that is based on a single study.

p. 50, l. 1038, "… a reduced spread across land models". I don't think this is a correct characterization of the results. The CMIP6 models including N limitation have a smaller spread than the models without N limitation (Fig. 6) but the overall spread is not reduced compared to CMIP5.

p. 50, l. 1052-1055: Again, the manuscript lacks discussion supporting this conclusion.

p. 59, Fig. 5 caption: Equation references need to be corrected.

p. 62, Fig. 8: The upper panel is reproduced in Fig. 9, so this figure could be cut.

p. 65-66: Figs. 11 and 12 could be combined. Also, the figure caption needs to draw attention to the different vertical scale used in the panels.

Editorial comments

p. 3, l. 61-63: Style of sentence could be more fluid

p. 4, l. 83-84: Delete text in parenthesis

p. 4, l. 97: Delete "of course"

p. 15, l. 333: Delete "some".

p. 32, l. 680: Delete "and" after "7".

p. 34, l. 714: Sentence unclear.

p. 36, l. 748-749, "While …. Feedback over land ($\beta_L$),": Delete

p. 48, l. 1011-1014: Sentence is convoluted. Delete "that allows…."

p. 49, l. 1027: Replace "but" with "and".

p. 49, l. 1029: Delete "very".

---

## Author Comment (AC1) · 9 Mar 2020

**We would like to thank both reviewers for their helpful and useful comments on our manuscript. Our response to reviewers' comments are shown in bold below.**

Reviewer #1

I did not find any obvious errors or omissions in the manuscript, but I believe that the abstract should report the mean values of the carbon-concentration and carbon-climate feedback parameters for land and for the ocean calculated using the fully-coupled configuration.

[Figure]

**We agree with reviewer 1 and we will show mean values of the carbon-concentration and carbon-climate feedback parameters as well as TCRE for land and for the ocean in the abstract when revising our manuscript.**

Besides, I would recommend to revise the first abstract of the 'Introduction' if the authors would have some spare time. I think it would be better to replace it with a brief historical overview of the climate feedbacks problem citing some seminal work on this Subject.

**Thank you for this good suggestion. We will mention the primary feedbacks that operate in the climate system and put the carbon cycle related feedbacks in context of these primary climate system feedbacks in the first paragraph of the introductory section.**

Reviewer #2 (Dr. Kirsten Zickfeld)

General comments: This manuscript provides an update on carbon cycle feedback parameters in Earth system models based on simulations conducted under the CMIP6 framework. This is a useful manuscript that demonstrates how feedback metrics have evolved from CMIP5 to CMIP6 models and provides explanations for these differences. However, the manuscript is very long, and inconsistent in its quality. A few sections are merely descriptive and do not offer much insight into the reasons for inter-model differences, whereas other parts are missing important details (see specific comments). Also, I am not convinced of the usefulness for this manuscript of the analysis that decomposes feedback parameters into various terms (section 4.4). The decomposition has meaning to experts in the respective land and ocean biogeochemistry disciplines, but in my view has little value to readers not familiar with the concepts of one or either discipline. I suggest that the authors consider removing these analyses from the manuscript and preparing separate disciplinary submissions. This would free room to strengthen other aspects of the manuscript. For instance, the abstract concludes that the approach of calculating the feedback parameters from COU and BGC simulations

is "most relevant", yet there is no discussion of this point in the main manuscript. Also, an interesting result is that despite the fact that several models now include N limitation, the model-mean land carbon uptake has increased relative to CMIP5. This is an important result, which in my view warrants more attention in the results and conclusion sections.

**We thank Dr. Zickfeld for her thorough review. She makes two major comments. First, she recommends a shorter manuscript that reports only the carbon-concentration and carbon-climate feedbacks, and suggests that section 4.4 (which delves into the reasons for differences in feedback parameters among models) be removed and reworked as a separate manuscript. Second, that the conclusion that the use of COU and BGC experiments yields "most relevant" values of feedback parameters should be explained clearly in the main text.**

**While we completely agree with her second major comment and will revise our manuscript accordingly, we respectfully disagree with the suggestion that the manuscript should be split in two. Previous papers that summarized carbon-concentration and carbon-climate feedback parameters from ESMs (Friedling-stein et al., 2006; Arora et al., 2013) did not address the reasons for differences in feedback parameters across models. At a workshop in Bern in April 2018, about 25 members of both the land and ocean carbon cycle communities agreed that analyzing reasons for differences among models is desirable. This knowledge is expected to lead to model improvements, and in the long term perhaps a stronger consensus about the most appropriate approaches. This is especially true for land carbon cycle models given the large spread. The question of removing Section 4.4 was put again to all of the co-authors, and there is a strong consensus that Section 4.4 provides a novel analysis that is valuable to the present manuscript, especially given that the underlying framework for assessing differences among models is the same for land and ocean. In addition, providing results from land and ocean in a single manuscript in an integrated**

manner provides an opportunity for the land carbon cycle community to learn about the ocean carbon cycle and vice versa, and hopefully contributes to dialogue across the the boundaries that have traditionally separated the two communities. Despite its somewhat longer page count, which we will try hard to reduce while revising our manuscript, we feel that the intellectual and scientific value of the additional analysis justifies the length.

Specific comments:

**In the following lines, we make note of how we will address these specific comments when revising our manuscript.**

Abstract, l. 48-49: How are they different?

**We will mention the values of feedback parameters in the abstract.**

Abstract, l. 50-51: This conclusion is not supported by the discussion in the manuscript. There is discussion on this point in the Appendix that should be elevated to the main manuscript if this point is to be kept in the abstract.

**Agreed. We will enhance the discussion about higher relevance of feedback parameters values calculated using COU and BGC configurations.**

Abstract, l. 51-55: Report interesting findings rather than methodological approaches.

**Agreed. We will revise the abstract accordingly.**

p. 4, l. 97: "Offers several benefits": which specifically?

**Organized MIPs offer benefits of common standards and experiment protocol, coordination, infrastructure, and documentation that facilitates the distribution of model outputs and the characterization of the mean-model response (Eyring et al., 2016). We will revise this sentence.**

p. 4, l. 116: "Comparison is useful". I don't think comparison between A13 and F06 is

particularly useful. The two studies differ with regard to a number of key assumptions that make resulting feedback metrics hard to compare. In addition to different scenarios these include emissions versus concentration-driven runs and different methods to compute gamma.

**We will remove the word useful and reword this sentence.**

p. 6, l. 142: Section 2 title: I suggest to rename the title to "Feedbacks metrics in the coupled climate-carbon system". The current title suggests a section on processes, which it is not.

**Thank you for this suggestion.**

p. 8, l. 183: Even though c' does not appear in Eq. 1a I suggest to mention that c' is the same for RAD as well.

**Actually this is already mentioned on line 185.**

p. 9, l. 192-193: Unclear what is meant by "evolve over time" (from one year to another or from CMIP5 to CMIP6?).

**This sentence implies the evolution of parameters over the duration of the experiment. We will revise this to make it more explicit.**

p. 10, l. 225: Include reference to Zickfeld et al. (2011). The paper provides a detailed analysis of non-linearities in the coupled climate-carbon cycle system.

**Agreed.**

p. 11, l. 234-235: First part of the sentence is repetitive.

**We will reword this sentence.**

p. 11, l. 237: How is it different, i.e. is it larger or smaller?

**The gamma (carbon-climate feedback) values may be higher or lower for land, but for ocean it seems the gamma values are always higher when calculated**

**using the COU-BGC approach. We will revise this sentence.**

p. 13, l. 276-277, "explicitly considering. . .": contradictory. Needs clarification.

**We will reword this sentence.**

p. 17, l. 361-362: Need to say that changes in biological carbon inventory are assumed to be small.

**Agreed.**

p. 19, l. 407: How is the function f defined?

**The function in equation (16) represents the familiar solution of the ocean carbon system from two of the "four pillars" (DIC, total alkalinity, pH, and pCO2), in this case with total alkalinity and pCO2 on the RHS and DIC the unknown for which we are iterative solving. To explain, we assume an initial guess for H+ and use it to remove the effect of minor species (e.g., borate) from the total alkalinity in order to get the carbonate alkalinity. Then from the carbonate alkalinity and the pCO2 we estimate a new H+. This process is repeated for a number of iterations until the solution reaches a convergence. This new H+ and pCO2 is used to estimate the carbonate speciation.**

**We are using the preformed alkalinity instead of the instantaneous total alkalinity, and a surface ocean pCO2 in equilibrium with the atmosphere, to estimate the preformed DIC for water masses formed in earlier times when atmospheric CO2 was lower. All of the quantities in these equations are defined in the text. We will rework the wording to make sure that the meaning of each is clear and unambiguous.**

**In the revised manuscript we explicitly state that this function corresponds to the iterative algorithm of Follows et al. 2006, but prefer not to go into detail in the paper since this function is well documented in the work of Follows et al. 2006.**
p. 20, l. 415-416, "Do slightly affect": Can this be quantified?

**Our diagnostics of the ocean feedbacks and carbon pools depend primarily upon changes in DIC, the preformed and regenerated pools, relative to the pre indus-trial. There are inter-model differences in the pre-industrial ocean that slightly affect the changes in saturated DIC. Differences in the pre-industrial saturated part of DIC as shown in Figures S1a to S8a, second column for all models and Figs. 11a and 12a for UKESM.**

**For example, for a doubling in pCO2 (no climate change) for a pre-industrial state with**

**pCO2=280 ppm,**
**To=5 deg. C,**
**So=34.5 PSU,**
**P= 2 $\mu$mol/kg,**
**Si= 80 $\mu$mol/kg,**
**Alkpre= 2300 $\mu$mol/kg,**

**the resulting change in $\triangle$DICsat**
**=DICsat(2xpCO2)-DICsat(to)**
**=(2199-2093) $\mu$mol/kg**
**= 106 $\mu$mol/kg.**

**If instead the ocean at pre-industrial is warmer by 1deg. C then the change in $\triangle$DICsat**
**=DICsat(2xpCO2)-DICsat(to)**
**=(2193-2085) $\mu$mol/kg**
**= 108 $\mu$mol/kg (i.e.1.9% difference).**

**If instead the ocean at pre-industrial has higher Alkpre by 50 $\mu$mol/kg then the**

[Figure]

**change in △DICsat**
**= △DICsat =DICsat(2xpCO2)-DICsat(to)**
**=(2245-2136) $\mu$mol/kg**
**= 109 $\mu$mol/kg (i.e. 2.8% difference)."**

p. 21, l. 450-452. This statement is confusing. After reading it I thought that the additional figure panels show CMIP6 results for the subset of models that was used in CMIP5, but from the figure captions I gather that those panels show the CMIP5 results from A13. Please clarify.

**Thank you. We will clarify this statement to make it explicitly clear that the model results are shown for CMIP6 and CMIP5 models that participated in Arora et al. (2013) and this study, respectively, and not any subset.**

p. 22, L. 470-472: Mention that the CMIP6 model ensemble includes some high climate sensitivity models.

**Agreed.**

p. 23, l. 481-482: "fitting a polynomial"? Justify why you chose to do this. Fitting procedure needs to be described in the Methods section.

**This is best illustrated by considering temperature change. The mean model temperature change, although averaged over a number of models, doesn't increase monotonously with CO2 because of inter-annual variability. So the mean-model value at the end of simulation may be higher or lower, than if it were increasing monotonously, depending on the values from individual models. Fitting a polynomial to mean-model values yields a more reliable estimate of warming at the end of the simulation. We will expand the text around this to justify our reason for fitting a polynomial and also describe the approach.**

**For calculation of TCRE we will follow the more standard approach that calculates TCR using 20 annual values of temperature centered on the year when CO2**

**doubles.**

p. 23, l. 489-490: It is not intuitive why temperature in the RAD simulation is sensitive to inclusion of NorESM2-LM whereas land and ocean carbon fluxes are not. Please explain.

**We now have results from the RAD configuration of the 1pctCO2 experiment from the NorESM2-LM model so this issue will become moot.**

p. 24, l. 512, "has not meaningfully declined". The bottom panels in Fig. 2 suggest that it has actually increased.

**Yes, true indeed. We will reword this sentence.**

p. 25, l. 523-524: How about changes in ocean circulation?

**In the RAD simulation, the overall loss of carbon is caused mainly by a depletion of upper ocean DIC, which is primarily due to surface warming (70-86% for CMIP5 models) and to a second-degree due to changes in alkalinity (10-28% for CMIP5 models) (Schwinger et al., 2014). The changes in the circulation/stratification in the RAD also increase the isolation of deep water, which lead either to a near neutral or a small positive increase in the storage of DIC in deep waters (in this RAD run where there is no increase in pCO2).**

p. 25, l. 526-540: I don't find this pargraph particularly useful as it merely describes what is evident in the figure. I suggest to either include an explanation for intermodel differences, or delete the paragraph. The figure could then be moved to the Appendix.

**Agreed. This figure and the associated discussion is an ideal candidate for moving it to the Appendix to reduce the overall length of the main text.**

p. 26, l. 543: Which simulation – RAD? The factor is lower (about two) for the COU and BGC simulations.

**In fact this sentence refers to the COU simulation. We will clarify this.**

p. 26, l. 552-553: Need to clarify that $\Delta$C' refers to a change in a reservoir. As such, $\Delta$C'$_A$

is not the atmospheric growth rate (PgC/yr) but the change in atmospheric carbon burden (PgC).

**Yes, this is correct. We will make this change.**

p. 26, l. 555: Which equation/section of the Appendix? Could also refer to Eq. (18).

**Equation A6 in the appendix.**

p. 27, l. 564: Which equation/section of the Appendix?

**This refers to section A1 of the appendix.**

p. 28, l. 588: It should be emphasized that the difference between models with and without representation of the N cycle is much smaller than in CMIP5.

**Actually this is kind of a red herring since there was only essentially one land model with N cycle (CLM4, implemented in CESM1-BGC and NorESM) in the Arora et al., 2013 study.**

p. 28, l. 595-596: It would he helpful to have a brief explanation of the increase in CO2 fertilization effect in CanESM5.

**Agreed. We will expand on this sentence.**

p. 28, l. 602-603: I suggest to remove quantitative information from this and the subsequent paragraph (not needed).

**Agreed.**

p. 29, l. 611 – p. 30 l. 628: The discussion would be easier to follow if differences in land models were first discussed, followed by a discussion of differences in ocean Models.

**Agreed.**

p. 28, l. 592 – p. 30 l. 628: It would be worth emphasizing (here and in the conclusions) that with implementation of N limitation in several models land carbon uptake increased in CMIP6 relative to CMIP5.

**Actually, the subtlety here is that the mean land carbon uptake in CMIP6 models has increased due to models which do not include N cycle. We will convey this message clearly.**

p. 30, l. 641-646: Avoid repeating information from the figure legend in the text.

**Agreed.**

p. 31, l. 650-652: This is not immediately evident from the figure (e.g. $beta_L$

calculated with RAD-COU differs from that calculated with other approaches for CMIP6 models). What measure was used to quantify the sensitivity?

**The absolute mean $beta_L$**

**values in Figure 6b calculated using the RAD-COU approach (-55.1 Pg C/degree C) are higher than that calculated using the BGC-COU approach (-45.1 Pg C/degree C) by 22%. We will make this clear when revising our manuscript.**

p. 31, l. 656-658: It could again be noted that difference between models with and without N limitation is smaller than for CMIP5.

**As mentioned above, since there was only essentially one land model with N cycle (CLM4, implemented in CESM1-BGC and NorESM) in the Arora et al., 2013 study it is difficult to draw such a conclusion.**

p. 31, l. 663: It is worth mentioning in my view that the spread in feedback parameters for models with and without N cycle has widened compared to CMIP5.

**We are not sure what its being implied here. But the fact that there was only**

**one land model in Arora et al. (2013) CMIP5 study with N cycle means no sound conclusions can be drawn here.**

p. 32, l. 680, "existing studies": Provide references.

**We will provide reference to ocean carbon cycle studies here.**

p. 33, l. 703-707: Avoid repeating information from the figure legend in the text.

**Agreed.**

p. 32, l. 680-681: The preferred use of COU and BGC over other approaches to calculate the feedback parameters is a conclusion highlighted in the abstract, yet the discussion is relegated to the Appendix. If the conclusion is to be kept in the abstract, the text in A2 should be elevated to the main manuscript.

**We agree and we will make this discussion part of the main text.**

p. 34, l. 722-724: Suggest to delete quantitative information in parenthesis (not needed).

**Agreed.**

p. 37, l. 772-773, "This is one of the few times...": Include references.

**We will include this reference.**

p. 38, l. 798-801: Unclear why this needs to be stated upfront.

**This statement clarifies the dominant control by the air-sea exchange, rather than the response being controlled by land-ocean exchanges (as explained in the second sentence). This issue is expanded upon in the Appendix in Table A2.**

p. 46, l. 961: climate response to cumulative carbon emissions

**Thank you. We will include the word "carbon".**

p. 47, l. 977-978: Is the increase in the mean value of the TCRE since CMIP5 due to changes in TCR, diagnosed emissions or both?

**We have not analyzed this aspect. However, since we are not aware of any other recent paper documenting and analyzing TCRE in detail from CMIP6 models we have decided to mention TCRE numbers in the abstract as well. In this respect it would make sense to analyze and report this aspect too (i.e. the reason for increase in TCRE).**

**We will also cite Jones Friedlingstein (in revision for an ERL paper) who document TCRE and component uncertainty terms. Given the small and uneven sample of ESMs available, we cannot say that TCRE has changed significantly since CMIP5, although the contribution to its uncertainty has moved towards a greater component from climate feedbacks than carbon cycle - this is primarily due to reduced spread in land-beta in CMIP6.**

p. 47, l. 986-987, "representation of the nitrogen cycle is helpful in reducing this uncertainty": unclear what results this statement is based on.

**What is meant here is that the spread across land models is smaller for models with N cycle. We will clarify this when revising our manuscript.**

p. 47, l. 993-p. 48, l. 1003: Several studies have explored the decomposition of TCRE into various terms and their contribution to TCRE uncertainty. Given that a comprehensive discussion of this literature is out of scope here I suggest to delete this paragraph that is based on a single study.

**Although a single study, the cited work corroborates our conclusion that the inter-model spread in the TCRE are dominated by the thermal/temperature part. It also identifies that after 80 years carbon part of the TCRE becomes equally important. We feel this is important and needs to be retained.**

p. 50, l. 1038, "... a reduced spread across land models". I don't think this is a

correct characterization of the results. The CMIP6 models including N limitation have a smaller spread than the models without N limitation (Fig. 6) but the overall spread is not reduced compared to CMIP5.

**This is what we meant and your comment indicates that we need to reword our sentence. We are implying that if all land models had N cycle implemented the spread between the models will be reduced.**

p. 50, l. 1052-1055: Again, the manuscript lacks discussion supporting this Conclusion.

**As mentioned above, we will move text from appendix to the main manuscript to support this conclusion and bring out the message (that COU and BGC configurations providing the most relevant parameter values) more clearly.**

p. 59, Fig. 5 caption: Equation references need to be corrected.

**Thank you for pointing this.**

p. 62, Fig. 8: The upper panel is reproduced in Fig. 9, so this figure could be cut.

**The reason for reproducing Figure 8a in the upper panel of Figure 9 is for easy correspondence between models and the reasons for their behaviour as described by the decomposition terms.**

p. 65-66: Figs. 11 and 12 could be combined. Also, the figure caption needs to draw attention to the different vertical scale used in the panels.

**Both Figure 11 and 12 already have 16 panels each. Combining them will likely make their axes labels and other text completely unreadable. We will make note of the different colour scales used in the panels.**

**We will also include other minor editorial suggestions made.**

**References**

Arora, V. K., Boer, G. J., Friedlingstein, P., Eby, M., Jones, C. D., Christian, J. R., Bonan, G., Bopp, L., Brovkin, V., Cadule, P., Hajima, T., Ilyina, T., Lindsay, K., Tjiputra, J. F. and Wu, T.: Carbon–Concentration and Carbon–Climate Feedbacks in CMIP5 Earth System Models, J. Clim., 26(15), 5289–5314, doi:10.1175/JCLI-D-12-00494.1, 2013

Eyring, V., Bony, S., Meehl, G. A., Senior, C. A., Stevens, B., Stouffer, R. J., and Taylor, K. E.: Overview of the Coupled Model Intercomparison Project Phase 6 (CMIP6) experimental design and organization, Geosci. Model Dev., 9, 1937–1958, https://doi.org/10.5194/gmd-9-1937-2016, 2016.

Follows, M. J., Ito, T. and Dutkiewicz, S.: On the solution of the carbonate chemistry system in ocean biogeochemistry models, Ocean Model., 12(3), 290–301, doi:https://doi.org/10.1016/j.ocemod.2005.05.004, 2006.

Friedlingstein, P., Cox, P., Betts, R., Bopp, L., von Bloh, W., Brovkin, V., Cadule, P., Doney, S., Eby, M., Fung, I., Bala, G., John, J., Jones, C., Joos, F., Kato, T., Kawamiya, M., Knorr, W., Lindsay, K., Matthews, H. D., Raddatz, T., Rayner, P., Reick, C., Roeckner, E., Schnitzler, K.-G., Schnur, R., Strassmann, K., Weaver, A. J., Yoshikawa, C. and Zeng, N.: Climate–Carbon Cycle Feedback Analysis: Results from the C4MIP Model Intercomparison, J. Clim., 19(14), 3337–3353, doi:10.1175/JCLI3800.1, 2006.

Schwinger, J., Tjiputra, J. F., Heinze, C., Bopp, L., Christian, J. R., Gehlen, M., Ilyina, T., Jones, C. D., Salas-Mélia, D., Segschneider, J., Séférian, R. and Totterdell, I.: Non-linearity of Ocean Carbon Cycle Feedbacks in CMIP5 Earth System Models, J. Clim., 27(11), 3869–3888, doi:10.1175/JCLI-D-13-00452.1, 2014.

---

## Author Response (AR1)

2 May, 2020

Dear Editor,

We have incorporated all of reviewers' comments in revising our manuscript as explained in our
response to reviewers' comments in the Interactive Discussion.

The remaining pages of this document show the changes we have made using the track option in
Word since our last submission.

In particular,

1)  we have moved text from the Appendix to the main document to support our case that
  the BGC and COU configurations are the preferred simulations to diagnose the feedback
  parameters,
2)  modified our abstract substantially to report the values of the feedback parameters,
3)  moved the figure showing individual model results from the main text to the appendix,
  and
4)  made the analysis related to transient climate response to cumulative emissions a bit
  more prominent.

Best regards,

Vivek Arora (on behalf of all co-authors).

[revised manuscript text omitted]

**Commented [A[3]:** Ric/Anna is it okay to say "primarily" since changes in ocean circulation do not contribute to loss of C from ocean in the RAD simulation. This is in response to Kirsten's question.

¶

[revised manuscript text omitted]

Font: 11 pt, Superscript

| Page 2: [2] Formatted | Arora,Vivek [CCCMA] | 21/04/2020 2:42:00 PM |
|---|---|---|

Font: 11 pt, Superscript

| Page 74: [3] Deleted | Arora,Vivek [CCCMA] | 19/03/2020 12:48:00 PM |
|---|---|---|

Figures 6 and 7 provide justification for using the BGC-COU approach, over the RAD-BGC and

RAD-COU approaches, in calculating the feedback parameters as discussed below. In Figure 7, the absolute magnitude of $\gamma_O$ when using the BGC-COU approach is about twice in CMIP5 models (and more than three times in CMIP6 models) compared to its model-mean value calculated using the RAD-BGC and RAD-COU approaches. The reason for this is that the RAD simulation misses the suppression (due to weakening of the ocean circulation) of carbon drawdown to the deep ocean. This is because there is no buildup of a strong carbon gradient from the atmosphere to the deep ocean in the RAD simulation. This process is important when climate change is forced by increasing atmospheric $CO_2$, and therefore feedback parameters calculated using the BGC-COU approach are more likely to include all processes relevant to application for realistic scenarios. In Figure 6, although the carbon-climate feedback parameter over land ($\gamma_L$) is larger in absolute amount, it is comparatively less sensitive to the approach used, than over ocean, because over land an increase in temperature not only increases the respiratory losses but also affects photosynthetic processes especially in conjunction with increasing $CO_2$. Warmer temperatures increase photosynthesis over mid to high latitude regions where photosynthesis is currently limited by temperature and more so with increasing $CO_2$, but decrease photosynthesis over tropical regions where the temperatures are already too warm for optimal photosynthesis. The net result of these compensating processes plays out very differently in different models and in the model-mean sense this results in less sensitivity of the calculated value of carbon climate feedback parameter over land ($\gamma_L$) to the different approaches than over ocean. This is seen in both CMIP5 and CMIP6 models. When $\gamma_L$ is calculated using the RAD-BGC and RAD-COU approaches, it is exclusively calculated using results from the RAD simulation. However, since over land photosynthesis is also affected by temperature in addition to respiration (with widely varying responses between models) the $\gamma_L$ values vary widely between models between the RAD-BGC/RAD-COU approach and the BGC-COU approach. This is seen, for example, for ACCESS-ESM1.5, IPSL, and CanESM5 models in Figure 6b. The very different values of $\gamma_L$ for individual models, when using different approaches to calculate them, are the result of the differing responses of the vegetation and soil+litter carbon pools, in the RAD and COU simulations, and this is supported by results that were presented in Section 4.3.2.